# Dynamics of genome reorganization during human cardiogenesis reveal an RBM20-dependent splicing factory

Alessandro Bertero [1,2,3], Paul A. Fields [1,2,3], Vijay Ramani[4], Giancarlo Bonora [4], Galip G. Yardimci[4], Hans Reinecke[1,2,3], Lil Pabon [1,2,3], William S. Noble [4], Jay Shendure[4,5] & Charles E. Murry [1,2,3,6,7]

Functional changes in spatial genome organization during human development are poorly understood. Here we report a comprehensive profile of nuclear dynamics during human cardiogenesis from pluripotent stem cells by integrating Hi-C, RNA-seq and ATAC-seq. While chromatin accessibility and gene expression show complex on/off dynamics, large-scale genome architecture changes are mostly unidirectional. Many large cardiac genes transition from a repressive to an active compartment during differentiation, coincident with upregulation. We identify a network of such gene loci that increase their association inter-chromosomally, and are targets of the muscle-specific splicing factor RBM20. Genome editing studies show that *TTN* pre-mRNA, the main RBM20-regulated transcript in the heart, nucleates RBM20 foci that drive spatial proximity between the *TTN* locus and other inter-chromosomal RBM20 targets such as *CACNA1C* and *CAMK2D*. This mechanism promotes RBM20-dependent alternative splicing of the resulting transcripts, indicating the existence of a cardiac-specific *trans*-interacting chromatin domain (TID) functioning as a splicing factory.

[1] Department of Pathology, University of Washington, 1959 NE Pacific Street, Seattle, WA 98195, USA. [2] Center for Cardiovascular Biology, University of Washington, 850 Republican Street, Brotman Building, Seattle, WA 98109, USA. [3] Institute for Stem Cell and Regenerative Medicine, University of Washington, 850 Republican Street, Seattle 98109 WA, USA. [4] Department of Genome Sciences, University of Washington, William H. Foege Hall, 3720 15th Ave NE, Seattle 98195 WA, USA. [5] Howard Hughes Medical Institute, Seattle, WA, USA. [6] Department of Medicine/Cardiology, 1959 NE Pacific Street, University of Washington, Seattle 98195 WA, USA. [7] Department of Bioengineering, University of Washington, 3720 15th Ave NE, Seattle, WA 98195, USA. These authors contributed equally: Alessandro Bertero, Paul A. Fields. Correspondence and requests for materials should be addressed to C.E.M. (email: murry@uw.edu)

During human development diverse differentiated cell types emerge from a single pluripotent population of cells. This differentiation process requires mechanisms to both activate the gene networks of target lineages and repress those of alternative lineages. The human heart is the first major organ to form and thus in vitro modeling of cardiogenesis using human pluripotent stem cells (hPSCs) provides an ideal system to study the regulation of early fate commitment. Moreover, congenital heart disease, the most common human birth defect, is often caused by mis-regulated transcriptional pathways[1]. Spatial organization of the genome in 3-D has emerged as a key mechanism by which cells can control regions of chromatin accessibility[2,3]. Chromosome conformation capture technologies such as Hi-C have identified complex three-dimensional dynamics during development, and abnormal chromosomal architecture has been implicated in a range of diseases, including cancer and limb malformations[4–8]. While these studies have demonstrated regulated regions of chromosomes, how these transitions occur temporally across multiple stages of human lineage commitment remains to be addressed. Further, to what extent such changes causally impact gene expression is still poorly understood.

We have previously shown that epigenetic modifications are dynamically regulated during cardiomyocyte differentiation of hPSCs[9], and that integration of epigenetic data defines general patterns of gene regulation. Here we utilize Hi-C[10], which captures higher order chromatin structure, along with ATAC-seq, as a mark of local accessibility, along with RNA-seq during this process of differentiation. We demonstrate new insights into how global chromatin organization changes both coordinately and independently of local changes and how that feeds back on gene regulation. Further we find that genomic regions dynamically regulated during hPSC-cardiomyocyte differentiation resolve to a state that closely resembles that of human fetal cardiomyocytes, indicating these dynamics likely reflect early steps in human cardiogenesis. At the macro level, 19% of the genome changes active (A) vs. inactive (B) compartment state during differentiation. During differentiation the heterochromatic regions pack more tightly, increasing *cis* interactions. Transcriptional activation of large cardiac genes is associated with B to A transitions, chromatin decompaction and a gain in *trans* interactions. These properties are exemplified in the *TTN* (titin) locus, which encodes the largest human protein. In particular, we identify a network of *TTN*-associated genes co-regulated by the muscle-specific splicing factor RBM20[11] that increasingly interact in *trans* during differentiation. Cross-validation by imaging, and functional experiments using pharmacology or CRISPR/Cas9 gene editing indicate a mechanism whereby RBM20 nuclear foci are nucleated by *TTN* pre-mRNA and facilitate the interaction of target genes and proper alternative splicing. Overall, this study demonstrates the dynamic interplay between global and local chromatin architecture during human development and exemplifies how this can influence gene expression patterns.

## Results

### Chromatin structure dynamics during human cardiogenesis.
To understand the temporal dynamics of nuclear architecture during cardiac differentiation, we generated highly pure cardiomyocytes (CM; > 90% cTnT+) from undifferentiated RUES2 human embryonic stem cells (hESCs; Fig. 1a, Supplementary Fig. 1a–c). These cells pass through stages representative of early development including mesoderm (MES), and cardiac progenitor (CP), before reaching definitive CMs[9] (Supplementary Fig. 1d). We performed in situ DNase Hi-C[10] on these stages of differentiation with two independent biological replicates, along with two fetal heart samples (Supplementary Table 1).

Chromosome-wide contact maps demonstrate the expected checkerboard pattern, indicative of local associations (topologically associating domains, or TADs) and long-range compartmentalization (A/B compartments) (Fig. 1b). Genome-wide contact maps between whole chromosomes demonstrate that smaller and larger chromosomes tend to self-associate (Supplementary Fig. 2).

We computed the first principal component (PC1) from the contact matrix to segregate chromatin bins at 500 kb resolution into A/B compartments, which reflect regions of active and repressive chromatin, respectively[12]. Using t-SNE to visualize and cluster in two dimensions either PC1 scores or HiC-Rep scores[13] closely pairs replicates while generating a differentiation trajectory, demonstrating the reproducibility of the assay (Fig. 1c, Supplementary Fig. 3a, Supplementary Table 2). Fetal heart Hi-C most closely resembles in vitro cardiomyocytes but clusters separately, likely reflective of lower cardiomyocyte purity. Early fetal hearts, while consisting of ~70% cardiomyocytes[14], include other cell types such as fibroblasts. Overall the genome is split into ~50% A, ~50% B compartments at each time point (Fig. 1d), and there is little change in the distribution of compartment sizes across differentiation (Supplementary Fig. 3b). The majority of compartment assignments are invariant during differentiation (Fig. 1e–g). However, 19% of the genome changes compartment during differentiation, and hierarchical clustering of dynamic regions recapitulates the differentiation trajectory and clusters cardiomyocytes most closely with fetal heart (Fig. 1h). Most of these changes are unidirectional (B–A or A–B). A small subset exhibits a transitory switch, either A–B–A or B–A–B (Fig. 1g). Together these data show that A/B compartments are dynamic during cardiac differentiation, and that these changes are validated by analyses of fetal hearts.

By integrating the A/B compartment information across differentiation with the interaction contact maps, we noticed that many of the strongest gains in long-range intra-chromosomal (*cis*) interactions are associated with the B compartment (Supplementary Fig. 3c). This is consistent with prior studies in mouse neuronal specification[15], suggesting it may be a general phenomenon of differentiation. Genome-wide analysis shows that stronger *cis* contacts occur between homotypic regions (A–A or B–B compartments), compared to between heterotypic regions (A–B) (Supplementary Fig. 3d). In the pluripotent state, the strongest *cis* interactions occur between A compartments, while during differentiation this switches to favor signal between B compartments—a trend supported by patterns in fetal heart (Fig. 1i, Supplementary Fig. 3d, f). This switch occurs as a result of a gain in long-range (>10 Mb) B–B interactions during differentiation (Fig. 1j, Supplementary Fig. 3g), as seen in the contact map. In contrast, inter-chromosomal interactions (*trans*) favor A–A interactions independent of differentiation state and show no enrichment between B–B contacts (Supplementary Fig. 3e, f). Together these observations suggest a model whereby during differentiation heterochromatic (B compartment) regions condense and pack more tightly, specifically within chromosomes in *cis*, while inter-chromosomal interactions are most likely to occur between active regions (A compartment) independent of cell state. This is consistent with recent electron micrograph studies showing that heterochromatic regions are more densely packed relative to euchromatic regions in differentiated cells[16].

### A/B compartmentalization changes reflect expression dynamics.
To investigate genes whose expression may be regulated in relationship with compartment changes, we performed RNA-seq on the same stages of differentiation, along with the fetal hearts

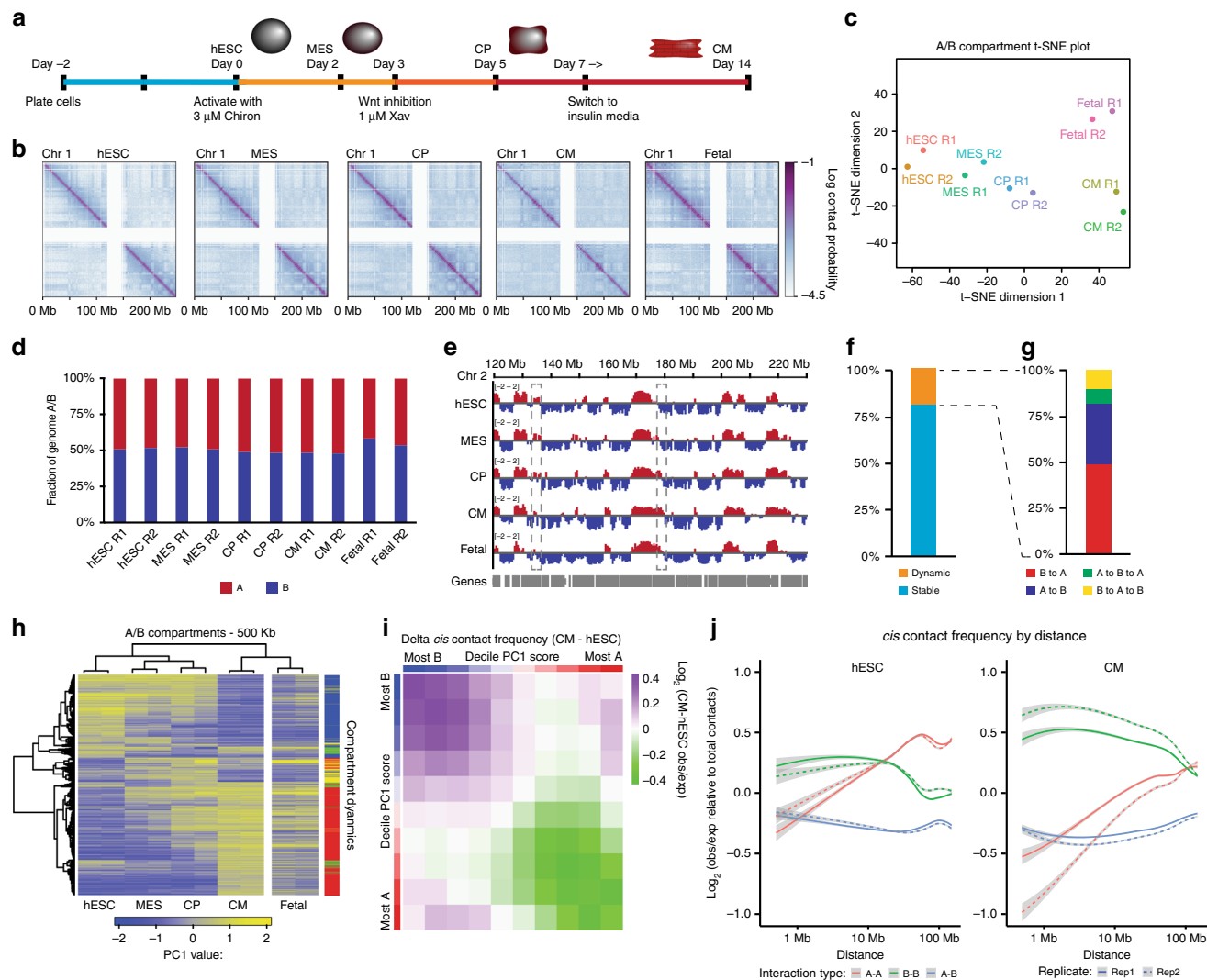

**Fig. 1** Hi-C across cardiac differentiation. **a** Schematic of the cardiomyocyte differentiation. **b** Log transformed contact maps of chromosome 1. **c** t-SNE plot of PC1 scores on the contact matrices. **d** Fraction of genome in A and B compartment by sample. **e** PC1 scores for a region of chromosome 2, gray boxes highlight regions transitioning from A to B and B to A. **f** Genomic regions divided by stable (81%) and switching (19%) A/B compartment (PC1 scores significantly different by one-way ANOVA, $p$-value < 0.05; $n = 2$ independent differentiations). **g** Regions switching A/B compartment divided by types of transitions. A–B (33%), B–A (49%), A–B–A (8%), B–A–B (10%). Two percent of switching regions were A–B–A–B or B–A–B–A and were combined with A–B and B–A. **h** Heatmap of the PC1 scores of the compartment switching regions. Clustering of rows based on the four time points of differentiation. Dendrogram of columns was ordered to match the temporal status of differentiation. **i** Delta compartmentalization saddle plot in *cis* contacts CM vs. hESC. Bins were assigned to ten deciles based on PC1 score, average observed/expected distance-normalized scores for each pair of deciles were calculated. **j** Distance plot of A–A, B–B, and A–B interactions for hESC and CM, values are normalized to all contacts at a given distance. Data was smoothed using R, raw maps in Supplementary Fig. 3g. Source data are provided as a Source Data file

(Supplementary Table 3). Similar to Hi-C, in principal component analysis samples separated by differentiation state and cardiomyocyte purity (Fig. 2a). In contrast to Hi-C, differentially expressed genes are largely expressed in a cell-type specific manner (Fig. 2b, Supplementary Fig. 4a). While RNA-seq reveals similarities between hESC-CMs and the fetal heart (Fig. 2b), the fetal heart has an increase in the ratio of adult to fetal myosin heavy chain genes (*MYH7* and *MYH6*), suggesting that these mid-gestational cardiomyocytes are more developmentally advanced (Supplementary Fig. 4a). Genes that are upregulated in CPs and CMs are enriched in regions that switch B to A, and underrepresented in A to B transitioning regions (Fig. 2c–e, Supplementary Fig. 4b–c). Gene ontology (GO) analysis reflects that genes upregulated in CMs are enriched in categories related to metabolism along with development and cardiac function (Supplementary Table 4). Focusing on those genes in regions that

transition from B to A highly enriches for genes involved in heart development, such as the structural gene alpha-actinin (*ACTN2*) (Fig. 2d–e, Supplementary Table 5). This suggests that many heart development genes may initially be sequestered in the B compartment and move to A upon activation. Noticeably, even though only 4% of the genome shows two-step dynamics (A–B–A or B–A–B; Fig. 1g), genes that are transiently in the A compartment show peak expression in the CP stage (Fig. 2c). Among these, *BMPER* and *CXCR4* have peak expression in CPs and have important roles in cardiomyocyte function (Fig. 2f, Supplementary Fig. 4d). Gene repression is also associated with compartment switching, including the mesoderm regulator *EOMES* which switches to B compartment upon downregulation at the CP stage (Supplementary Fig. 4e). Of note, A/B compartment dynamics were largely reproducible in a second hPSC line despite its slower cardiogenic differentiation (Supplementary Note 1 and

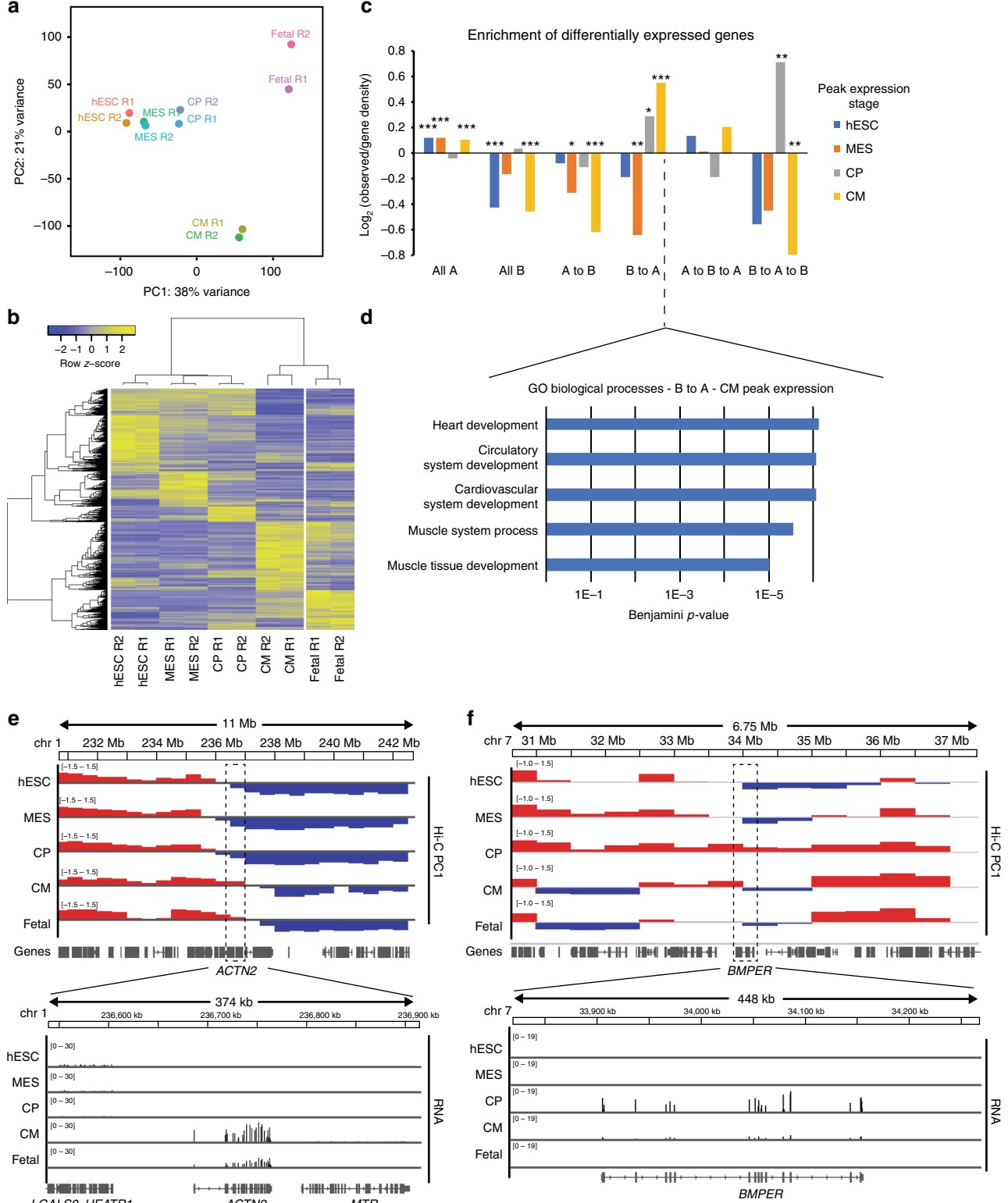

**Fig. 2** Dynamic gene expression correlations with changes in genome architecture. **a** PCA plot on the replicates of the RNA-seq samples. **b** Heatmap of the differentially expressed genes during differentiation, clustering of rows based on differentiation time points. **c** Enrichment of differentially regulated genes by time point of peak expression against A/B compartment dynamics. Log$_2$ values are observed/gene density; $p$-values by chi-squared test for the indicated gene set overlaps, * < 0.05, ** < 0.01, *** < 0.001. **d** GO term enrichment in CM peak expression genes in B to A compartments, $p$-value plotted on log scale. **e** Gene track of Hi-C PC1 and RNA-seq reads of *ACTN2*. **f** Gene track of Hi-C PC1 and RNA-seq reads of *BMPER*. Source data are provided as a Source Data file

Supplementary Fig. 5). Taken together, these observations show that dynamic compartment transitions in genomic regions occur coincident with transcriptional regulation, and the observed patterns are consistent between stem cell derivatives and the fetal heart, suggesting important developmental control.

**TAD dynamicity is independent from A/B compartmentalization.** While A/B compartments reflect the global organization of the genome across full chromosomes, local organization can be summarized as topologically associating domains (TADs). We utilized two methods to asses TAD structure across differentiation: directionality index[17] and insulation score[18] (Fig. 3a, Supplementary Fig. 6a, b, Supplementary Table 6). Using a stringent cutoff, both methods show that a majority of TAD boundaries are constitutive across differentiation (Fig. 3a). While genome-wide TADs range in size between approximately 500 kb and 1 Mb, TADs in the A compartment are smaller in size (Fig. 3b, Supplementary Fig. 6c), which is likely necessary for proper gene regulation in gene-dense areas. Comparing hESCs to CMs, boundaries lost during differentiation are enriched within regions that are constitutive B or switch from A to B (Fig. 3c,

Supplementary Fig. 6d). Consistent with prior studies[19], we find that TAD boundaries are located near to CTCF peaks in both hESCs and CMs with ~70% of TAD boundaries having a CTCF peak within 1 kb (Supplementary Fig. 6e). Moreover, CTCF peaks are located closer on average to constitutive boundaries rather than differential boundaries by both TAD methods (Supplementary Fig. 6e, chi-sq test, $p < 0.001$). Together with our previous observation that B domains compact during differentiation, this suggests that as regions compact TAD boundaries may be lost and adjacent regions condense together. In contrast, boundaries that are gained during differentiation are associated with a modest but significant activation of the nearest gene, but are not associated with a transition from inactive to active compartment (Fig. 3c, d, Supplementary Fig. 6f). Examples include, the loci of *LMO7* and *KCNN2*, which change local TAD structure coincident with upregulation of transcription (Fig. 3e and Supplementary Fig. 6g). Overall, the selective acquisition of new boundaries during differentiation is linked to control of gene expression largely independent of A/B compartment changes, while loss of boundaries is not associated with gene regulation and is predominantly associated with heterochromatin compaction.

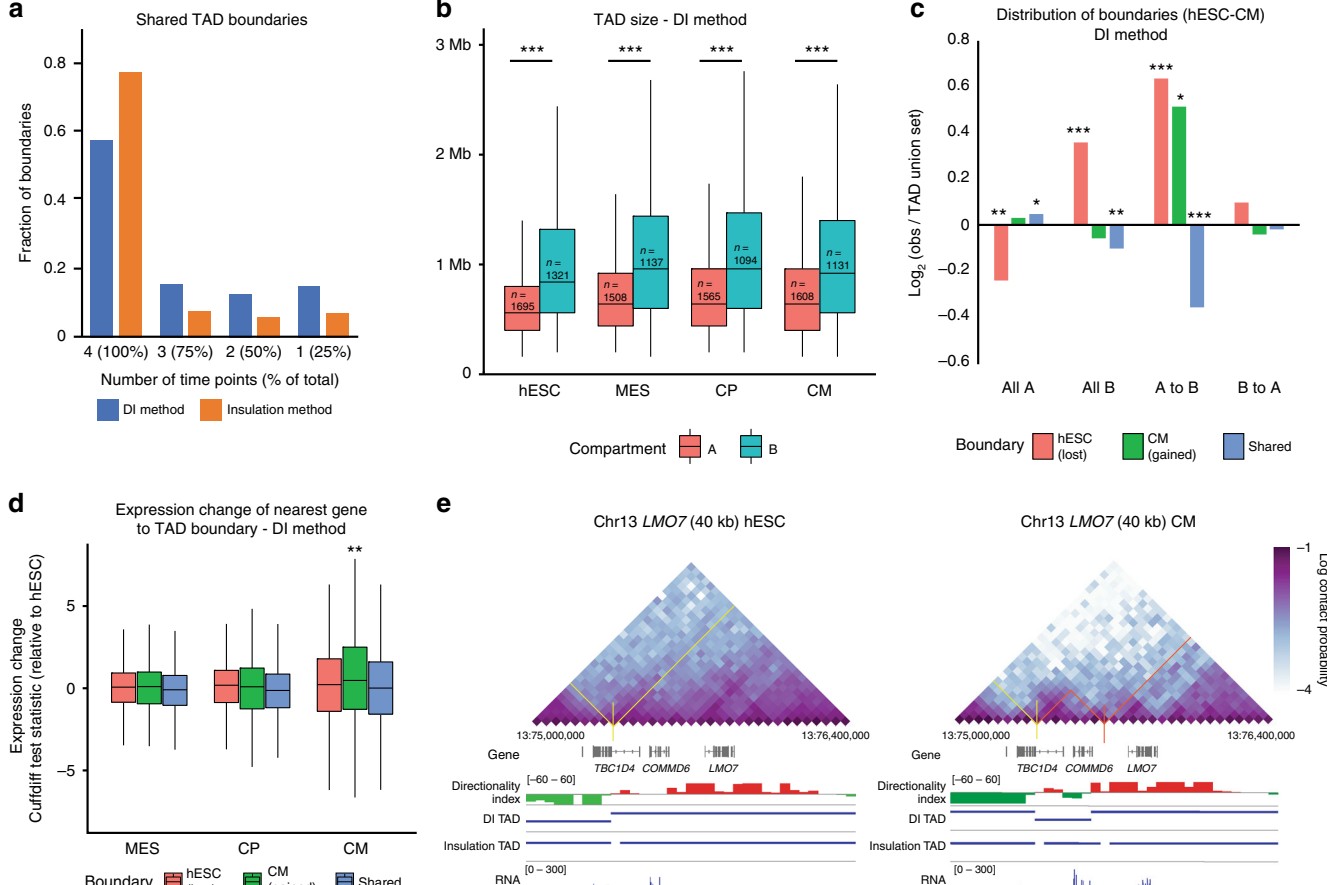

**Fig. 3** TADs are dynamically regulated independent of A/B compartment changes. **a** TAD boundaries shared between time points as calculated on the union set of all four time points using both DI and insulation method. **b** TAD size and number within A and B compartments across differentiation for DI method. Boxplots present the median and 25th and 75th percentile, with the whiskers extending to 1.5 times the inter-quartile range. $n$ = TAD number; $p$-values by Wilcoxon test, *** < 0.001. **c** Enrichment of TAD boundaries between hESC and CM state within A/B compartment dynamics for DI method. $Log_2$ values are observed/TAD union set; $p$-values by chi-squared test for the indicated TAD boundary set overlaps, * < 0.05, ** < 0.01, *** < 0.001. **d** Expression of nearest gene to TAD boundaries that are either stage specific or shared between hESC and CM for DI method. Box and whisker plots as in panel (**b**). $n$ = 308 genes for hESC (lost) boundaries, 207 genes for CM (gained) boundaries, and 1326 genes for shared boundaries; $p$-values by one-sample, two-sided $t$-test relative to a hypothesized mean value of 0 (no expression change vs hESC), ** < 0.01. **e** Gene track of DI score, DI-determined TADs and insulation score-determined TADs and RNA, along with a Hi-C heatmap of the *COMMD6* and *LMO7* locus in hESC and CM. Source data are provided as a Source Data file

**Chromatin accessibility and 3D organization are correlated**. To determine whether changes in higher order organization occur coincident with local changes, we performed ATAC-seq to measure local accessibility across differentiation (Supplementary Fig. 7a, b, Supplementary Table 7). Similar to our previous study using DNase hypersensitivity[20], we find a decreasing number of peaks during differentiation (Fig. 4a). During differentiation, there is an increase in the fraction of peaks in A compartment, while peaks being lost are enriched in regions that are constitutively B compartment (Fig. 4b, c, Supplementary Fig. 7c, d). This supports the model of increased heterochromatin packing in CM coincident with decreased accessibility. Similar to the gene expression data, we find that CM-specific ATAC-seq peaks are enriched in regions that transition from B to A and are depleted in A to B regions (Fig. 4c). While we also observe an enrichment of hESC-specific peaks in B to A regions, this is less strong than CP- or CM-specific enrichment and is likely an effect of the overall more open chromatin in hESCs. Stage-specific peaks are on average more distant from TSSs and enriched in motifs corresponding to developmental transcription factors, consistent with possible enhancer activity (Supplementary Fig. 7e, f), in agreement with another recent study[21]. Interrogating the regions that transition from B to A, these are enriched for peaks present in CP and CM stages and depleted in constitutive peaks (Fig. 4d). The top motifs within these regions match important cardiogenic transcription factors (TF) of the GATA and NKX family (Fig. 4e), which may play pivotal roles in opening up the chromatin during differentiation.

To explore this possibility in more detail we integrated chromatin organization changes with the binding profiles of these TF. We took advantage of published ChIP-seq datasets from in vitro-differentiated CMs for GATA4[22], NKX2-5[23], and TBX5[22] (another known cardiac TF), as well as hESC ChIP-seq data for GATA4[24]. We find that only the CM ChIPs were enriched in regions that transition from B to A compartment (Fig. 4f). Furthermore, we find a modest but significant increase in the overlap of GATA4 peaks with both NKX2-5 and TBX5 in B to A regions compared with the rest of the genome (Fig. 4g). These observations suggest a cooperative role for these TF in driving chromatin transitions. Examining the enrichment of reads across stage-specific ATAC-seq peaks we find that the CM ChIPs are exclusively enriched at CM-specific peaks (Fig. 4h and Supplementary Fig. 7g). In contrast, GATA4 binding in hESCs is highly enriched in constitutive peaks but not hESC-specific peaks (Fig. 4h), suggesting it may mainly have "housekeeping" roles in this cell type. Given that GATA and NKX motifs were enriched in B to A regions in both CM and CP-CM peaks we examined the enrichment of TF binding across these peaks. Remarkably, GATA4 shows comparable enrichment at CP-CM and CM peaks, while NKX2-5 and TBX5 have greater enrichment at CM-specific peaks. This suggests that GATA4 may be an earlier driver of chromatin transition, perhaps working as a "pioneering" TF, while NKX2-5 and TBX5 may act to further strengthen and reinforce this GATA4-dependent transcriptional network (Fig. 4i). Example of genes co-bound by these TFs are the important sarcomeric proteins *NEBL* (Fig. 4j) and *ACTN2* (Fig. 2e, Supplementary Fig. 7h). Notably, both of these genes transition from B to A compartment during differentiation. Thus, local changes in chromatin accessibility occur coincident with changes in RNA expression and large-scale genome organization.

**Compartmentalization controls large cardiac-specific genes**. Given the strong enrichment of CM-upregulated genes in the regions of the genome that transition from B to A, we sought to determine whether there are specific characteristics of these genes. Regions that transition from B to A have lower gene density (Supplementary Fig. 8a), suggesting these genes may be bigger or more isolated and therefore exert a stronger influence on their region of the genome. Indeed, upregulated genes in B to A regions are larger than other upregulated genes, independent of GO term (Fig. 5a). In contrast, heart development genes show greater separation from the nearest gene promoters (start site to start site), independent of compartment dynamics or gene size (Supplementary Fig. 8b, c). Using expression data from 37 adult tissues[25] we find that those genes that are upregulated and go from B to A are more heart-enriched compared with other upregulated genes (Supplementary Fig. 8d). Together, this supports a model in which expression of large lineage-specific cardiac genes is regulated by changes in chromatin architecture, as they are initially insulated in B compartment and transition to A upon activation. On the other hand, other developmental genes are regulated by different mechanisms that act more locally, which is facilitated by their promoter isolation.

**Dynamic regulation of *TTN* by 3D structural changes in *cis***. A prominent example of the regulation of large cardiac genes just described is titin, the largest protein in the human genome, which when mutated is the leading cause of familial dilated cardiomyopathy (DCM)[26]. The *TTN* locus transitions from B to A, coincident with an opening up of the local chromatin structure and upregulation of transcription (Fig. 5b, c). There is a corresponding increase in accessibility at the two primary *TTN* TSSs, both the full-length transcript as well as a shorter isoform, termed Cronos *TTN*[27] (Fig. 5d). Both isoforms have important developmental roles as mutations downstream of the Cronos TSS are significantly more deleterious than those only in the long isoform[27]. Both TSS-associated ATAC peaks are marked by the cardiac TF GATA4, NKX2-5, and TBX5 (Fig. 5d). Interestingly, we also observe peaks of accessibility within *TTN* that decrease during differentiation, all of which overlap CTCF peaks[28]. Using CTCF ChIP-seq data from hESCs and CMs[28], we find that the CTCF sites that overlap hESC ATAC peaks show decreased occupancy during differentiation. With the exception of one CTCF site located at the transcriptional termination site, the CTCF motifs on *TTN* are orientated in a convergent direction, predictive of the presence of intragenic looping[29]. Indeed, during differentiation there is a decreased interaction between the *TTN* TSS and the *TTN* gene body, while there is little change involving the upstream region (Fig. 5e). Together this suggests that intragenic looping may be a mechanism to maintain silencing of *TTN* within the B compartment in hESCs, and that this is potentially mediated through CTCF[30]. Genome-wide we find a much greater overlap of hESC-specific ATAC peaks and hESC CTCF binding, compared with ATAC peaks that are CM-specific and CM CTCF binding, and this trend is not dramatically different between regions that go B to A and the rest of the genome (Fig. 5f). Thus while cardiac transcription factors are enriched in gained ATAC peaks and in B to A regions (Fig. 4g), CTCF is enriched in hESC-specific peaks and may be important to maintain differentiation permissive chromatin.

**TTN interacts in *trans* with multiple RBM20-regulated loci**. Having analyzed in detail the regulation of the *TTN* locus in *cis*, we explored the possibility that its genomic interactions with other chromosomes (in *trans*) could also be of regulatory importance. Interestingly, coincident with *TTN*'s move to the A compartment there is a switch in *trans* interactions from an enrichment with B compartment to A (Fig. 6a, hESC median Z-score A: −0.26 B: −0.07, CM median Z-score A: 0.19, B: −0.29). Genes upregulated and within the enriched *trans* contacts include

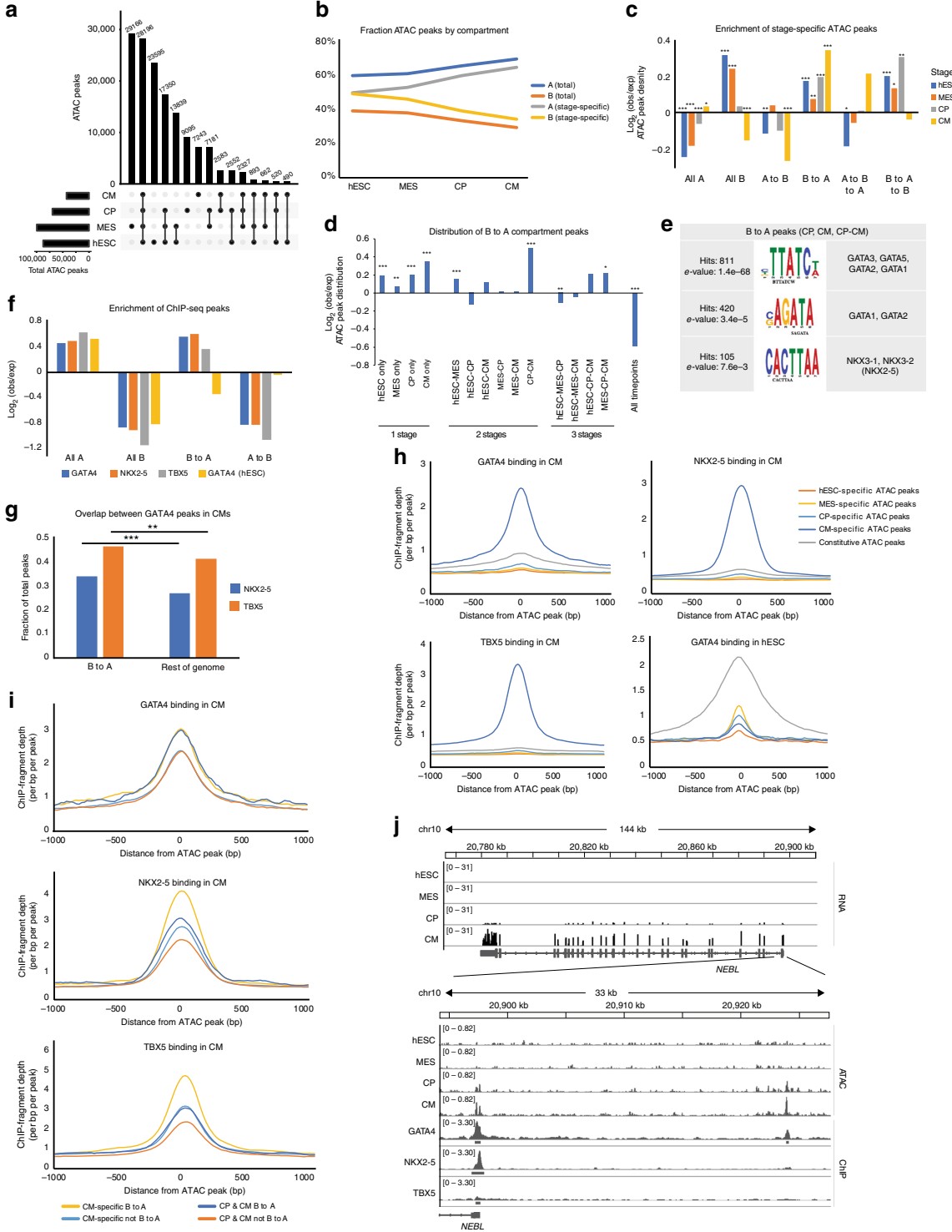

**Fig. 4** Changes in local chromatin accessibility occur coincident with global architecture changes. **a** ATAC peaks across differentiation. **b** Fraction of ATAC peaks divided by A and B compartment for stage specific and total peaks. **c** Enrichment of stage-specific ATAC peaks within A/B compartments. Log₂ values are observed/ATAC peak density; *p*-values by chi-squared test for the indicated ATAC peak set overlaps, * < 0.05, ** < 0.01, *** < 0.001. **d** Enrichment of peaks within the B to A compartment regions by stage specificity, normalized to ATAC peak density. Plot and statistical analysis as in panel (**c**). **e** Motifs from DREME and TOMTOM within CP and CM-specific peaks in B to A compartment bins. **f** Enrichment of GATA4, NKX2-5, TBX5 (CM), and GATA4 (ESC) ChIP-seq peaks within constitutive A and B regions and B–A and A–B regions. **g** Overlap of GATA4 peaks with NKX2-5 and TBX5 in B–A regions or the rest of the genome. *p*-values by chi-squared test for the indicated ChIP-seq peak set overlaps, *** < 0.001. **h** Metaplot of GATA4, NKX2-5, TBX5 (CM), and GATA4 (ESC) binding across stage specific and constitutive ATAC peaks. Peaks are centered at the mid-point of ATAC peaks and extended ± 1000 bp. **i** Metaplot of GATA4, NKX2-5, TBX (CM) across CM or CP&CM-specific peaks in B–A regions or the rest of the genome. **j** Gene track of RNA-seq, ATAC-seq and ChIP-seq reads for *NEBL* gene and promoter region. Source data are provided as a Source Data file

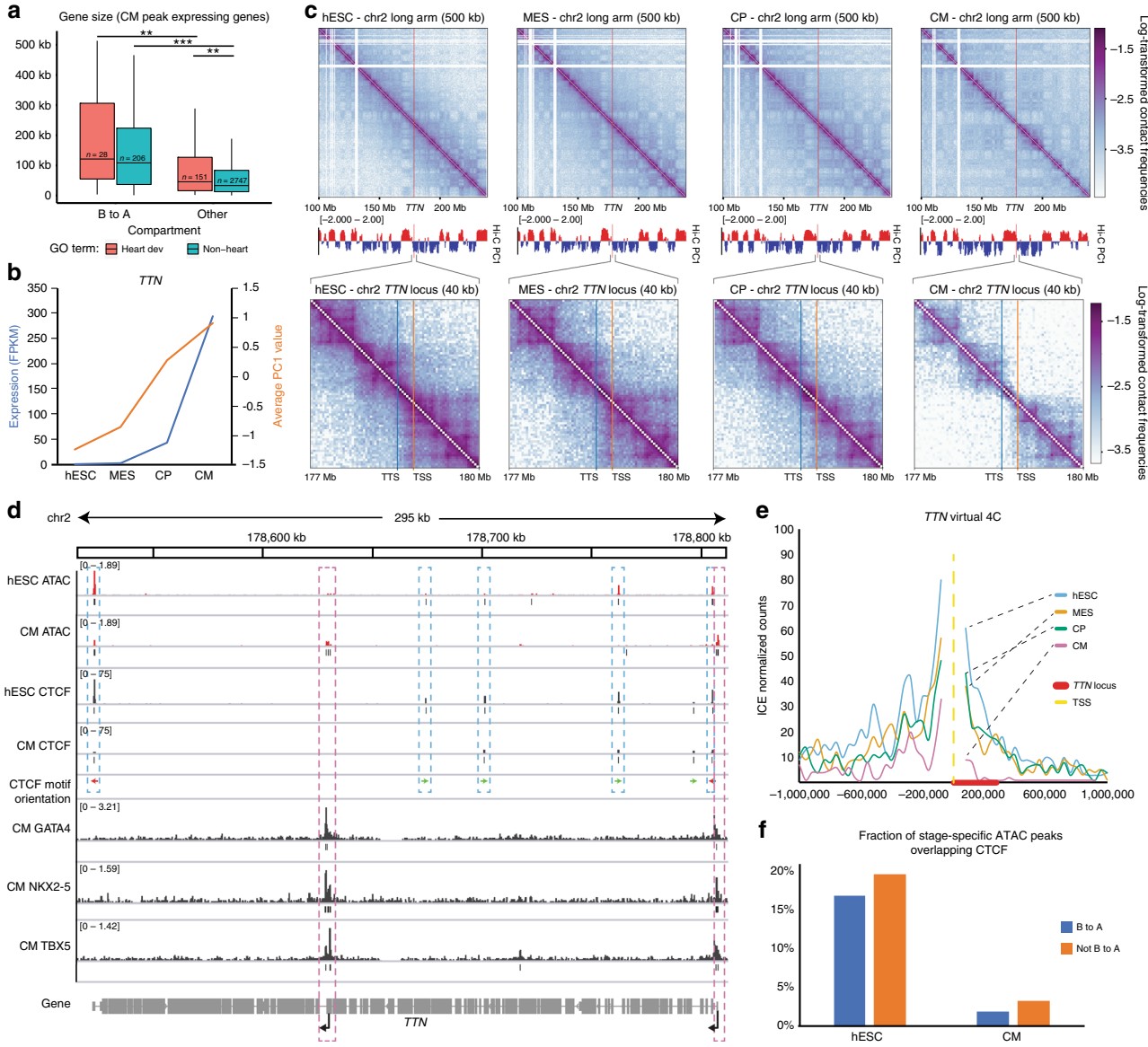

**Fig. 5** *TTN* locus is spatially regulated during differentiation. **a** Gene size of upregulated genes peaking in CM stage subdividing either B to A compartment or heart development genes (GO term). Boxplots present the median and 25th and 75th percentile, with the whiskers extending to 1.5 times the inter-quartile range. *n* = gene number; *p*-values by Wilcoxon test, ** < 0.01, *** < 0.001. **b** PC1 value of the *TTN* locus and the FPKM expression levels across differentiation. **c** Log transformed contact maps of the long arm of chromosome 2 containing the *TTN* locus at 500 kb resolution and a zoomed in view at 40 kb resolution, along with the PC1 values. **d** Gene track of the ATAC-seq signal, CTCF ChIP-seq signal (ESC and CM) and motif orientation, and GATA4, NKX2-5, and TBX5 (CM) ChIP-seq signal across the *TTN* locus. Dynamic ATAC peaks are highlighted, including increase (in red) at the promoters and decreases (in blue). **e** Virtual 4C of the *TTN* promoter at 40 kb resolution. **f** Fraction of stage-specific ATAC peaks (ESC and CM) overlapping CTCF peaks in the corresponding time point in B–A regions and the rest of the genome. Source data are provided as a Source Data file

factors involved in cell surface signaling and muscle contraction (Supplementary Table 8). Two of those *TTN*-associated cardiac genes are *CACNA1C* (which encodes the central subunit of the L-type calcium channel) and *CAMK2D* (which encodes a calcium/calmodulin dependent kinase that regulates excitation–contraction coupling; Supplementary Fig. 8e). All of these genes are subject to extensive alternative splicing, and are known targets of the muscle-specific splicing regulator RBM20[11] (Fig. 6b). RBM20 controls tissue-specific isoform expression of several cardiac proteins mainly by promoting exclusion of selected exons or by controlling mutually exclusive exon selection, which is mediated by direct binding of RBM20[11,31,32]. Mutations in RBM20 are a relatively common cause of familial DCM, indicating its key importance in cardiac homeostasis[33]. Notably, while

RBM20 is present throughout the interphase nucleus of muscle cells, it forms two clearly identifiable foci which overlap with newly synthetized *TTN* mRNA[32]. Thus, we hypothesized that the interaction of *TTN* with *CACNA1C* and *CAMK2D* could indicate the existence of a broader network of *trans*-interacting RBM20 targets.

To computationally test this, we focused on a set of 16 CM-upregulated genes whose spicing is regulated by RBM20 in both human and rat hearts[11]. These genes show a significant Hi-C association in CMs but not in hESCs (Fig. 6c–e). To control for the possibility that this could be simply the result of compartment switching, we repeated the analysis with only those genes that are constitutively in the A compartment against a comparable background gene set and still observed a gain in association

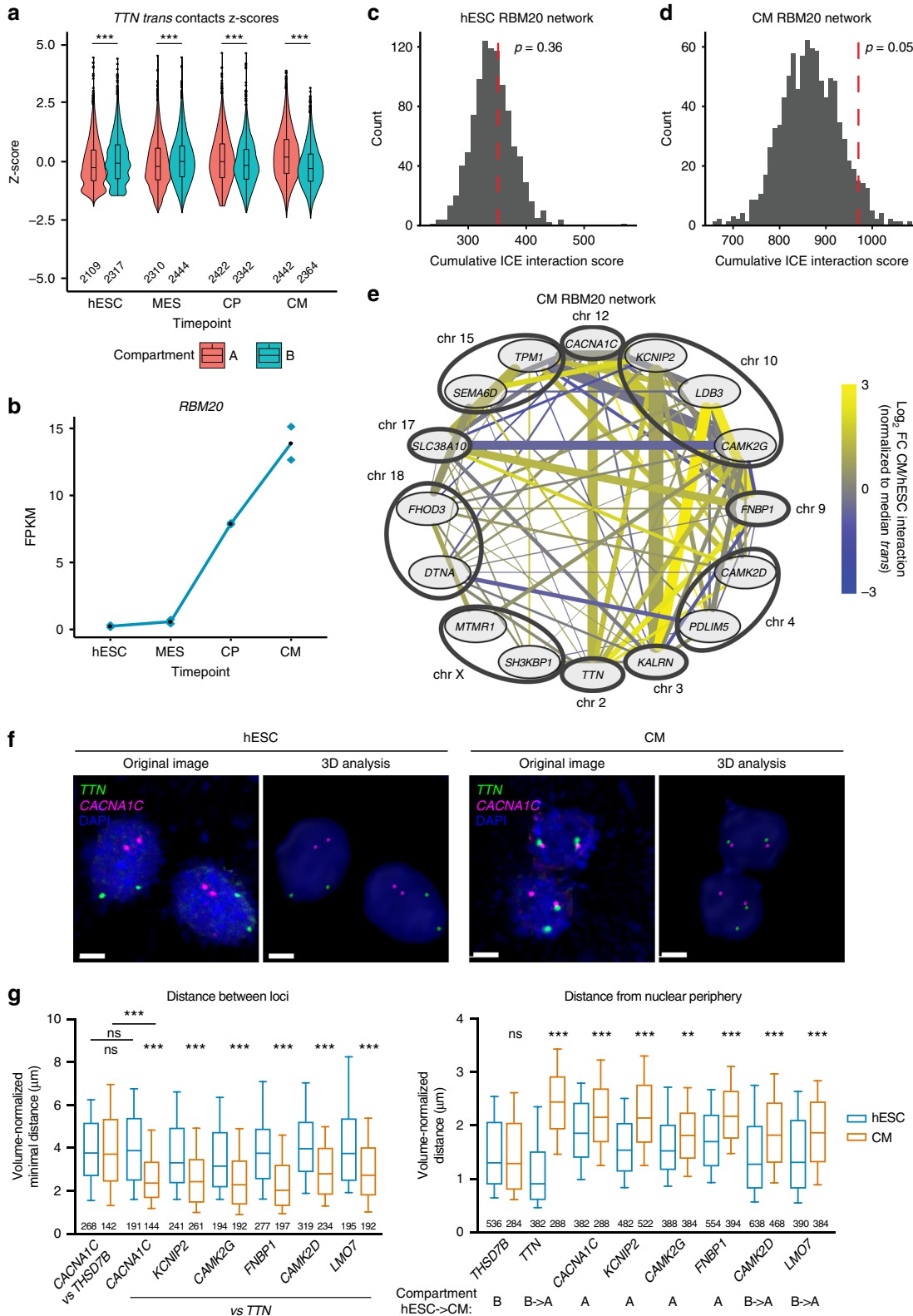

(Supplementary Fig. 8f). We validated these findings by 3D DNA FISH for *TTN* and its top five *trans*-interacting genes within the RBM20 network (*CACNA1C*, *CAMK2D*, *CAMK2D*, *KCNIP2*, and *FNBP1*; Fig. 6f, g). We also included *LMO7* because while this gene was not present in the set of RBM20 targets we originally considered, it is a well-established RBM20 target[34] which shows strong interaction with *TTN* in CM. Remarkably, each of these

loci moves significantly closer to the *TTN* locus when hESCs differentiate into CMs (Fig. 6f, g and Supplementary Fig. 9a). Accordingly, these loci were found in proximity to *TTN* in 33–58% of all CM (compared with 8–24% in hESCs; Supplementary Fig. 9a). On the contrary, differentiation does not affect the distance between *CACNA1C* and *THSD7B*, a negative control region on the same chromosome as *TTN* but located in a

**Fig. 6** *TTN* locus becomes associated with RBM20 target genes. **a** *Trans* contacts of *TTN* by compartment, Z-scored by time point at 500 kb resolution. Boxplots within violin plots present the median and 25th and 75th percentile, with the whiskers extending to whole data range. The number of genomic bins is indicated; p-values by Wilcoxon test, *** < 0.001. **b** FPKM values for *RBM20* across differentiation. **c**, **d** Association of upregulated *RBM20* target genes in hESCs (**c**) and CM (**d**). Dashed red line indicates cumulative sum of ICE normalized Hi-C reads between target genes in *trans*. Histogram represents the background model of 1000 random permutations of selected genes from similar chromosomal distribution. The resulting random shuffling p-value is indicated. **e** Network of upregulated RBM20 genes, line thickness proportional to contact score, only scores greater than median *trans* contact displayed. **f** Representative 3D DNA FISH images in hESC and CM, and 3D reconstructions after spot calling processing (nuclei counterstained with DAPI); scale bars: 5 μm. **g** On the left, normalized minimum distance per diploid cell between the indicated loci (the number of cells is indicated). On the right, normalized distance of each locus from the nuclear periphery (the number of loci is indicated); A/B compartment transitions based on Hi-C data are reported. Box and whiskers plots present aggregated data from two independent cultures, and indicate median, 25th and 75th percentile, and the 10–90 percentile range. p-values by Kruskal–Wallis test followed by Dunn's multiple comparisons vs hESC (unless otherwise indicated); ns ≥ 0.05; ** < 0.01; *** < 0.001. Source data are provided as a Source Data file

constitutive B domain ~40 Mb away (Fig. 6g). This indicates that the increased vicinity between *TTN* and *CACNA1C* in CM does not result from mere changes in the position of chromosome territories. Interestingly, these same 3D FISH experiments showed that the position of *TTN* and the other tested loci relative to the nuclear periphery closely correlate with their compartment status as predicted by Hi-C data (Fig. 6g). In particular, during differentiation *TTN* moves from the nuclear periphery to the interior, suggesting that this resides in a heterochromatic lamin-associated domain in hESCs, and moves into euchromatic nucleoplasm coincident with its B to A transition in cardiomyocytes (Fig. 6f, g). *CAMK2D* and *LMO7* show a similar strong change in localization which matches their movement from B to A, while there is only a limited effect on the distance from the nuclear periphery for genes found always in the A compartment. Collectively, these observations indicate that as multiple RBM20 targets are upregulated they migrate toward the center of the nucleus, where they increase their proximity to the *TTN* locus.

**_TTN_ _trans_ contacts need transcription-dependent RBM20 foci**. To functionally test the hypothesis that RBM20 forms a splicing factory involving multiple loci found on different chromosomes, we first took advantage of pharmacology. It was previously shown that RBM20 foci in HL-1 immortalized mouse cardiomyocytes are transcription-dependent[32], which offers an avenue to test the effect of their disruption. We confirmed that RBM20 forms foci also in CM, and that these closely co-localize with the *TTN* loci (Fig. 7a, b). Treatment with the transcription inhibitor Actinomycin D leads to a rapid loss of such RBM20 foci in most CM (Fig. 7a–c). Remarkably, this correlates with a strong reduction in the proximity between *TTN* and both *CACNA1C* and *CAMK2D*, while the localization of these loci relative to nuclear periphery was either unaffected or only modestly affected (Fig. 7d, e and Supplementary Fig. 9b). Thus active transcription is necessary for the establishment and/or maintenance of these *trans* interactions, but not for localization of the individual loci within the nucleoplasm.

**The _TTN_ pre-mRNA nucleates an RBM20 splicing factory**. To further probe the mechanism behind the formation of RBM20 foci and of the *trans* chromatin interactions involving *TTN*, we generated a number of gene-edited lines using CRISPR/Cas9 (Supplementary Fig. 10a–h). First, we deleted the *TTN* promoter to test the effect of abolishing its transcription (Fig. 8a; referred to as *TTN* ΔProm). As a control, we inserted a nonsense mutation in the first coding exon of *TTN*, which is predicted to interfere only with mRNA translation but not with *TTN* transcription nor splicing (Fig. 8a; referred as *TTN* KO). We confirmed that both mutations abolish titin protein expression (Fig. 8b), while only

*TTN* ΔProm impairs steady state mRNA levels (Fig. 8c). Remarkably, while RBM20 is expressed at normal levels in *TTN* ΔProm and localized throughout the nucleus, the number of RBM20 foci is strongly reduced (Fig. 8c–e). On the contrary, the localization of RBM20 is normal in *TTN* KO CM (Fig. 8c–e). This indicates that the *TTN* mRNA is specifically required for the nucleation of RBM20 into foci, which can be explained by fact that *TTN* contains the highest number of RBM20 binding sites among all of its targets (more than 100 locations compared with an average of ~1.7 for other genes[34]).

As a second genetic model, we knocked out RBM20 itself (Fig. 9a). Previous studies have examined the effect of DCM-causing heterozygous point mutations in *RBM20* using hPSC-CM[35,36], but the functional effect of complete RBM20 loss of function in this context remains unknown. By targeting the second exon of *RBM20* with two independent single guide RNAs (sgRNAs) we abolished nuclear expression of RBM20 in CM (Fig. 9b). While the RBM20 antibody showed some perinuclear staining in *RBM20* KO CM, western blot indicated complete loss of full-length RBM20 with no detectable truncated protein (Fig. 9c and Supplementary Fig. 11). To validate the knockout from a functional standpoint we examined the expression of *TTN* splice variants. Two major developmentally-regulated *TTN* isoform exists: the fetal N2BA isoform, which includes a number of alternatively spliced exons encoding PEVK elastic domains and leads to a long and compliant protein, and the adult N2B isoform, which lacks such PEVK exons and is stiffer[37]. This developmental switch is physiologically important to increase the stiffness of the developing myocardium as it sustains higher blood pressure. Previous findings in Rbm20 knockout rats and mice demonstrated that Rbm20 is required for the exclusion of PEVK exons[11,38], and that complete loss of function of Rbm20 leads to expression a non-physiological giant form of *TTN*, named N2BA-G. We confirmed that while *RBM20* KO CMs express normal levels of *TTN* mRNA, the transcript is entirely represented by the N2BA-G isoforms, while the N2B isoform is undetectable (Fig. 9d–f). We also examined alternative splicing of *CACNA1C* and *CAMK2D*, and confirmed that *RBM20* KO affects the choice of alternative exons for both transcript (Fig. 9g–j). For *CACNA1C*, this results in inclusion of exon 9*, which is associated with hyperpolarization of L-type calcium channel[39]. For *CAMK2D*, this leads to an increased ratio of the CaMKIIδ$_A$ isoform, which is found at the intercalated disks and T tubules, over the CaMKIIδ$_B$ isoform, which is nuclear localized[40]. Overall, these findings validated the functional knockout of RBM20 in CM.

Having developed these cellular models, we tested the effect of each genetic perturbation with regards to the interaction of *TTN* with RBM20 target loci. Remarkably, we observed that *TTN*'s interaction with *CACNA1C* and *CAMK2D* is significantly reduced in both *RBM20* KO and *TTN* ΔProm CM, while *TTN* KO CM

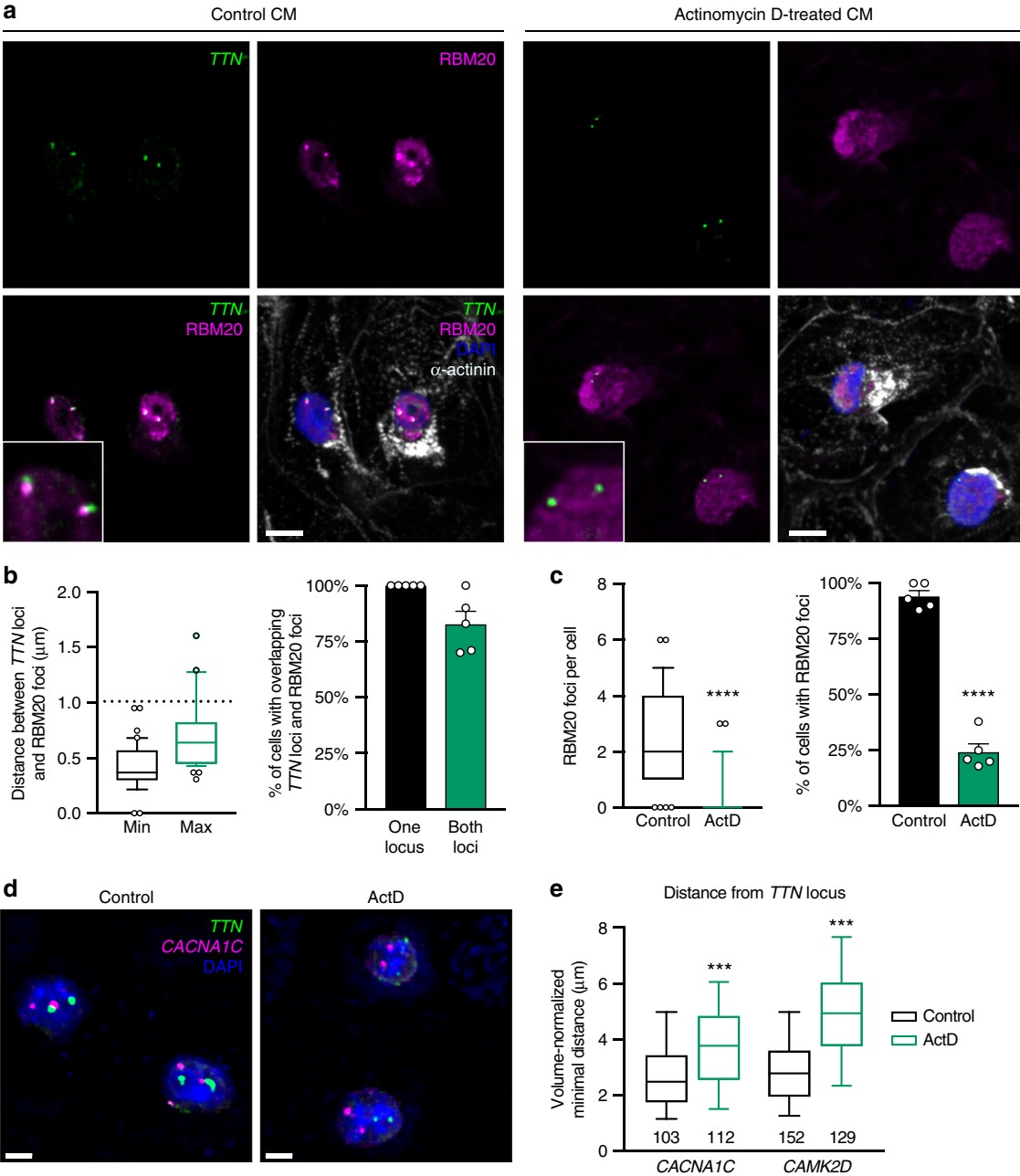

**Fig. 7** *TTN* trans interactions are transcription-dependent. **a** Representative results of immunofluorescence for RBM20 and α-actinin combined with DNA FISH for the *TTN* locus in CM (immunoFISH; nuclei counterstained with DAPI); scale bars: 10 μm. Insets show magnified views. Cells were maintained in standard culture conditions or treated with 5 μM Actinomycin D. **b** Quantification of distance relationships between RBM20 foci and *TTN* loci in control conditions. Center-to-center distance below 1 μM (twice the radius of 3D-rresconstructed spots) was used to determine overlap (see Methods). On the left, box and whiskers plots showing median, 25th and 75th percentile, and the 10–90 percentile range plus outliers; $n = 40$ diploid cells. On the right mean ± s.e.m.; $n = 5$ field of views. **c** Quantification of RBM20 foci in control and Actinomycin D-treated CM. Graphs are as described for panel (**b**) except that on the left $n = 64$ and 55 cells for CTR and ActD, respectively. Note that all CM were considered, including polyploid cells with more than two RBM20 foci. *p* values by Mann–Whitney test (left) or Welch's *t*-test (right); **** $< 0.0001$. **d** Representative 3D FISH images of control and Actinomycin D-treated CM; scale bars: 5 μm. **e** Normalized minimum distance per diploid CM between *TTN* and the indicated loci (the number of cells is reported). Box and whiskers plots show median, 25th and 75th percentile, and the 10–90 percentile range. *p*-values by Kruskal–Wallis test followed by Dunn's multiple comparisons vs Control; *** $< 0.001$. Source data are provided as a Source Data file

behave similarly to wild types (Fig. 10a, b and Supplementary Fig. 9c). Further, the localization of these loci relative to the nuclear periphery is either unaffected or only modestly affected, indicating that RBM20 is not required for the transition from the nuclear periphery (Supplementary Fig. 9c). Thus, these multiple

*trans* chromatin interactions involving the *TTN* locus require both the expression of RBM20 and the transcription of *TTN* mRNA to nucleate RBM20 into foci.

Finally, we investigated whether disruption of this mechanism affects alternative splicing of RBM20 targets by profiling RBM20-

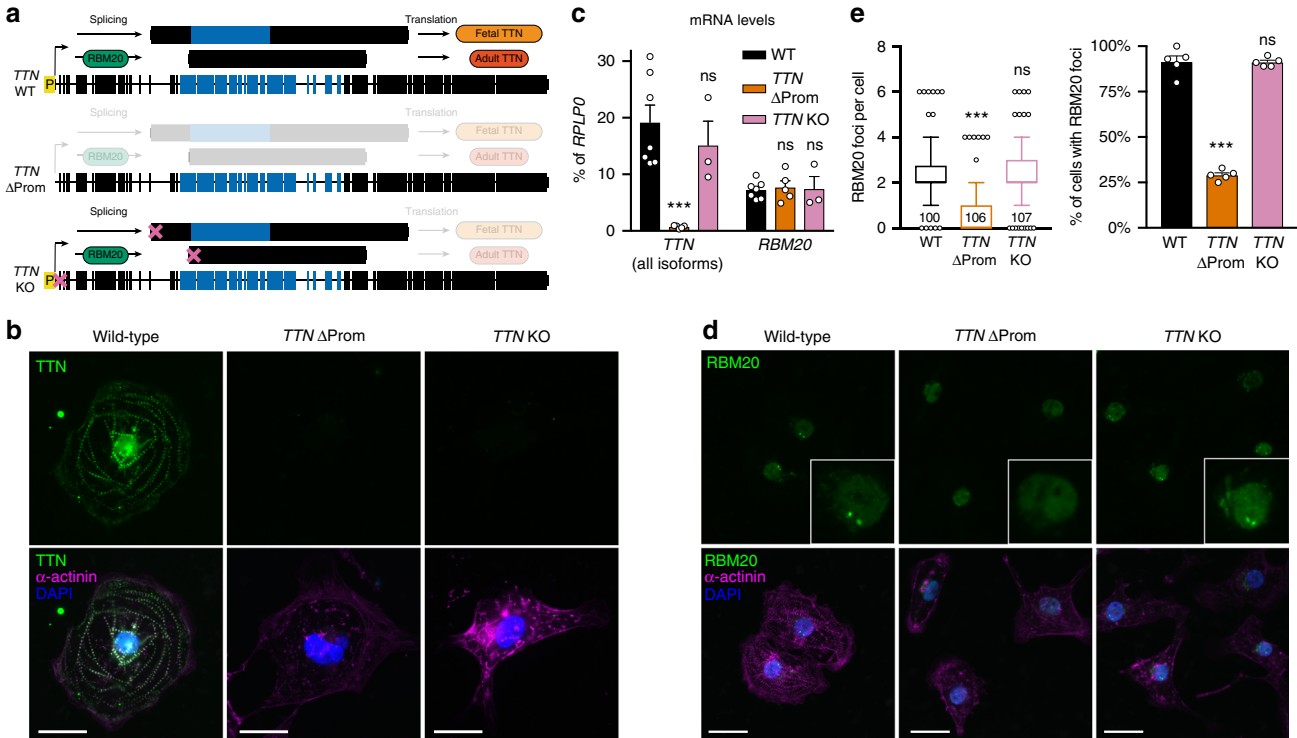

**Fig. 8** *TTN* transcription is required for nucleation of RBM20 into foci. **a** Schematic of the hESC gene editing strategies used to test the role of transcription at the *TTN* locus. ΔProm: promoter (P) deletion; KO: functional knockout by frameshift mutation (indicated by the X). Aspects of *TTN* biogenesis predicted to be impaired by each modification are indicated by increased transparency (loss of transcription following promoter deletion; loss of translation following knockout). Alternatively spliced exons regulated by RBM20 are shown in blue. **b** Representative immunofluorescence results in CM derived from the indicated hESC lines (nuclei counterstained with DAPI); scale bars: 10 μm. **c** RT-qPCR in CM derived from the indicated hESC lines. Expression is relative to the housekeeping gene *RPLP0* and mean ± s.e.m. is shown. $n = 3$, 5, or 7 independent differentiations for *TTN* KO, *TTN* ΔProm, and WT, respectively. **d** As in panel (**b**); insets show magnified views. **e** Quantification of RBM20 foci in CM derived from the indicated hESC lines. On the left, box and whiskers plots showing median, 25th and 75th percentile, and the 10–90 percentile range plus outliers; the number of cells is indicated (non-diploid CM were included in the analysis). On the right mean ± s.e.m.; $n = 5$ field of views. All *p*-values in this figure are calculated vs WT by one-way ANOVA followed by Holm-Sidak's multiple comparisons (for (**c**) and the right graph in (**e**)) or by Kruskal–Wallis test followed by Dunn's multiple comparisons (for the left graph in (**e**)); ns ≥ 0.05; *** < 0.01. Source data are provided as a Source Data file

dependent isoform expression in *TTN* ΔProm CM. Notably, both *CACNA1C* exon 9* and CaMKIIδ_A are upregulated in these conditions, compared with both wild-type CM and *TTN* KO CM (Fig. 10c, d). Based on these collective observations, we conclude that *TTN*, *CACNA1C*, and *CAMK2D* are part of a RBM20-dependent *trans*-interacting chromatin domain, and that this mechanism controls their alternative splicing (Fig. 10e).

## Discussion
This study provides a comprehensive integration of Hi-C, RNA-seq, and ATAC-seq across a defined time course of human hPSC differentiation into cardiomyocytes, which demonstrates the dynamic interplay of local and global chromatin structure on gene regulation during cell fate transitions. Moreover, we confirm that in vitro derived human cardiomyocytes are similar to their in vivo counterparts with respect to genome architecture and gene expression, supporting their use in developmental and disease modeling. We find that ~19% of the genome transitions either from active (A) to repressive (B) compartment or the reverse. Regions that transition from B compartment to A include many large cardiac genes, and this transition occurs coincident with an increase in local accessibility and increased transcription. While we do not find a strong association between genes that transition from A to B compartment with downregulation, this

could reflect a mechanism to permanently silence genes related to alternative lineages which were never activated. These findings are in line with recent Hi-C analyses during mouse cardiogenesis[41], indicating that dynamic compartmentalization is a conserved mechanism for gene regulation. Active regions are associated both in *cis* and *trans* and have high chromatin accessibility and transcription throughout differentiation. In contrast, heterochromatin is relatively accessible in hESCs compared with differentiated cells, but this compacts during differentiation. Heterochromatin compaction coincides with loss of ATAC peaks and TAD boundaries while long-range Hi-C signal increases. This process is similar to that resulting from CTCF or cohesin depletion, which results in loss of local TAD structure but does not alter compartmentalization, and in fact strengthens long-range interactions[42,43]. Thus, we speculate that loss or decrease of CTCF/cohesin activity in B compartment along with a gain in heterochromatin proteins during differentiation may provide the driving force behind compaction.

Mis-regulation of the spatial organization of the genome has been implicated in a host of diseases[44]. Approximately 1% of live births manifest congenital heart disease (CHD), which is often caused by mutations in transcription factors or chromatin modifiers[1]. Furthermore, over 500 loci are associated with risk of cardiovascular diseases (CVD) in the adult, and a large fraction of these are intergenic regions with likely regulatory functions

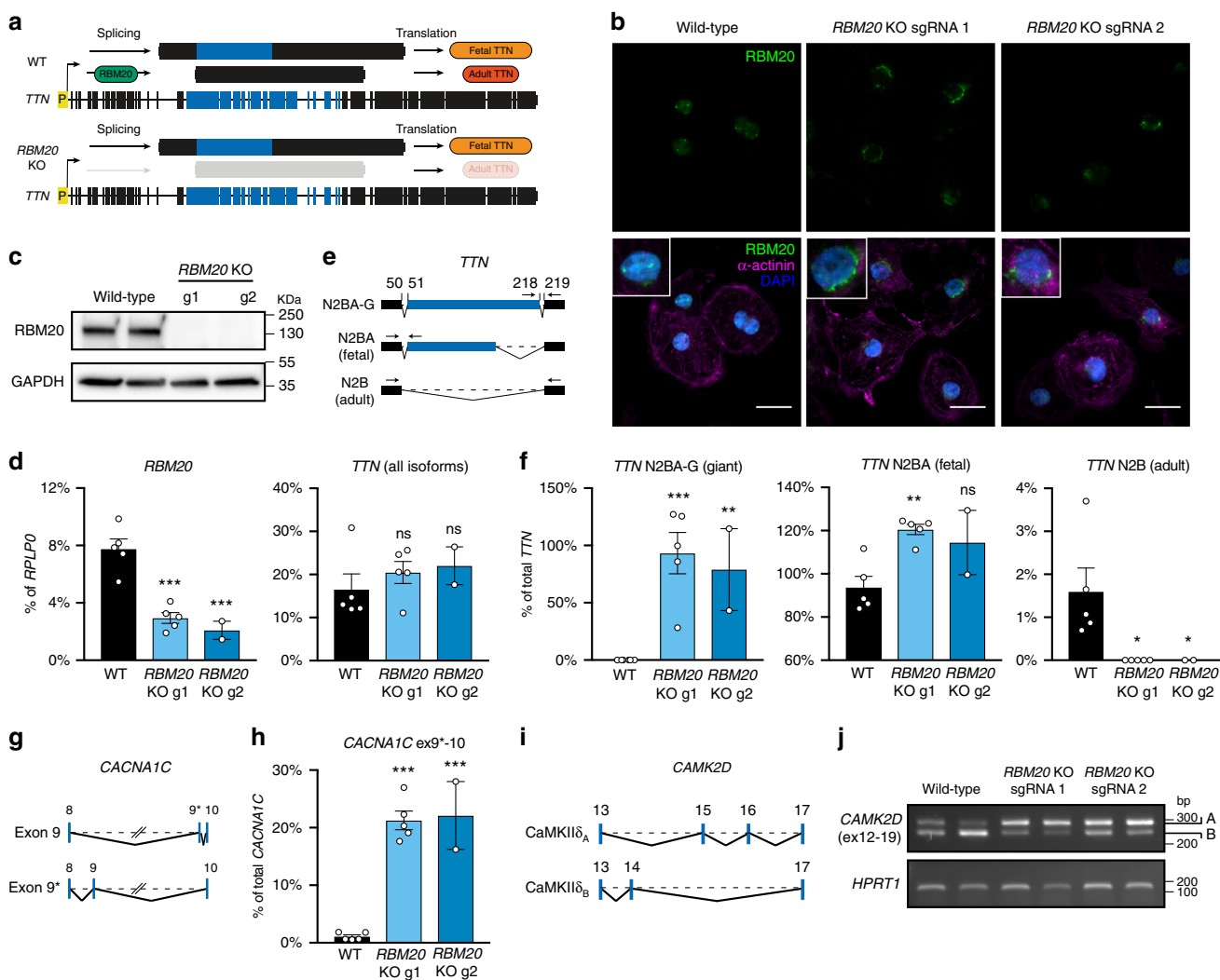

**Fig. 9** Knockout of RBM20 dysregulates alternative splicing in hESC-derived cardiomyocytes. **a** Predicted effects of the knockout of RBM20 on *TTN* biogenesis (loss of RBM20-dependent alternative exon exclusion). Refer to the legend of Fig. 8a for the abbreviations. **b** Representative immunofluorescence results in CM derived from wild-type or RBM20 knockout hESCs generated with two independent CRISPR/Cas9 single guide RNAs (sgRNAs). Nuclei are counterstained with DAPI, and insets show magnified views; scale bars: 10 μm. **c** Western blot validation of RBM20 knockout (uncropped images shown in Supplementary Fig. 11). g: sgRNA. **d** RT-qPCR in CM derived from the indicated hESC lines. Expression is relative to the housekeeping gene *RPLP0* and mean ± s.e.m. is shown. $n = 5$ (WT and *RBM20* KO g1) or 2 (*RBM20* KO g2) independent differentiations. **e** Expected major *TTN* isoforms in CM. Exons predicted to be excluded by RBM20 are in blue, and arrows indicate the location of RT-qPCR primers. **f** As in panel (**d**), but expression of the indicated *TTN* isoform is relative to the total amount of *TTN*. **g** Expected *CACNA1C* isoforms due to RBM20-dependent alternative exon selection. RBM20 is predicted to favor inclusion of exon 9 over exon 9*. **h** As in panel (**d**), but expression of the *CACNA1C* exon 9* isoform is relative to the total amount of *CACNA1C*. **i** Expected *CAMK2D* isoforms due to RBM20-dependent alternative exon selection. RBM20 is predicted to favor inclusion of exon 14 over exons 15–16. **j** Semiquantitative RT-PCR in CM for the indicated *CAMK2D* isoforms and the housekeeping gene *HPRT1*; $n = 2$ independent differentiations. All *p*-values in this figure are vs WT and calculated by one-way ANOVA followed by Holm-Sidak's multiple comparisons (ns ≥ 0.05; * < 0.05; ** < 0.01; *** < 0.001). Source data are provided as a Source Data file

involving long-range chromatin interactions. Therefore, the use of genomics resources such as those described here represents a valuable approach to clarify the molecular bases of CHD and CVD. Recent examples of this approach include using promoter capture Hi-C[45–47] or deeply sequenced Hi-C[48] to prioritize functional targets of CVD associations, and combining Hi-C from non-cardiac cells with GWAS analysis to identify gene modifiers associated with high risk of Tetralogy of Fallot[49]. Overall, by providing a detailed genome-wide chromatin organization map in differentiating cardiomyocytes together with extensive gene expression and chromatin accessibility data, our work will represent a valuable resource for the study of both cardiac development and disease.

We observed that *trans* interactions are enriched within A compartments. We suspected that these interactions were clues to gene regulatory mechanisms, which we explored in detail for the key cardiac gene *TTN*. This led us to identify a novel 3D chromatin feature specific to cardiomyocytes: the spatial coalescence of multiple genes from different chromosomes into foci marked by the splicing regulator RBM20. This mechanism relies on the newly-transcribed *TTN* pre-mRNA, which has a large number of RBM20 binding sites and thus appears to function as a scaffold to nucleate RBM20 foci. The concentration of RBM20 and multiple target loci results in more efficient RBM20-mediated regulation of alternative splicing of its other target transcripts, and it may reciprocally enhance the efficiency of *TTN* RNA splicing. RBM20

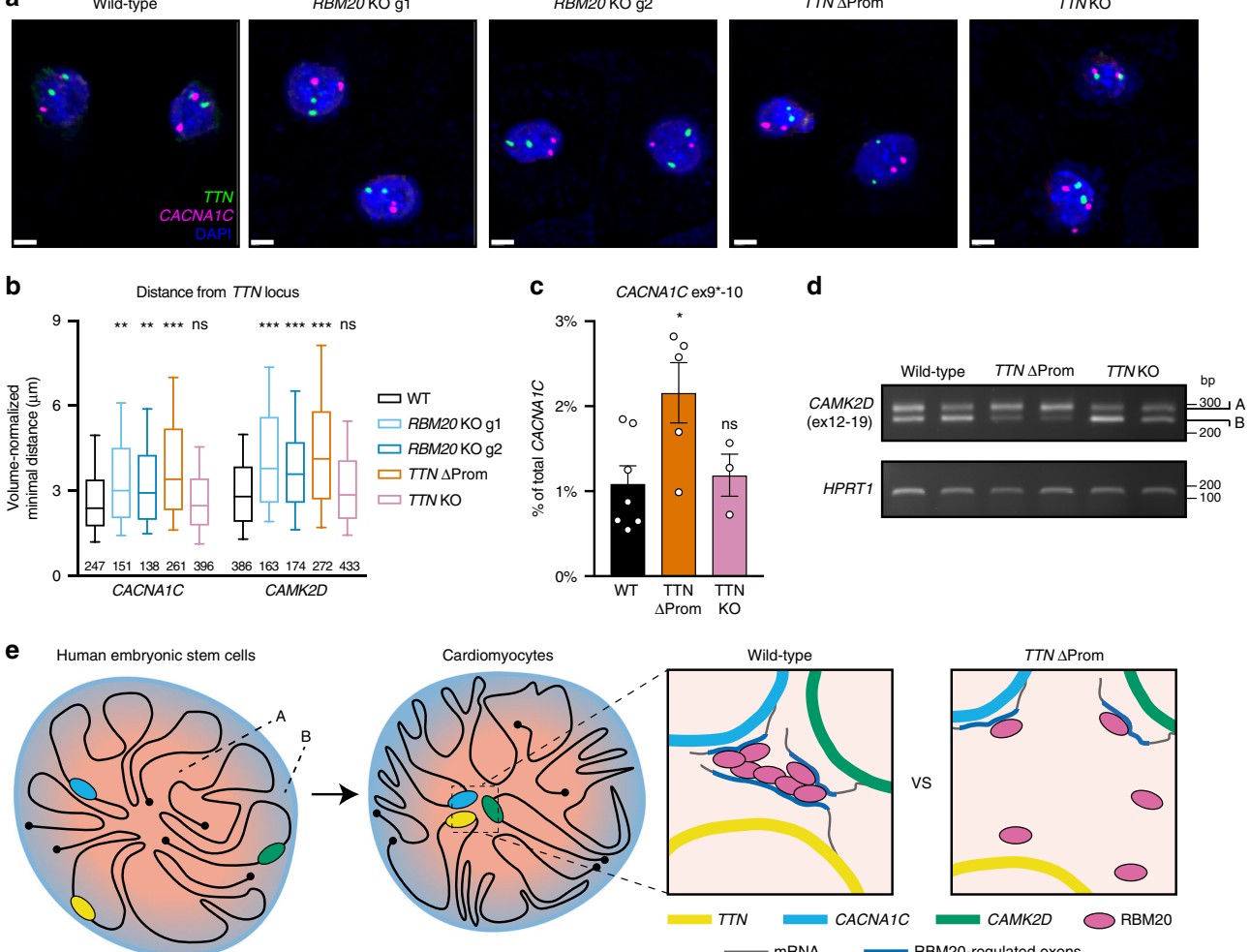

**Fig. 10** RBM20 foci promote alternative splicing of genes interacting with *TTN* in *trans*. **a** Representative 3D DNA FISH images in CM derived from the indicated hESC lines (nuclei counterstained with DAPI); scale bars: 5 μm. **b** Normalized minimum distance per diploid CM between *TTN* and the indicated loci (the number of cells is reported). Box and whiskers plots show median, 25th and 75th percentile, and the 10–90 percentile range. *p*-values by Kruskal–Wallis test followed by Dunn's multiple comparisons vs WT; ns ≥ 0.05; ** < 0.01; *** < 0.001. **c** RT-qPCR in CM derived from the indicated hESC lines. Expression of the *CACNA1C* exon 9* isoform is relative to the total amount of *CACNA1C* and mean ± s.e.m. is shown. *n* = 3, 5, or 7 independent differentiations for *TTN* KO, *TTN* ΔProm, and WT, respectively. *p*-values calculated by one-way ANOVA followed by Holm-Sidak's multiple corrections vs WT; ns ≥ 0.05; * < 0.05. **d** Semiquantiative RT-PCR in CM for the indicated *CAMK2D* isoforms and the housekeeping gene *HPRT1*; *n* = 2 independent differentiations. **e** Proposed model for the regulation of global and local chromatin organization during human cardiogenesis. Upon differentiation the heterochromatin compacts while large cardiac genes such as *TTN* transition from the inactive to the active compartment. Transcription of *TTN* nucleates foci of its splicing regulator RBM20 leading to a *trans*-interacting chromatin domain (TID) involving other RBM20 targets. This mechanism promotes alternative splicing of the resulting transcripts, and can be disrupted by preventing *TTN* transcription. Source data are provided as a Source Data file

has been shown to affect splicing in a concentration-dependent manner, and to compete with the splicing regulator PTB4, which binds to the same RNA motif and antagonizes its function[31]. Thus, foci of highly concentrated RBM20 are expected to promote its activity by increasing binding to the target transcripts. We propose that this property explains why RBM20-dependent alternative splicing is less effective when RBM20 foci are disrupted due to deletion of the *TTN* promoter. It is possible that this mechanism may be similarly dysregulated in certain forms of hereditary or acquired cardiac disease and could lead to aberrant alternative splicing of key proteins involved in excitation–contraction coupling such as CACNA1C and CAMK2D, similarly to what has been described for mutations in *RBM20* itself[50]. For instance, heterozygous mutations in *GATA4* that cause CHD lead to reduced transcription of *TTN*[22] (in agreement with the presence of a strong GATA4 binding site on the *TTN* TSS), which could weaken RBM20 foci and

thus indirectly affect splicing of other RBM20 targets. Another interesting observation is that the N2BA *TTN* isoform normally repressed by RBM20 is upregulated in ischemic heart disease and in dilated cardiomyopathy[51,52]. This suggests weakening of the RBM20-dependent splicing factory, which could influence a number of other RBM20 targets. Thus, modulation of RBM20 activity or localization could offer a therapeutic avenue to correct aberrant alternative splicing in CHD and CVD.

Generalizing our findings beyond the cardiomyocyte context, we propose that the RBM20-dependent splicing factory is an example of *trans*-interacting chromatin domains (TIDs) functionally involved in the regulation of gene expression (similarly to what achieved in *cis* by TADs). While *trans* interactions in Hi-C datasets are generally poorly explored due to our limited understanding of their significance, another well-established example of a TID-like structure has been described in mouse olfactory neurons, in which the transcription factor LHX2 and adaptor protein

LDB1 organize a multi-chromosomal super-enhancer to enable activation of a single olfactory receptor gene[53]. A wide array of nuclear speckles exists, in which factors that regulate transcription or various aspects of mRNA metabolism are highly concentrated[54]. By analogy to our findings for RBM20 foci, we hypothesize that many other TIDs exist in association to such nuclear speckles or to other yet to be determined subnuclear structures. Overall, we submit that the nature of TIDs and their properties such as stability, size, heterogeneity, regulation, and cell specificity represent promising avenues of future investigation.

## Methods

**Cell culture**. RUES2 hESCs (RUESe002-A; WiCell) and WTC-11 hiPSCs (a gift of Bruce R. Conklin; also available from Coriell #GM25256) were maintained on recombinant human Laminin-521 matrix (rhLaminin521; Biolamina) at $0.5 \, \mu g \, cm^{-2}$ with Essential 8 (E8) media (ThermoFisher) and passaged as single cells with Versene (ThermoFisher). Cells were tested to be karyotypically normal by conventional G-banding. hESC differentiation was conducted with modifications from a published method[55]. In total, $4 \times 10^5$ cells in E8 with 10 μM Y-27632 (Tocris) were seeded per well of a 12-well dish that was pre-coated with $2 \, \mu g \, cm^{-2}$ rhLaminin521 (denoted day −2). After 24 h media was changed to E8 with 1 μM CHIR-99021 (Cayman), denoted day −1. On day 0 media was changed to RPMI (ThermoFisher) with 500 μg mL$^{-1}$ BSA (Sigma, A9418) +213 μg mL$^{-1}$ ascorbic acid (Sigma, A8960), denoted RBA media, supplemented with 3 μM CHIR-99021. Samples for day 0 were collected prior to media change. Day 2 samples were collected 48 h later. Wells for further differentiation were left unchanged until day 3, when media was changed to RBA plus 1 μM Xav-939 (Tocris). On day 5, samples were either collected or changed to RBA without inhibitors. On day 7 media was changed to RPMI plus B27 (Thermo Fisher, 17504044), with further media changes every other day until day 14 samples were collected. hiPSC differentiation was conducted according to this same protocol with the following optimizations: on day −2, $1.5 \times 10^5$ cells were seeded per well of a 12-well dish; on day 0, RBA was supplemented with 6 μM CHIR-99021; on day 3, RBA was supplemented with 2 μM Wnt-C59 (Selleckchem).

**Fetal heart samples**. Fetal Heart samples were collected by the Laboratory of Developmental Biology at the University of Washington (UW), in accordance with IRB approval from the UW. Collected hearts were washed in PBS and flash frozen in liquid $N_2$ and stored at −80 °C prior to further analysis.

**In situ DNase Hi-C**. Hi-C was performed on $2–3 \times 10^6$ cells from each time point from two independent differentiations according to the published protocol[10]. Cells were fixed in the dish with fresh RPMI supplemented with 2% formaldehyde (with methanol) while in gentle orbital rotation for 10 min, and then quenched with 1% 2.5 M Glycine for 5 min at room temperature followed by 15 min at 4 °C. Cells were then treated with Trypsin (0.05%, ThermoFisher) for 5 min at 37 °C, rinsed in RPMI with 10% FBS, and scraped off the plate. Cells were washed once with PBS and flash frozen at stored at −80 °C until library prep. For fetal heart samples, frozen hearts were first homogenized with a tissue dismembrator (2000 RPM, 1 min) treated with Trypsin for 5 min at 37 °C, and then subjected to cell lysis. Whole heart samples were lysed in 500 μL buffer (10 mM Tris-HCl, pH 8.0, 10 mM NaCl, 0.2% Igepal CA-630, 1× Protease Inhibitor) and then dounce homogenized 30 times with a tight pestle. Cell culture samples were lysed directly in 500 μL lysis buffer. Samples were treated similarly from here forward. Nuclei were first resuspended in 300 μL DNase buffer with 0.2% SDS plus MnCl$_2$ and incubated at 37 °C for 60 min with periodic vortexing, and then 300 μL DNase buffer with 2% Triton X-100 and RNase A were added and incubated for another 10 min. Seven units of DNase (ThermoFisher EN0525) were added and incubated for 7 min at RT. The reaction was stopped with 30 μL of 0.5 M EDTA and 15 μL 10% SDS. Nuclei were spun down and resuspended in 150 μL water and added with 300 μL AMPure XP beads (Beckman) to irreversibly bind nuclei. In situ reactions were performed for end repair (T4 DNA Polymerase, ThermoFisher EP0062, and Klenow, ThermoFisher EP0052) and dA-tailing (Klenow exo-, ThermoFisher EP0422) prior to overnight ligation of biotin adapters at 16 °C. After incubation nuclei were washed twice with AMPure buffer (20% PEG in 2.5 M NaCl) followed by two washes with 80% ethanol to remove un-ligated adapter. Nuclei were then treated with PNK to phosphorylate the adapters and ligation reaction was performed for 4 h at RT with shaking. Nuclei were then treated with proteinase K overnight at 62 °C, and DNA was precipitated with 3 μL glycogen, 3 M Na-acetate (pH 5.2) and isopropanol for 2 h at −80 °C. DNA was resuspended in 100 μL water and purified with 100 μL of AMPure beads. Biotin pull-down was performed on the purified DNA to isolate adapter containing DNA. Myone C1 beads (ThermoFisher 65001) were mixed with DNA for 30 min at RT with rotation. Following this, samples were washed four times with B&W buffer (5 mM Tris-HCl pH 8.0, 0.5 mM EDTA, 1 M NaCl, 0.05% Tween-20). DNA was then treated on beads for end repair (ThermoFisher K0771) and dA-tailing (Klenow Exo-), four washes with B&W buffer and two washes with

TE were performed between each reaction. Sequencing Y-adapters were then ligated for 1 h at RT. Libraries were amplified between 9 and 12 cycles with Kapa HiFi ReadyStart Master Mix (KK2602) with barcode-containing primers. Libraries were then purified with 0.8 volumes of Ampure XP beads and quantified with a Qubit for sequencing.

**Hi-C processing**. Samples were paired-end sequenced on an Illumina NextSeq 500 in a high output run with 150 cycles (75 for each end). Fastq files were mapped to the hg38 genome using BWA-MEM with default parameters, mapping each end of the read pairs individually. Primary mapping sequences were extracted, and then the mapped files were processed through HiC-Pro[56], filtering for MAPQ score >30, and excluding read pairs that mapped within 1 kb, to generate valid pairs and ICE balanced matrices at 500 and 40 kb resolution[56]. Downstream analysis for female RUES2 hESCs excluded the X chromosome due to the confounding factor of X inactivation, while sex chromosomes were included for male WTC-11 hiPSCs. Heatmaps for all Hi-C data were generated through Cooler (https://github.com/mirnylab/cooler), based on the raw counts as cooler includes its own ICE balancing, and were used only for visualization purposes using default parameters and log values on the contact matrices, without diagonals. Virtual 4C data were extracted from ICE balanced matrices. HiC-Rep scores were calculated using a resolution of 500 kb with a max distance of 5 Mb and $h = 1$[13].

The valid pairs file from HiC-Pro (after filtering for distance and MAPQ score) was used as input into HOMER[57] for eigenvalue decomposition of the contact maps to calculate PC1 (A/B) scores at 500 kb resolution and with no additional windowing (super-resolution also set at 500 kb). Bins were assigned to A compartment if their average value between replicates were > 0 and B if < 0. Visualization of sample similarity by PC1 scores was performed with the t-sne function in R. Gene tracks were generated in IGV[58]. Switching A/B compartments were determined by a one-way ANOVA $p$-value < 0.05 (two replicates per time point, across the four time points of differentiation) and at least one time point having an average PC1 score > 0 and at least one < 0. A–B–A and B–A–B transitions represented 2% of the switching regions and were combined with A–B and B–A transitioning regions for downstream analysis. Saddle plots were generated by assigning each bin to its corresponding percentile value and dividing the genome into ten sets of deciles. Each interaction was normalized to the average score at the corresponding distance for *cis* interactions or to the average of all *trans* interactions and assigned to the pair of deciles based on the two bins. The plot represents the log$_2$ average value for pairs of deciles. The change between CM and hESC or hiPSC is the log$_2$ value of the difference. For the distance curves by compartment, each interaction was assigned to A–A, B–B, or A–B based on the pairs of bins and then the average interaction score for a given distance was normalized to the average interaction score for all pairs of contacts for that distance. For presentation, data are plotted on a log scale. For smoothing, the R function geom_smooth from ggplot2[59] was used with a loess smoothing function.

TADs were determined using two distinct methods: (1) Directionality Index (DI) method and domain call pipeline[17]; (2) insulation score method[18] at 40 kb resolution. Similarity in TAD boundaries (edges of assigned TADs) between samples were quantified by Jaccard Index (number of shared boundaries divided by union set of boundaries). For stringent analysis between replicates, boundaries were considered shared if there was a boundary within ± 80 kb in the other sample. Given that TAD boundaries cluster by replicate we used TAD calls on the merged samples for each condition for subsequent analyses. For comparison between time points boundaries were considered shared if there was a boundary within ± 200 kb, time point specific boundaries were identified by the lack of a shared boundary with the respective comparison. For comparison between A/B compartments and TADs we used to bedtools intersect -u -f 0.51 -a <TADs.bed> -b <compartment. bed> to assign the TADs to a specific compartment, and the majority compartment if it spans a boundary. For differential and shared boundaries between hESC and CM we used bedtools intersect -u -f 0.51 -a <TADs_boundary.bed> -b <compartment.bed>, $p$-values were calculated based on a chi-squared test between the distribution of differential or shared TAD boundaries versus the distribution of the union set of TAD boundaries.

For *trans*-interaction analysis reads from the two replicates were pooled. For analysis of a specific locus, all inter-chromosomal ICE balanced counts at 500 kb resolution were extracted for the bin of interest and then Z-scored after filtering out the top 0.5% values. Z-scores were assigned to A/B compartments based on their Hi-C PC1 score for the given time point. *P*-values between time points were determined by the Wilcoxon test in R. Network association for RBM20 was determined by totaling the ICE balanced counts at 500 kb resolution for all associated *trans* contacts within the set of 16 genes, excluding the counts between genes on the same chromosome. Bin assignment was determined by the location of the gene promoter. The background model was based on a set of 1000 random permutations, each time selecting a new set of genes with the same number of genes per chromosome as the seed set, similar to an earlier report[60], and totaling the respective ICE balanced counts. Visualization of the network was generated with Cytoscape[61]. Edge widths are proportional to the ICE balanced counts. Only edges with a count greater than the median *trans* count for CM are presented. Edge color is scaled to the log$_2$ fold change of CM versus hESC of values divided by the median *trans* count for the respective time point.

**RNA-seq**. RNA-seq libraries were prepared from total RNA using the TruSeq Stranded Total RNA Ribo-Zero H/M/R kit (Illumina RS-122-2201). RNA-seq libraries were paired-end sequenced on a Illumina NextSeq 500 in a high output run with 150 cycles (75 for each end). Reads were mapped to hg38 using STAR with default parameters[62] and then quantified and processed through the Cufflinks suite[63]. Differential expressed genes exhibited a $q$-value < 0.05 for a pairwise comparison and were expressed at least 5 FPKM in one time point. Clustering and data processing were performed with the CummeRbund suite in R. For gene tracks, values were normalized to reads per million with Bedtools[64].

Enrichment of differential genes with A/B compartments was normalized to reflect the genome-wide gene density as calculated by the number of genes in each type of compartment transition (A–B, B–A, etc.). P-values were calculated by a chi-squared test between the total number of genes in a type of compartment transition relative to the number of differentially expressed genes within that region. For example comparing the number of genes in B to A regions with the number of genes upregulated at CM. Gene ontology was conducted by DAVID[65]. Gene sizes and gene distances were calculated from the Ensembl hg38 annotation. P-values were calculated by the Wilcoxon test in R. For comparison of genes across adult tissues including adult heart from the EMBL Protein Atlas, RNA expression values were downloaded from EMBL-EBI (E-MTAB-2836). For each gene, expression was ranked by tissue from 1 (highest expression value tissue) to 37 (lowest expression value tissue). Genes were then sub-divided first by whether they were upregulated in CMs and second by whether they were in B to A regions or other genomic regions. Cumulative distribution functions were plotted in R, and p-values determined by K-S test.

**ATAC-seq**. ATAC-seq was performed according to the protocol from the Greenleaf lab[66]. Briefly, cells were harvested from cell culture and lysed in lysis buffer (10 mM Tris-Cl, pH 7.4, 10 mM NaCl, 3 mM MgCl₂, 0.1% Igepal CA-630, 1× Protease inhibitor, Sigma). In total, $5 \times 10^4$ nuclei were washed in lysis buffer and resuspended in 25 μL TD buffer, 24 μL lysis buffer, 1 μL TDE1 enzyme from the Nextera DNA Prep kit (Illumina FC-121-1030). The reaction was incubated for 30 min at 55 °C and DNA was then extracted with a MinElute PCR Purfication kit (Qiagen 28004). DNA was amplified using NEBNext High-Fidelity 2× Master Mix (M0541L) and quantified with a Qubit. All samples were pooled with Nextera primers and paired-end sequenced on a Illumina NextSeq 500 in a high-output run with 150 cycles (75 for each end).

**ATAC-seq processing**. Sequenced reads were mapped with BWA-MEM to hg38 with default parameters for paired-end reads. Mapped reads were filtered for their primary mapping location and a MAPQ score >30. Duplicate reads were removed with Picard Tools (available at http://broadinstitute.github.io/picard/). Hotspots were called with MACS2 using the parameters "--nomodel -q 0.01 --shift -100 --extsize 200". The intersection and union sets were determined by the Bedtools intersect function[64]. Quantification was performed with a custom python HTSeq script[67] that treats each read of the paired-end reads separately. Normalization and correlation was of ATAC hotspots was performed with DESeq2[68]. The overlap figure was generated with the UpSetR package[69]. Motif analysis was performed with DREME and Tomtom using a cut-off of $E$-value < 0.05[70]. Enrichment analysis with types of compartment dynamics was calculated based on the expected distribution given the distribution of the union set of all ATAC peaks. P-values were calculated based on chi-squared test between counts in a given compartment dynamic for stage-specific peaks and total ATAC peaks. Gene tracks were normalized to reads per million and produced in IGV.

**ChIP-seq processing**. Reads were downloaded from GEO for the corresponding samples: GATA4 hESC[24] (GSE61475); GATA4 and TBX5 CM[22] (GSE85628); NKX2-5 CM[23] (GSE89457). Peak files were downloaded from GEO and lifted over to hg38 using UCSC liftover. Gene tracks were generated by mapping sequence to hg38 with BWA-MEM with default parameters for single reads. Duplicate reads were removed with Picard Tools (available at http://broadinstitute.github.io/picard/). Genome coverage was calculated with bedtools genomeCoverageBed program with an extension of 200 bp. Coverage over ATAC peaks was calculated with HOMER annotatePeaks.pl using -hist for cumulative plots and -ghist for heatmaps. CTCF data were downloaded from ENCODE (https://www.encodeproject.org) for hESC CTCF (ENCSR000AMF) and CM CTCF (ENCSR713SXF). Pseudoreplicated idr thresholded peaks were used for peak calls and fold change over control was used for gene tracks.

**Gene editing**. Single guide RNAs (sgRNAs) for CRISPR/Cas9 were identified using GuideScan[71] to achieve a specificity score higher than 0.7. Two sgRNAs were designed up- and downstream of the *TTN* promoter with the goal of deleting this region upon co-delivery. Individual sgRNAs were designed toward exon 2 of *TTN* and exon 2 of *RBM20* with the aim of inducing knockout indels leading to nonsense mutations following repair of double-strand breaks by non-homologous end joining (NEHJ). The 20 bp sgRNA targeting sequences (no PAM) were: *TTN* prom up 5′-GTGAAACTCTACTTAGAGGG-3′; *TTN* prom down 5′-TTTAAGGGAA TCAACTGCTG-3′; *TTN* ex2 5′-GCAGCCGTTACAAAGCGTTG-3′; *RBM20* ex2-1 5′-AGGGTCCGGGGCCTCGTGTC-3′; *RBM20* ex2-2 5′-CGGGCAGTGTGAC

CTATGAA-3′. These sequences were cloned as double stranded oligonucleotides in pSpCas9(BB)-2A-Puro (PX459) V2.0 (a gift from Feng Zhang; Addgene plasmid #62988). All sgRNAs were verified by Sanger sequencing.

In total, $3 \times 10^5$ RUES2 hESCs were seeded per well of 6-well dish in E8 with 10 μM Y-27632 (dishes pre-coated with 2 μg cm⁻² rhLaminin521), and immediately transfected with 3 μg of sgRNA-containing plasmid using 9 μL of GeneJuice (Millipore). To delete the *TTN* promoter, 1.5 μg of each of the two sgRNA was co-transfected. After 16 h, cells were rinsed in PBS, and fed with E8 containing 10 μM Y-27632 and 0.5 μg mL⁻¹ puromycin dihydrochloride (Sigma-Aldrich) to enrich for transfected hESCs. After 24 h, media was replaced with conventional E8 and changed daily. After 72 h, hESCs were dispersed to single cells using Versene and seeded at a low density of $5 \times 10^3$ cells per 100 mm culture dish in E8 with 10 μM Y-27632. After daily media changes with E8 for 7–10 days, clonal lines were isolated by picking well-separated individual colonies and expanded for genotyping.

Genomic DNA was extracted using the QIAGEN DNeasy Blood & Tissue Kit, and used as template for PCR with LongAmp Taq (NEB). *TTN* promoter deletion was assessed using primers located ~400 bp up- and downstream to the CRISPR/Cas9 cut sites (fw 5′-ATGCCATCTATGGGGGTGGAA-3′ and rev 5′-CTTTGGG ATGAACAACTTTCTGTGT-3′). The resulting product was run on an 1% Agarose-TBE gel and detected by ethidium bromide staining, and clones with homozygous deletion of the ~1.7 kb *TTN* promoter fragment were isolated (Supplementary Fig. 10b). The deletion was further confirmed by Sanger sequencing of the PCR product. *TTN* and *RBM20* knockout was assessed by Sanger sequencing of genomic PCRs for *TTN* exon 2 (fw 5′-GGAGAAACGTGTGTC TCTGCTA-3′; rev 5′-GTGTTGGACTAATTTTCCGAAGTG-3′; 316 bp amplicon) and *RBM20* exon 2 (fw 5′-GTCAGTAACCCGAACCCTCTG-3′; rev 5′-TTGGTT CCTCGGGGTCGTA-3′; 841 bp amplicon). Clones with homozygous mutations predicted to lead to premature truncation of the resulting protein were isolated (Supplementary Fig. 10c, f). All the gene-edited hESC lines maintained a normal kayotype throughout the procedure as assessed by standard G-banding.

**3D DNA FISH**. hESC-CMs were obtained as described above. At day 14 of differentiation, CMs were rinsed in PBS, dissociated to single cells with 0.25% Trypsin in Versene for 5–7 min at 37 °C, washed twice in RPMI 20% FBS, and frozen in CryoStor CS10 (Sigma-Aldrich) at a density of $1 \times 10^7$ cells mL⁻¹. In preparation for 3D DNA FISH, CMs were thawed and seeded at a density of $2 \times 10^5$ cells cm⁻² on rhLaminin521-coated dishes in RPMI-B27 with 10 μM Y-27632 and 5% FBS. After 16 h, cells were fed with fresh RPMI-B27 and cultured for 4 days to allow for CM recovery (indicated by synchronous beating). CM were metabolically selected by 4 days of culture in DMEM without glucose and pyruvate but supplemented with 4 μM lactate. This well-established procedure efficiently removed contaminating non-CM cells (which do not tolerate lactate metabolic selection), thus highly enriching for CM (> 95% as assessed by Flow Cytometry for cTnT and immunofluorescence for α-actinin). CM were then cultured in RPMI-B27 for 4 more days before being collected for 3D DNA FISH.

3D DNA FISH was performed with modifications from a published protocol[72]. Unless otherwise indicated, all steps were performed at room temperature. Poly-Lysine slides (Sigma-Aldrich, P0425) were pre-coated with 0.5 or 2 μg mL⁻¹ rhLaminin521 (for hESCs or CMs, respectively) dispersed on a ~2 cm² spot outlined by a hydrophobic pen for 2 h at 37 °C. Cultures were rinsed in PBS, dissociated to single cells with TrypLE at 37 °C, washed in culture media, and plated in culture media supplemented with 10 μM Y-27632 at a density of 40,000 cells per spot onto the pre-coated slides. Cells were allowed to attach to for 4 h at 37 °C. For transcription inhibition experiments, 5 μM Actinomycin D was added to the culture media both 3 h before collecting the cells and during the seeding procedure. Cells were fixed in 4% PFA and then quenched in 0.1 M Tris-HCl pH 7.4 for 10 min each. Cells were subsequently permeablized in 0.1% Saponin/0.1% Triton X-100 in PBS for 10 min, followed by two washes in PBS. Cells were equilibrated in 20% Glycerol/PBS for 20 min, and stored overnight in 50% Glycerol/PBS at −20 °C.

FISH probes were prepared using the Vysis Spectrum Nick Translation Kit (Vysis, 32-801300) from BACs obtained from the BACPAC Resources Center: CH17-275G10 (*TTN*); CH17-236B15 (*THSD7B*); CH17-82H11 (*CACNA1C*); CH17-266P23 (*CAMK2D*); CH17-473L14 (*CAMK2G*); CH17-468A13 (*FNBP1*); CH17-200I17 (*KCNIP2*); CH17-147J13 (*LMO7*). BACs were purified using the QIAGEN Plasmid Midi Kit, and validated by PCR with two primer sets for the expected genomic region. Two micrograms of BAC DNA was combined with either SpectrumGreen (Vysis, 30-803000) or SpectrumOrange (Vysis, 30-803200) and labeled according to manufacturer's protocol for 4 h at 15 °C followed by inactivation at 70 °C for 10 min. Probes were first purified with the QIAGEN PCR purification kit and then precipitated with 40 μL Salmon Sperm DNA (Thermo Fisher 15632-011), 60 μL Human Cot-1 DNA (Thermo Fisher 15279-011), 15 μL sodium acetate (3 M, pH 5.5), and 390 μL 100% Ethanol for 2 h at −20 °C. Probes were spun down at $16,000 \times g$ for 30 min at 4 °C and resuspended in 50 μL Hybridization Buffer (50% formamide, 2× SSC, 10% dextran sulfate). Probes were stored at −20 °C.

For hybridization, slides were removed from −20 °C and equilibrated in 20% Glycerol/PBS for 20 min. Slides were submerged in liquid nitrogen and then allowed to thaw; this procedure was repeated for three cycles. Slides were washed

twice in PBS and incubated in 0.1 M HCl for 30 min. Slides were washed once in PBS then permeabilized in 0.5% Saponin/0.5% Triton X-100/PBS for 30 min followed by two more washes in PBS. Cells were then equilibrated in 50% formamide/2× SSC for 10 min. Five microliters of green- and orange-labeled FISH probes were combined and put onto a coverslip which was inverted onto the slide. Slides were sealed with rubber cement and air dried in the dark for 30 min. Slides were incubated for 10 min at 78 °C to denature the DNA, then overnight at 37 °C for hybridization. After incubation the rubber cement was removed and slides were washed in (1) 50% formamide/2× SSC 15 min at 45 °C; (2) 0.2× SSC 15 min at 63 °C; (3) 2× SSC 5 min at 45 °C; (4) 2× SSC 5 min at RT; (5) PBS 5 min at RT. Slides were then counterstained with DAPI (5 µg mL$^{-1}$ in 2× SSC, ThermoFisher D3571) for 2 min and washed twice with PBS. Coverslips were mounted with 10 µL Slow Fade Glow (Thermo Fisher, S36936) and sealed with nail polish.

We captured images with a Yokogawa W1 spinning disk confocal mounted on an inverted Nikon Eclipse TiE base and a CFI PlanApo VC 60× Water Immersion lens (NA 1.2). Sample excitation was via full field exposure from 100 mW 405-, 490-, and 561-nm lasers. Sample emission was captured sequentially through Chroma Technology filter sets ET455/50, ET525/35, and ET605/52 by an Andor iXon EM + EMCCD camera. Z-stacks were acquired at a step size of 0.3 µm to cover a z-distance of 15 µm (51 stacks). Twenty randomly selected field of views (FOV) were imaged at a separation of two FOV from each other. Automated image analysis was performed with Imaris software (version 9.2.0; Bitplane). Background was removed with a Gaussian filter (9 µm for DAPI and 1 µm for FISH signals, based on the average size of each feature). Nuclei boundaries were defined with the surface feature with a surface detail of 1 µm (all other parameters as default; auto thresholding based on surface quality). FISH images were first masked based on the nuclear boundaries to retain only the nuclear signal, then loci were identified with the spot calling feature. One-micrometer-wide spots were determined based on the average size of FISH signal (all other parameters were set as default; auto thresholding based on spot quality). The distances between the various features was determined by applying the distance transformation function. The statistics for surfaces and spots were processed using a custom Python script which extracted and analyzed information only for diploid nuclei (containing exactly two green and two orange spots. At least 100 cells per conditions were analyzed (corresponding to at least 10 FOVs), and the exact number is indicated in each figure. Distances between loci and between a locus and the nuclear periphery were calculated relative to the center of 1-µm-wide FISH spots, and were normalized by the average hESC nuclear volume of based on the cube root of the cell volume for each cell ($X_{norm} = X_{raw} \times$ average volume$^{1/3} \times$ volume$^{-1/3}$). Proximity between two loci in a given cell was defined as the minimal distance between one pair being less than twice the diameter of a FISH spot (2 µm). In the representative pictures reported in the Figures, the orange FISH signal is depicted in magenta to the benefit of color blind readers.

**ImmunoFISH.** Immunofluorescence combined with DNA FISH was performed according to a published method[73]. CMs were plated on slides prepared for FISH analyses as described above. Upon removal from storage at −20 °C in 50% Glycerol/PBS, slides were washed two times in PBS for 5 min, then permeabilized and blocked in PBS with 0.1% Triton X-100 and 5% normal goat serum for 30 min. Primary antibodies (anti-RBM20 rabbit polyclonal at 1:200 dilution, ThermoFisher #PA5-57404; anti-sarcomerinc α-actinin antibodies monoclonal [EA-53] at 1:300 dilution, Abcam #ab9465) were diluted in antibody buffer (PBS with 0.1% Triton X-100 and 1% normal goat serum) and incubated overnight at 4 °C in a humid chamber. Following three washes with PBS for 5 min, secondary antibodies (goat anti-rabbit Alexa Fluor 647 and goat anti-mouse Alexa Fluor 594, both used at 1 µg mL$^{-1}$, ThermoFisher) were diluted in antibody buffer and incubated for 1 h at room temperature in the dark. Following three washes with PBS for 5 min, slides were fixed with 3% PFA in PBS for 10 min. Slides were washed two times in 2× SSC, denatured in 50% formamide 2× SSC pH 7.2 for 30 min at 80 °C, and washed three times with ice cold 2× SSC. Ten microliters of the *TTN* green-labeled FISH probe was pipetted on a coverslip that was then inverted onto the slide. Slides were sealed with rubber cement, air dried in the dark for 30 min, and hybridized overnight at 42 °C. After removing the rubber cement, slides were washed three times in 50% formamide and 50% 2× SSC, pH 7.2, at 42 °C. Slides were then counterstained with DAPI (5 µg mL$^{-1}$ in 2× SSC) for 2 min and washed twice with PBS. Coverslips were mounted with 10 µL Slow Fade Glow and sealed with nail polish.

ImmunoFISH were imaged as described for 3D DNA FISH, and subjected to the same automated image analysis using Imaris to identify nuclei boundaries and *TTN* loci. The RBM20 immunostaining signal was processed analogously to FISH signal to identify nuclear foci with the spot calling feature. One-micrometer-wide spots were determined based on the average size of RBM20 foci (all other parameters were set as default; autothresholding was based on spot quality). The distance transformation function was applied to compute the distance between the center of 1-µm-wide *TTN* loci and RBM20 foci in diploid cells. An overlap between these two features in a given diploid cell was defined as their distance being less than the diameter of a spot (1 µm).

**Immunofluorescence.** CMs were processed as just described for immunoFISH, but after secondary antibody incubation slides were counterstained with DAPI

(5 µg mL$^{-1}$ in PBS), washed three times in PBS for 5 min, and mounted. The anti-RBM20 and anti-sarcomerinc α-actinin antibodies were diluted as described for immunoFISH, while the anti-titin rabbit polyclonal antibody (Myomedix; #Z1Z2) was used at a concentration of 0.3 µg mL$^{-1}$.

**Flow cytometry.** CM were collected at day 14 of differentiation for flow cytometry analysis. Briefly, cells were washed with PBS and treated with TrypLE (ThermoFisher) for 5–10 min at 37 °C and collected in RPMI. Cells were spun down and resuspended in 200 µL 4% paraformaldehyde for 10 min at RT. Cells were spun down and resuspended in 500 µL PBS + 5% FBS and split into two tubes for staining. Tubes spun 2,000 rpm × 5 min and were resuspended in 50 µL PBS + 5% FBS + 0.75% saponin with either 0.5 µL mouse IgG1 (eBioscience 14-4714) or 0.5 µL mouse cTnT (ThermoFisher MS-295) and incubated for 30 min at RT. After incubation 150 µL PBS + 5% FBS + 0.75% saponin were added and samples were spun down and washed with 200 µL PBS + 5% FBS + 0.75% saponin. Both sets were resuspended after washing in 50 µL PBS + 5% FBS + 0.75% saponin with 0.25 µL goat anti-mouse PE secondary (Jackson 115-116-072) and incubated for 30 min RT in the dark. Following secondary 150 µL PBS + 5% FBS were added and samples were spun down and washed with 200 µL PBS + 5% FBS. Samples were then stored in 400 µL PBS + 5% FBS + 100 µL 4% paraformaldehyde at 4 °C until analysis. Samples were run on a BD FACSCanto II and data were acquired with the BD FACSDIVA software (both from BD Biosciences). Analysis was performed on FloJo version 10 software. Only samples >75% cTnT positive were used for downstream applications.

**RT-qPCR and semiquantitative RT-PCR.** Cell samples were collected in RLT buffer and homogenized with a Qiashredder (QIAGEN) before RNA purification with the RNeasy Mini Kit (QIAGEN) including on-column DNase digestion. RNA from whole hearts was extracted with RNeasy Fibrous Tissue Kit (QIAGEN). cDNA was generated from RNA with M-MLV RT (ThermoFisher) and random hexamer priming. RT-qPCR was performed in technical triplicate using the SYBR Select MasterMix (ThermoFisher) and run on a 7900HT Fast Real-Time PCR system (Applied Biosystems), all according to the manufacturer's instructions. Gene expression was calculated using the ΔCt method as $2^{(Ct\ gene\ -\ Ct\ housekeeping)}$, and the housekeeping gene is indicated in the Figure legends. For semiquantitative RT-PCR, 10 ng of cDNA was subjected to 25 cycles of amplification using GoTaq Flexi DNA Polymerase (Promega) in a 25 µL reaction. Ten microliters of the PCR product were run on a 2% Agarose-TBE gel and detected by ethidium bromide staining. Oligonucleotide primer sequences for RT-qPCR and semiquantiative PCR are listed in Supplementary Table 9.

**Western blot.** Protein lysates were obtained with RIPA buffer containing protease inhibitors. Following clarification of the lysate by centrifugation and assessment of protein centrifugation by BCA assay, samples were diluted in Laemmli buffer and boiled. Twenty micrograms of protein were electrophoretically separated using 4–20% Mini-PROTEAN TGX Precast Protein Gels (Bio-Rad). Proteins were transferred to PVDF membranes and blocked in PBS with 0.1% Tween-20 (PBST) and 5% bovine serum albumin (BSA). The anti-RBM20 rabbit polyclonal primary antibody (ThermoFisher #PA5-57404) was diluted at 1:500 in PBST 1% BSA and incubated overnight at 4 °C. Membranes were washed three times in PBST for 10 min at room temperature, incubated for 1 h at room temperature with goat anti-rabbit HRP-conjugated secondary antibody, and washed three times in PBST for 10 min. Chemiluminescent reaction was initiated by incubation with SuperSignal West Pico Chemiluminescent Substrate (ThermoFisher), and images were acquired using a ChemiDoc Imaging System (Bio-Rad) in "high resolution" mode. Before re-probing for the housekeeping protein GAPDH (mouse monoclonal [6C5] diluted at 1:5000; Abcam #8245) according to the same protocol but using goat anti-mouse HRP-conjugated secondary antibody, membranes were treated with Restore Plus western blot stripping buffer (ThermoFisher), washed three times, and re-blocked.

**Statistical analyses.** Standard statistical analyses were performed using R or Prism 6 (GraphPad). The type and number of replicates, the statistical test used, and the test results are described in the figure legends. All statistical tests employed were two-tailed. No experimental samples were excluded from the statistical analyses. Sample size was not pre-determined through power calculations, and no randomization or investigator blinding approaches were implemented during the experiments and data analyses.

**Reporting Summary.** Further information on experimental design is available in the Nature Research Reporting Summary linked to this article.

## Data availability
All data for Hi-C, RNA-seq and ATAC-seq is available on Gene Expression Omnibus accession number GSE106690. Source data for gel images is provided in Supplementary Fig. 11. The data for graphical representations in the Figures and Supplementary Figures is provided in the Source Data File.

## Code availability

Custom code is available on github at https://github.com/pfields8/Fields_et_al_2018/.

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

## Acknowledgements

Thank you to members of the Murry lab for helpful discussions during the development of this project and especially to Xiulan Yang for assistance with obtaining samples and Katie Mitzelfelt and Silvia Marchianò for experimental support. We thank members of the Shendure and Noble labs and in particular Ruolan Qiu and Kate Cook for assistance with Hi-C technology and analysis. Flow cytometry was done with assistance from UW Cell Analysis Facility in the Department of Immunology. We would like to acknowledge the Mike and Lynn Garvey Cell Imaging Lab at UW and its director Dale Hailey for assistance with sample imaging and analysis. We thank members of the Birth Defects Research Laboratory for assistance in obtaining human heart tissue. A.B. is funded by an EMBO Long-Term Fellowship, ALTF 448-2017. P.A.F. is funded through Experimental Pathology of Cardiovascular Disease training grant, NIH T32 HL007312. This work is part of the NIH 4D Nucleome consortium (NIH U54 DK107979, to C.E.M., W.S.N. and J.S.), with additional support from P01 GM081619, R01 HL128362, and the Foundation Leducq Transatlantic Network of Excellence (to C.E.M.). The Birth Defects Research Laboratory is supported by NIH grant R24 HD000836.

## Author contributions

A.B. designed, performed and analyzed FISH, pharmacology, and gene editing experiments, performed Hi-C on hiPSCs, and wrote the manuscript. P.A.F. designed and performed Hi-C, ATAC-seq, and RNA-seq experiments on hESCs, performed most bioinformatics analyses, and wrote the manuscript. V.R. contributed to experimental design and execution and assisted in writing the manuscript. G.B. and G.G.Y. contributed to high-throughput sequence analysis. H.R. and L.P. contributed to experimental design and supervised the experiments. W.S.N. and J.S. contributed to study design and supervision. C.E.M. conceived and supervised the study, obtained research funding and contributed to writing the manuscript.

## Additional information

**Competing interests:** The authors declare no competing interests.

