## [Peer Review File · Nature Communications]

Reviewers' Comments:

Reviewer #1:

Remarks to the Author:

In this study, Fields et al characterize the dynamic genomic landscape of a differentiating hESC to cardiomyocyte and of human fetal hearts. They find, as expected, that while a large fraction of the genome architecture remains invariant during the process of differentiation, a small, yet important fraction does undergo dynamic alterations that are coincident with changes in chromatin accessibility and gene expression. This study confirms some predicted patterns of cell-type specific gene regulation, while also providing an important and useful resource from which new mechanisms of transcriptional regulation may be uncovered. However, some major revisions are required for this to be suitable for publication in this journal.

Major concerns:

- a. Currently, these data do not robustly support the simple notion that lost or gained TAD boundaries correlate with gene expression changes during differentiation. The data, in its current form, suggest that are several classifications of genes within each group – some that do follow expected correlations of compartments, chromatin accessibility and gene expression changes, as well as some that do not. These trends should be further categorized with relevant examples.
 - The observation that TAD boundaries gained in CMs are generally depleted in B to A transitioning compartments conflicts with the gene expression data in Figure 2. While CAMK2D does better exemplify the correlation between a gained boundary and transcription (Fig 3f), there does seem to be residual TAD structure even in the CM stage.
 - The correlations between open chromatin regions and transcriptional activity are also unclear. For eg: B to A switching compartments are enriched for ESC ATAC peaks, although they are depleted for any differentially expressed genes at this stage
- b. The relevance of the analysis about TAD size and numbers within compartments (Fig 3) is currently unclear.
- c. The analysis of open chromatin regions and the underlying motifs may be further strengthened by integrating ChIP-Seq data for the transcription factors identified, such as those published by Ang et al (2016).
- d. In Fig 5, does the analysis pertaining to gene size at the CM stage hold true even with the exclusion of the TTN gene? There are several predicted CTCF sites within the TTN gene, but only two show progressive loss of ATAC peaks through differentiation – does this correlate with CTCF binding strength, orientation or other attributes?
- e. The postulated mechanism of an RBM20-mediated trans-interaction is not supported by these data and will require far more detailed experiments to validate. Instead, more DNA FISH experiments to validate other interactions identified by this study will strengthen the current narrative.

Minor concerns:

- a. The example of transient A compartment dynamics is not easily discernable from Fig 2. Either a better example or a clear detailing of the region undergoing the compartment switching is necessary to make this point. Additionally, an example of compartment dynamics of genes expressed in CP that undergo B to A compartment switching would also be interesting to add to this figure.
- b. The text should be edited for minor typos. Eg.s; line 37 – "...during human development.", line 43 – "...emerged as a key mechanism..", line 57: "...feeds back on gene regulation".

Reviewer #2:

Remarks to the Author:

Fields et al have performed in situ DNase Hi-C on samples from 4 stages of a cardiomyocyte differentiation protocol (following the same approach in their previous publication in Cell 2012, ref 7 where they interrogated histone profiles and RNA-seq, but not Hi-C then), and this time, along with two fetal human heart samples. Here, they have described changes in higher order chromatin architecture, and propose the presence of a cell type-specific "splicing factory". Although they are the first to perform these analyses in human cardiomyocytes, nearly all of their findings are not surprising (e.g. ATAC-seq peaks enriched in A compartment, cardiac genes changing from B to A compartment and so on). While it is interesting to uncover another angle which is to group genes together in a splicing factory, and also to propose that the splicing units are pulled together at their templates by a splicing factor such as RBM20, the authors fall short of having strong evidence for the phenomenon.

Criticism (major and minor):

1. QC metrics are provided in supplement table 1. However, adequate details of the QC assessment are lacking to conclude if these hi-C libraries are robust and of good quality for downstream interpretation. Some of the few important QC metrics are % duplication of read pairs, fragment length distribution, fraction of hi-C read pairs within the same restriction fragments. It is also noted that the sequencing depth for each replicate does not seem sufficient to provide a high resolution interactome.
2. The ATAC-seq QC metrics provided are also inadequate and lacking of key information such as total number of peaks detected, FRiP, insert length, etc. It is also strange (and concerning) that the sequence data contains high percentage of mitochondrial RNA.
3. The authors perform A/B compartment analysis on the samples, and found that the proportion of A and B compartments are roughly equal. The authors mentioned "PCA of A/B scores separates the samples by differentiation time point (PC1) and cardiomyocyte purity (PC2)". It is not conventional to compare PC scores between samples as the PC scores are generated individually for individual replicates, and it is hard to put meaning to the magnitude of difference between samples .
4. Line 43. "Spatial organisation of the genome in 3-D has emerged as key mechanism by which cells can control regions of chromatin accessibility" sounds a fundamentally important biological idea. what is the reference to support this?
5. Line 47. "... implicated in a range of diseases, including cancer and limb malformations" references 3-6 do not include the reference on Limb Malformations.
6. The motivation statement in Line 48. "... how these transitions occur temporally across stages of human lineage remains unaddressed, or how structural arrangement differs between pluripotent progenitors and terminally differentiated cells" is debatable accuracy. Many of the references in this manuscript precisely address these topics. This is exactly what lends to the lack of novelty for this manuscript.
7. In Figure 1f, compartments were defined as "dynamic". The statistics used here is unusual. Can the authors justify why is ANOVA used here. It is unclear how was the comparison done to obtain the p-value. Was it done across the time point? Are the PC scores normally distributed, and independent? If not, this test is deemed inappropriate.
8. Figure 1i. What is the meaning of "cis-interaction score"? How was it generated quantitatively? Similar to figure 1j, what is trans interaction score?
9. Similar questions for statistic significance arise in other figures: e.g. Fig3e, Fig6a. There do not appear to be differences in the medians between groups, and yet the difference is deemed statistically significant?
10. There are some bits of novelty within: e.g. Line 97. "Examining the dynamic regions, many of the strongest gains in long-range interactions are associated with the B compartment. ... stronger intra-chromosomal contacts occur between homotypic regions" .. but what does this mean functionally? There is only one example given in Suppl Fig 3A. what is the global picture?
11. Line 111. states the authors set out to "investigate for genes that may be drivers of compartment changes..." but they end up not doing so. At least not with their RNA-seq. Instead it was their motif

analysis of ATAC-seq related sites that throw up GATA and NKX which may be implicated in this role?

12. Line 115. "In contrast to Hi-C, most of the dynamic changes in gene expression are cell-type specific." The authors should elaborate further on this. What are the gene signatures that are cell type specific? Are these differentially expressed genes?

13. Line 174. Only one example of the motif is given at the NEBL locus. what are others? It is the important message that GATA and NKX could be the key drivers of chromatin organisational change?

14. Fig 2f. the PC1 levels to suggest the transition B -> A -> B are small vis a vis neighbouring PC. Besides the regions that go from A-B-A and B-A-B appear to be regions that are already in the borderline zone of compartments. This makes it concerning that the change could be noise as generated by the algorithm calling the compartments. How do the authors exclude the possibility that these small changes are not noise?

15. Figure 3a. The dendrogram is not visible to make any meaningful conclusion. Also, by given a 2-window width (80kb) allowance for error seems less stringent. Even at such low stringency, the agreement between two replicates is only about 60-70%.

16. Fig 3f. TAD changes as defined by the algorithm dont match to the heat map visually. It simply looks like the original TAD marks between ANK2 and CAMK2D are blurred? The authors called A/B compartment at 500kb resolution while they call TADs at 40kb resolution. Yet, they claim that TAD organization is dynamic across differentiation but independent of AB compartment changes. The differences observed could be an artefact of different resolution used. It is also not justified on the criteria used to assess if a TAD is overlapping with a AB compartment. It is also unsure how many boundaries are called by the authors, It is very hard to assess the quality of the TAD calling as well unless the authors are diligent in providing clearer heatmaps showing TADs. The authors are lacking evidence of TADs transition during the differentiation process.

17. As commented, the presence of a splicing factory is not convincing. Chromatin organization, particularly CTCF mediated chromatin loops has been shown to regulate splicing, however the link between spatial proximity of genes and co-splicing is relatively new. The concept of "splicing factories" in addition to the more conventional "transcription factories" is important. The genes may be grouped together because they are co-transcribed, but not necessarily because they are co-spliced. Since RBM20 is an RNA binding protein, would there have to be communication with another DNA binding protein that brings these gene loci together. More evidence is needed to make this bold claim.

Finally, given that the changes in higher order chromatin architecture as described here are not surprising, the authors should have to give more insight into the local Enhancer-Promoter interaction changes during development. For example, In Page 9, line 193, they imply that the isolation of important developmental genes allows for individual regulation of these genes, they arrive at this conclusion by analysing separation of these cardiac genes from the next gene promoter. While these genes may not be situated very close to other genes, they may be regulated by distal enhancers which a deeper interactome would be able to map out. Hence it will be interesting to look at the Enhancer-Promoter interaction dynamics, beyond just the structural changes which have turned out to be as expected.

Reviewer #3:

Remarks to the Author:

I thought this was an interesting paper and would be happy to recommend publication if the authors could improve some of the computational analysis as detailed below:

1) At around line 97 the authors describe their examination of the dynamic regions, but it was not clear how they corrected for the fact that interactions are more likely to be with adjacent regions in the same compartment. They say – "In the pluripotent state, the median cis interaction signal

between A compartments...". How was the median cis interaction signal derived? For (e.g. A-A) is it simply the median of normalized contacts for all possible pairs of A compartment bins? Or does the calculation involve further normalization e.g. comparing observed to that expected? If it is the former, I think that some comparison against the background contact level in relation to sequence separation is needed.

2) The authors suggest in their analysis that the average proportion of the genome belonging to either the A or B compartments does not change much during differentiation. However, it is not clear whether the A or B compartments become more/less fragmented (i.e. what are the average sizes of pieces of A and B compartment)? If there is a big change, the shift in the distribution of interaction signals could be partially attributed to fragmentation. To resolve this, the authors should calculate the observed/expected ratio instead of using the contact score itself. The expected value could be straightforwardly derived from an empirical averaging of contacts for different sequence separations (e.g. at 40 kb, 80 kb, 120 kb etc.) within each chromosome. Alternatively, and perhaps more informatively, instead of using a simple median value they could group the A-A or A-B contacts into different groups according to sequence separation and compare each group specifically. That would give a better idea whether the observed change in homotypic/heterotypic compartment score is mostly due to short or long range contacts, or both.

3) Moving to the analysis of the dynamics of TAD organisation during differentiation (lines 140-154), there are a lot of statements made where the results are only weakly significant (see e.g. Fig 3e) or unsubstantiated. For example, is it really true that the changes in TAD boundaries are independent of A/B dynamics (lines 150-151)?

4) In Figure 5d, there are two TSS within the TTN locus – I think one must be missing from this Figure? In the references there are papers describing the internal TSS in more detail, but it would be helpful for the reader's if the authors could show them in the Figure and discuss the two TSS briefly and what is special about them.

5) In Figure 5d, the authors do not say what sort of cells were used to determine the CTCF binding data? The citation is for the ENCODE project paper, but the dataset used is not specified. Is it relevant for the cells they are looking at, or do the authors need to carry out their own experiments?

6) Furthermore, also in Figure 5d, the observation that ATAC signal diminished around CTCF sites during lineage specification is interesting. However, I didn't understand how the authors concluded this might be due to a physically proximal chromatin hub in hESC's? It would be interesting to discuss this.

7) In Figures 6c-d, according to a background model of random gene distribution along each chromosome, the association of 16 co-regulated genes is significantly different from the background in CM but not in ESC. However, the randomization of genes does not take into account their A/B compartment identity. It could simply be that those genes get activated and move from B to A in CM, which leads to their observed higher than expected interaction by chance if they are randomly distributed in either A/B. It would therefore be useful to compare this with random sets of genes with the same A/B identity as the 16 genes of interest.

8) Regarding the definition of A/B compartments, did the authors assign a genome region to be within A if $PC1 > 0$ and B if $PC1 < 0$? Or were there further probabilistic measures involved?

9) In Figure 6f, in the example cell for the FISH experiments it looks as if there is one allele that shows co-localisation between the red and green dots, but not the other. It would be interesting to

know whether this occurs across the majority of cells or it is just a co-incidence?

Textual matters: Line 702: Figure legend: The upper value is the number of TADs, and the lower value is the median size.

Not sure what the upper/lower values refer to in Figure 3c?

Lines 219 to 224: Would this be better in the Discussion?

From line 239: "In contrast, heterochromatic, silent regions in hESCs are relatively accessible compared to differentiated cells but compact during differentiation coincident with increased long range Hi-C signals and a loss of ATAC-peaks. This is similar to results seen in CTCF and cohesin depletion studies where loss of local TAD structure does not alter compartmentalization and in fact strengthens long-range interactions." I understand that the two observations mentioned above were made independently, but find it hard to follow the logic of why those two observations are similar.

Point by point answer to the Reviewers

Dynamics of genome reorganization during human cardiogenesis reveal an RBM20-dependent splicing factory

Bertero, Fields et al.

We thank the Reviewers for their helpful and encouraging comments. In our revised manuscript we have addressed all the concerns raised by the Reviewers by performing several additional analyses and experiments, and by modifying the text based on their suggestions. Toward that end, we have strengthened our bioinformatics analyses both methodologically and by including additional datasets, thus reinforcing and expanding our initial conclusions. Additionally, we have greatly expanded the mechanistic side of the manuscript by providing robust functional validation of RBM20-mediated *trans* chromatin interactions using both pharmacological and genetic perturbations combined with extensive 3D DNA FISH analyses. Collectively, these experiments support our original hypothesis that RBM20 is involved in a cardiac specific splicing factory, and they demonstrate that this factory is important for efficient alternative splicing of key cardiac transcripts. Finally, we have generated DNase Hi-C data from a second hPSC line, which confirm our general findings regarding chromatin organization changes during human cardiogenesis and provide another useful dataset for the community. You will find below our detailed point by point answer to all of the Reviewers' comments, with detailed references to the new data and text changes. Furthermore, all text changes are indicated in magenta in the revised manuscript and supplementary information.

Reviewer 1

Reviewer 1 general considerations:

“In this study, Fields et al characterize the dynamic genomic landscape of a differentiating hESC to cardiomyocyte and of human fetal hearts. They find, as expected, that while a large fraction of the genome architecture remains invariant during the process of differentiation, a small, yet important fraction does undergo dynamic alterations that are coincident with changes in chromatin accessibility and gene expression. This study confirms some predicted patterns of cell-type specific gene regulation, while also providing an important and useful resource from which new mechanisms of transcriptional regulation may be uncovered. However, some major revisions are required for this to be suitable for publication in this journal.”

We are thankful to the Reviewer for his/her constructive feedback. In our revised manuscript we have performed major revisions based on the Reviewer's suggestions to strengthen and expand our analyses on TADs and chromatin accessibility, and integrated these with ChIP-seq data for cardiac transcription factors and CTCF. Further, we have added substantial new data which not only expand the resource provided by our study, but also showcase how this could be mined to uncover a new mechanism involved in stage-specific gene expression control.

Reviewer 1 major comments:

Comment 1: “Currently, these data do not robustly support the simple notion that lost or gained TAD boundaries correlate with gene expression changes during differentiation. The data, in its current form, suggest that are several classifications of genes within each group – some that do follow expected correlations of compartments, chromatin accessibility and gene expression changes, as well as some that do not. These trends should be further categorized with relevant examples. “

We agree that our data shows that chromatin organization dynamics at the local or global level is not always predictive of gene expression changes and *vice-versa*. This is not unexpected, as chromatin organization is but one of many layers of epigenetic regulation, and the field as a whole is striving to resolve this complex puzzle. While it would be far beyond the scope of the current study to reconcile all of our observations within a unified model of gene regulation for different classes of genes, our results do make substantial progress in this direction and provide some significant examples. Most notably, our analyses regarding the relationship between A/B compartment dynamics and gene expression reveal that large cardiac genes are often regulated by a transition from the inactive to the active compartment during differentiation. In contrast, smaller developmental genes seem to rely on locally-acting regulatory mechanisms.

Aside from A/B compartments, we agree with the Reviewer that it was important to strengthen our analyses of TAD dynamics. A major limitation for this is that in contrast to A/B compartment calls, in which use of the PC1 from the correlation map is widely considered as gold standard within in the field, there is currently no single accepted standard for the TAD calling. As such, we have repeated all our analyses with a second TAD caller such that we now have results with both the DI method (Dixon et al., 2015) and the insulation score method (Crane et al., 2015), which are presented in the **updated Figure 3** and **updated Supplementary Figure 6**. While both TAD callers identified TAD boundaries that are distinct between hESCs and CMs, the insulation score method has a higher concordance between samples (68% shared between hESCs and CMs by DI method; 84% shared by Insulation score metric). Both methods confirm our initial observation that in general novel TAD boundaries appearing in cardiomyocytes are associated with upregulation of the nearest gene (**Figure 3d and Supplementary Figure 6e**), while loss of TADs during differentiation does not correlate with gene expression dynamics of the nearest gene. On the other hand, this process is associated with the increased condensation of B compartments (**Figure 3c,d and Supplementary Figure 6d-e**). The mechanism by which TADs can control gene expression of certain genes still remains largely unclear throughout the field, and will continue to be the subject of future investigation.

Comment 2: “The observation that TAD boundaries gained in CMs are generally depleted in B to A transitioning compartments conflicts with the gene expression data in Figure 2”.

While CM-gained boundaries in B to A regions show a slight depletion by the DI method (**updated Figure 3c**), there is actually a slight enrichment by the insulation score method (**new Supplementary Figure 6d**). However, as these boundaries represent only a small subset of TADs (27 and 16 for the DI and insulation score methods, respectively; refer to the relevant **new Source Data Tables**) neither of these changes reached statistical significance. The fact that the trends we observed depend on the specific method used to determine TAD boundaries highlights the current limitations in the field in reliably determining TADs and the associated complexity in interpreting the results from a biological standpoint. On the other hand, both TAD calling methods confirm that lost TAD boundaries are enriched in regions that go from A to B or are constitutively B, strengthening our confidence in this finding. Of note, the fact that neo-TADs are not significantly enriched in B to A regions is compatible with upregulated genes being enriched in such dynamic compartments. We think this observation reinforces the conclusion that TAD and compartment changes can act independently to control gene expression at different levels. We have sought to clarify this point with the **updated lines 171-173** *“In contrast, boundaries that are gained during differentiation are associated with a modest but significant activation of the nearest gene, but are not associated with a transition from inactive to active compartment”*

Comment 3: “While CAMK2D does better exemplify the correlation between a gained boundary and transcription (Fig 3f), there does seem to be residual TAD structure even in the CM stage.”

This was a valuable observation, and indeed our new analyses showed that the neo-boundary between *CAMK2D* was only identified by one of the two TAD calling methods. Thus, we have now included a different example in the genomic region surrounding *LMO7* (**updated Figure 3e**), which gains a TAD boundary that is identified by both methods coincidentally with *LMO7* upregulation.

Comment 4: “The correlations between open chromatin regions and transcriptional activity are also unclear. For eg: B to A switching compartments are enriched for ESC ATAC peaks, although they are depleted for any differentially expressed genes at this stage”.

We thank the Reviewer for pointing this out. We submit that this likely reflects the overall more accessible chromatin in ESCs, and we have added a sentence discussing this point at **updated lines 189-191**: “While we also observe an enrichment of hESC-specific peaks in B to A regions, this is less strong than CP- or CM-specific enrichment and is likely an effect of the overall more open chromatin in hESCs.” As we address in the replies to **Comments 6 and 7** below, the connection between chromatin accessibility and gene expression was further clarified by the new analysis of ChIP-seq data for cardiac transcription factors and CTCF.

Comment 5: “The relevance of the analysis about TAD size and numbers within compartments (Fig 3) is currently unclear”.

We sought to make the small point that when we separate out TADs by compartment we observe smaller TADs in the A compartment (and thus inherently more TADs, since the total amount of the genome in A and B compartments is roughly equal). We hypothesize that the higher degree of topological compartmentalization in the A compartment facilitates gene regulation, given that this compartment is more gene-dense. While statistically significant, this difference is not outside the expected TAD size of 500Kb-1Mb. We have clarified this point at **updated lines 165-167** “While genome-wide TADs range in size between approximately 500Kb-1Mb, TADs in the A compartment are smaller in size (Fig. 3b, Supplementary Fig. 6c), which is likely necessary for proper gene regulation in gene-dense areas.”

Comment 6: “The analysis of open chromatin regions and the underlying motifs may be further strengthened by integrating ChIP-Seq data for the transcriptions factors identified, such as those published by Ang et al (2016)”.

This was a very valuable suggestion, and we have included new analyses which integrate our ATAC-seq data with published ChIP-seq for GATA4, NKX2-5, and TBX5 in cardiomyocytes (Anderson et al., 2018; Ang et al., 2016) and for GATA4 in ESCs (Tsankov et al., 2015). These results strengthen and expand our original conclusions (**updated Figure 4, Supplementary Figure 7, and lines 200-221**). To summarize we found that:

- GATA4, NKX2-5, and TBX5 are enriched in CM-specific ATAC peaks, confirming our original predictions based on *de novo* motif discovery.
- GATA4 more strongly overlaps with NKX2-5 and TBX5 in B to A switching compartments compared to the rest of the genome, suggesting that these cardiac TFs could be involved also in large-scale chromatin organization changes.
- GATA4 is enriched both at CP&CM and CM-specific peaks in B to A regions, while NKX2-5 and TBX5 show much greater enrichment at CM-specific peaks. This suggests a temporal hierarchy for cardiac TF binding with GATA4 “pioneering” certain sites already in the CP stage followed by NKX2-5 and TBX5 in CM.
- Expanding our analyses of the *NEBL* locus, all three TFs bind at the promoter. We also included a second example of this regulation in the *ACTN2* gene.

Overall, these findings provide support for a model whereby key cardiac TFs, in particular GATA4, may act as a driving force facilitating both local chromatin accessibility and B to A transitions during differentiation.

Comment 7: “In Fig 5, does the analysis pertaining to gene size at the CM stage hold true even with the exclusion of the *TTN* gene?”

The statistics we present in **updated Figure 5a and Supplementary Figure 8b-c** are all based on the median value of 28 cardiac genes in B to A regions, and thus do not strongly suffer from the influence of possible outliers. Accordingly, excluding *TTN* from the gene set does will not alter the p-value statistics, as seen below in **Figure 1 for Reviewers only**.

Figure 1 for Reviewers only. Boxplot of gene size of upregulated genes peaking in CM stage subdividing either B to A compartment or heart development genes (GO term), outliers removed for clarity. *TTN* was excluded from the analysis.

Comment 8: “There are several predicted CTCF sites within the *TTN* gene, but only two show progressive loss of ATAC peaks through differentiation – does this correlate with CTCF binding strength, orientation or other attributes?”

We thank the Reviewer for this additional valuable suggestion. We have now integrated our ATAC-seq data with ChIP-seq for CTCF from ENCODE for both ESCs (ENCSR000AMF) and *in vitro* differentiated CMs (ENCSR713SXF), along with the associated motif orientation (**updated Figure 5d**). We find that the CTCF peaks that overlap decreasing ATAC signal also show decreased CTCF signal as measured by fold change over control. We also see a convergent orientation for all but one of these CTCF peaks, which is suggestive of chromatin looping within the locus (de Wit et al., 2015). This agrees with our earlier observation that the *TTN* gene is more compact in hESCs (**updated Figure 5e**). Further exploring the ChIP-seq data for cardiac TF (see the reply to **Comment 7** above), we find that the ATAC peaks that become accessible during differentiation are enriched for GATA4, NKX2-5, and TBX5 but not CTCF. These findings indicate that CTCF may contribute to silencing of the *TTN* locus in hESCs by promoting intergenic looping, a mechanism that is relieved during differentiation as cardiac TFs mediate transcriptional activation. We confirm this trend genome-wide, as CTCF is more associated with ESC-specific peaks compared CM-specific peaks (**updated Figure 5f**). Interestingly, differently from cardiac TFs, CTCF binding seems largely independent from A/B compartment changes.

Comment 9: “The postulated mechanism of an RBM20-mediated trans-interaction is not supported by these data and will require far more detailed experiments to validate. Instead, more DNA FISH

experiments to validate other interactions identified by this study will strengthen the current narrative”.

We acknowledge that the RBM20-mediated splicing factory involving loci on different chromosomes had not been functionally tested in our original manuscript. To address this weakness, we have performed a large number of additional experiments that validate our original findings and provide strong evidence for the mechanism we had postulated. These data are presented in **updated Figure 6f-g, new Figures 7-10, new Supplementary Figures 9-10**, and in the **updated text at lines 292-384**. Also refer to the updated discussion at **updated lines 422-461**. To briefly summarize:

- We expanded the 3D DNA FISH validation of Hi-C-identified *trans* interactions involving the *TTN* locus and other RBM20 targets. Data for six such genes are now presented, confirming an increased interaction with *TTN* in CMs compared to hESCs.
- We have functionally tested the hypothesis that such interactions are RBM20-dependent by disrupting RBM20 foci by means of transcriptional inhibition. In agreement with our prediction, this pharmacological intervention weakened the association between *TTN* and two important RBM20 targets (*CACNA1C* and *CAMK2D*).
- We have further probed our hypothesis by orthogonal approaches relying on gene editing. Knockout of RBM20 recapitulated the observations of the transcriptional inhibition, strengthening the notion that the *trans* interactions are RBM20-dependent. Further, deletion of the *TTN* promoter not only was sufficient to strongly impair the formation of RBM20 foci (indicating that the presence of the *TTN* mRNA is required for nucleation of these structures), but also weakened the association between *TTN-CACNA1C* and *TTN-CAMK2D*. Notably, a point mutation in *TTN* leading to loss of the protein product but not of the transcript had no effect on RBM20 foci nor on the *trans* interactions, further reinforcing our conclusion that the *TTN* mRNA is necessary for RBM20-dependent *trans* interactions.
- We observed that impairment of this mechanism leads to functional alterations in alternative splicing, as deletion of the *TTN* promoter impairs RBM20-dependent splicing in the *CACNA1C* and *CAMK2D* transcripts.

Collectively, these findings led us to propose the existence of a cardiac-specific RBM20-dependent *trans* interacting chromatin domain (TID) with the function of a splicing factory. We believe this is an important finding which will pave the way for future research into the prevalence, type, and function of similar TIDs in other cell types and conditions.

Reviewer 1 minor comments:

Comment 10: “The example of transient A compartment dynamics is not easily discernable from Fig 2. Either a better example or a clear detailing of the region undergoing the compartment switching is necessary to make this point. Additionally, an example of compartment dynamics of genes expressed in CP that undergo B to A compartment switching would also be interesting to add to this figure.”

We have included a clearer B-A-B transition in the CM-peaking gene *BMPER* (**updated Figure 2f**), and moved *CXCR4* to **updated Supplementary Figure 4c**. We have also added a gene track for *ELMO1*, a gene which expressed in MES and CP and transitions from B-A (**updated Supplementary Figure 4b**). Further, we have added dashed boxes to indicate the genomic regions that undergo compartment transitions in **updated Figure 2**, and **Supplementary Figures 4-5**.

Comment 11: “The text should be edited for minor typos. Eg.s; line 37 – “...during human development.”, line 43 – “...emerged as a key mechanism..”, line 57: “...feeds back on gene regulation”. “

We have corrected the manuscript accordingly, and have further edited the text during the revision process.

Reviewer 2

Reviewer 2 general considerations:

“Fields et al have performed in situ DNase Hi-C on samples from 4 stages of a cardiomyocyte differentiation protocol (following the same approach in their previous publication in Cell 2012, ref 7 where they interrogated histone profiles and RNA-seq, but not Hi-C then), and this time, along with two fetal human heart samples. Here, they have described changes in higher order chromatin architecture, and propose the presence of a cell type-specific “splicing factory”. Although they are the first to perform these analyses in human cardiomyocytes, nearly all of their findings are not surprising (e.g. ATAC-seq peaks enriched in A compartment, cardiac genes changing from B to A compartment and so on). While it is interesting to uncover another angle which is to group genes together in a splicing factory, and also to propose that the splicing units are pulled together at their templates by a splicing factor such as RBM20, the authors fall short of having strong evidence for the phenomenon”.

We appreciate the Reviewer’s helpful feedback. In our revised manuscript we have improved our bioinformatics analyses and improved the presentation of our datasets based on his/her advice. Furthermore, we greatly expanded the experimental validation of our original hypothesis regarding the existence of an RBM20-dependent splicing factory by combining pharmacological approaches and genetic perturbations via CRISPR/Cas9 gene editing.

Reviewer 2 comments:

Comment 1: “QC metrics are provided in supplement table 1. However, adequate details of the QC assessment are lacking to conclude if these hi-C libraries are robust and of good quality for downstream interpretation. Some of the few important QC metrics are % duplication of read pairs, fragment length distribution, fraction of hi-C read pairs within the same restriction fragments. It is also noted that the sequencing depth for each replicate does not seem sufficient to provide a high resolution interactome.”

We share the Reviewer’s concern regarding the need for high quality data sets, and indeed we had assessed this extensively. To confirm the quality of our data we have now included in **updated Supplementary Table 1** the percentage of PCR duplication during the sequencing library preparation (which was low at 1.8-8.04% for cell culture samples), and the distribution of strand directionality among the valid pairs (FF, RR, RF and FR; F=forward; R=reverse). HiC-Pro outputs this last statistic to evaluate the proportion of self-ligation events, which would increase the fraction of FR and RF reads. We do not see a bias toward these fractions; instead, the four strand combinations are approximately equally distributed, suggesting high quality ligation products. The fraction of Hi-C read pairs within the same restriction fragment is a statistic that does not apply to our DNase Hi-C data as interaction libraries were generated following random DNase digestion and therefore do not map to restriction enzyme sites (Ramani et al., 2016). Similarly, the fragment length distribution is not informative for DNase Hi-C data since the fragments represent ligation products of long-range interactions following randomly distributed DNase digestion (Ramani et al., 2016). On the other hand, **Supplementary Table 1** includes the number of *cis* interaction that involve regions >20 Kb apart (long-range) or < 20Kb apart (short-range): these figures confirm that as it would be expected our Hi-C libraries contain a majority of long-range interactions. To further assess the quality of our Hi-C data using orthogonal metrics we used QuASAR, which has been proposed in a recent ENCODE consortium pre-print as an effective way to compare quality across numerous datasets (Yardımcı et al., 2018). They found that an empirical cut-off of approximately ≥ 0.04

defines high-quality datasets, and the majority of our samples fell within this range as shown in **updated Supplementary Table 1**.

With regards to the sequencing depth, we acknowledge that the identification of significant local looping events would be facilitated by a larger number of reads in order to allow the generation of contact maps at a resolution of ~5Kb. However, our study was not designed to determine this type of chromatin interactions, which we submit would be more accurately studied using tethered or capture Hi-C (Hughes et al., 2014). On the other hand, we aimed for a sequencing depth sufficient to generate high quality contact maps at 40-500 Kb resolution in order to reliably determine A/B compartments and TADs, which represent the analytical focus of our study. Accordingly, our current depth of 140-280 million reads per sample is comparable to a number of recent studies that identified A/B compartments and TADs across multiple samples. These include

- Wu et al., 2017, Nat. Commun. → 2 replicates of ~200 million reads each
- Li et al., 2018, Cell Death Dis → 2 replicates with ~150 million reads total
- Schmitt et al., 2016, Cell Rep. → ~100-200 million reads per sample
- Nothjunge et al., 2017, Nat. Commun. → does not list reads, but ~30 million useable pairs per sample

Comment 2: “The ATAC-seq QC metrics provided are also inadequate and lacking of key information such as total number of peaks detected, FRiP, insert length, etc. It is also strange (and concerning) that the sequence data contains high percentage of mitochondrial RNA.”

We have included more information for the ATAC-seq as requested. (**updated Supplementary Table 7**). We included the number of detected peaks and the number of reads in peaks. Our values for the Fraction of Reads in Peak (FRiP) are above the ENCODE threshold of 0.2 for 5 out of 8 samples, with the remaining falling slightly below (<https://www.encodeproject.org/atac-seq/>). However, we note that FRiP values depend on the peak calling settings: since we wanted to focus on the most significant peaks, we used a q-value of 0.01, while the ENCODE pipeline for calculating FRiP utilizes a lower stringency threshold of 0.1. We have also included size distribution plots for all eight ATAC-seq samples (**updated Supplementary Figure 7a**). All these plots show an expected peak corresponding to sub-nucleosomal fragments smaller than 100 bp, and a peak at around 150-200 bp corresponding to mononucleosomes. Such findings collectively support the high quality of our data sets.

With respect to the burden of mitochondrial reads in ATAC-seq data (between 33% and 50%), this is a well-documented and unfortunate limitation of the technology, as mitochondria are difficult to separate from the nuclei during the preparatory steps (for instance refer to Montefiori et al., 2017 and Wu et al., 2016). Previous studies report that mitochondrial reads can make up 20-80% of the sequencing sample depending on the cell type and other technical differences. Since methods to minimize mitochondrial contamination are still largely in development, the current best practice is still to filter out mitochondrial reads during the analyses based on their sequence. In our samples, the number of mapped reads following such filtering is well within the range of what is required for reliable ATAC peak calling (36-72 million reads per sample).

Comment 3. “The authors perform A/B compartment analysis on the samples, and found that the proportion of A and B compartments are roughly equal. The authors mentioned “PCA of A/B scores separates the samples by differentiation time point (PC1) and cardiomyocyte purity (PC2)”. It is not conventional to compare PC scores between samples as the PC scores are generated individually for individual replicates, and it is hard to put meaning to the magnitude of difference between samples.”

We agree with the Reviewer that comparison between samples of raw PC1 values from the contact map generated by HOMER would be complicated, as “*precise qualitative nature of this association may differ*

slightly between experiments” [from <http://homer.ucsd.edu/homer/interactions/HiCpca.html>]. We were aware of this difficulty and took several steps to address it. First, to mitigate this weakness we used the HiCpca function, which performs a z-score normalization the PC1 output and thus generates scaled values that are comparable across samples. Moreover, we used a constitutively active subset of the genome to seed the positive values of the HiCpca output, with the aim of keeping the positive and negative values consistent between samples (manual inspection of the resulting PC1 tracks confirmed that no chromosomes were mis-assigned). Finally, and most importantly, we focused our analyses on those regions that significantly change PC1 sign, which indicates a substantial change in compartmentalization that is negligibly affected by small fluctuations in PC1. This aspect is further discussed in the reply to **Comment 7** below.

With regards to visualization of A/B compartmentalization changes across samples, we used a strategy recently employed by multiple other groups (Bonev et al., 2017; Stadhouders et al., 2018). We generated PCA plots of A/B compartment scores (PC1 values of the correlation matrix) to reduce the original 2-D genome matrix to a single value per bin, and then further reduce this multidimensional space to a single 2-D value for each sample. We have now improved this analysis by replacing PCA plots with t-SNE plots, which have the analytical advantage of visualizing the similarities and differences between samples using a non-linear dimensionality reduction to 2D without preserving specific variance in the axes (**updated Figure 1c and Supplementary Figure 5d**). Accordingly, we clarified the text on **updated lines 94-98** that we are using this approach to cluster and visualize the data, but not to assign quantitative variances: *“Using t-SNE to visualize and cluster in two dimensions either PC1 scores or HiC-Rep scores¹⁵ closely pairs replicates while generating a differentiation trajectory (Fig. 1c, Supplementary Fig. 3a, Supplementary Table 2). Fetal heart Hi-C most closely resembles in vitro cardiomyocytes but clusters separately, likely reflective of lower cardiomyocyte purity”*. To further support these findings we have also included an orthogonal method to visualize and cluster our data based on HiC-Rep to quantify the similarities and differences across samples (Yang et al., 2017). Results are included in **updated Supplementary Figure 3a and new Supplementary Table 2**, and confirm our original conclusions regarding the developmental trajectory of chromatin organization changes during hPSC differentiation.

Comment 4: “Line 43. “Spatial organisation of the genome in 3-D has emerged as key mechanism by which cells can control regions of chromatin accessibility” sounds a fundamentally important biological idea. what is the reference to support this?”

At **updated line 48** we have added references to two landmark studies which investigated the correlations between Hi-C patterns and open chromatin regions marked by histone marks and showing DNase hypersensitivity (Lieberman-Aiden et al., 2009; Wang et al., 2017). These studies support our statement that the 3D organization of the nucleus can regulate chromatin accessibility.

Comment 5: “Line 47. “... implicated in a range of diseases, including cancer and limb malformations” references 3-6 do not include the reference on Limb Malformations.”

We apologize for having omitted the reference to (Franke et al., 2016); this has now been added to **updated line 51**.

Comment 6. “The motivation statement in Line 48. “.. how these transitions occur temporally across stages of human lineage remains unaddressed, or how structural arrangement differs between pluripotent progenitors and terminally differentiated cells” is debatable accuracy. Many of the references in this manuscript precisely address these topics. This is exactly what lends to the lack of novelty for this manuscript.”

While previous studies have investigated chromatin organization changes during the differentiation of mouse embryonic stem cells (Bonev et al., 2017; Fraser et al., 2015) or the initial transition of human embryonic stem cells into early progenitors (Dixon et al., 2015), to our knowledge no study has yet integrated Hi-C with multiple other data sets at defined stages of differentiation of human embryonic stem cells into a fully differentiated cell type. Moreover, while a recent study published during the revision of our manuscript performed promoter capture Hi-C in hiPSCs and hiPSC-CM (Montefiori et al., 2018), the global chromatin architecture changes at multiple stages of *in vitro* cardiomyocyte differentiation have not been yet explored. Notably, the heart is a very important model to understand gene regulation as it is the first major organ to form during human development, and congenital heart disease represents a major cause of birth defects and infant mortality. Collectively, we respectfully disagree that our original manuscript lacked novelty, and submit that our findings and publicly available datasets will be very value to geneticists, developmental biologists, and cardiac biologists.

In addition to this, we trust that the Reviewer will agree that the novelty and impact of our revised manuscript were greatly increased by the additional validations and functional experiments which led us to identify a novel mechanism of gene expression control: the existence of an RBM20-dependent *trans*-interacting chromatin domain with the function of splicing factory (described in more detail in the response to **Comment 19** below). Finally, as a result of our major revisions we have also included DNase Hi-C data during differentiation of hiPSCs (**new Supplementary Note** and **new Supplementary Fig. 5**), further expanding the resource provided by our work.

To clarify the novel aspects of our study we changed our statement at **updated lines 51-54** to read: *“While these studies have demonstrated regulated regions of chromosomes, how these transitions occur temporally across multiple stages of human lineage commitment remains to be addressed. Further, to what extent such changes causally impact gene expression is still poorly understood”*.

Comment 7: “In Figure 1f, compartments were defined as “dynamic”. The statistics used here is unusual. Can the authors justify why is ANOVA used here. It is unclear how was the comparison done to obtain the p-value. Was it done across the time point? Are the PC scores normally distributed, and independent? If not, this test is deemed inappropriate.”

There is currently no gold standard in the Hi-C field to determine A/B compartment changes. Previous studies have either used no cut-off except for a PC1 sign change (Barutcu et al., 2015; Wu et al., 2017), an arbitrary PC1 score change (Nothjunge et al., 2017), or an ANOVA test of PC1 scores combined with requirement for a sign-change (Dixon et al., 2015). We chose to follow this last strategy by first applying a significance cutoff for PC1 scores differences across all samples based on one-way ANOVA-calculated p-value < 0.05, and then further filtering for regions where the average PC1 score changed from positive to negative (or vice versa). This stringent approach was aimed to reduce the burden of false positive regions where PC1 sign changes merely result from small fluctuations around 0 (thus representing unclear compartment calls). The use of an ANOVA-calculated p-value was preferred over an arbitrary PC1 change cut-off since as the Reviewer correctly pointed out in Comment 3 even though the PC1 values are z-scored these values are not completely comparable across multiple samples.

To rigorously test the appropriateness of a one-way ANOVA test for our dataset, we performed a Shapiro-Wilk test and a Bartlett test on the PC1 values of the 5293 genome bins analyzed. The Shapiro-Wilk test looks at the residuals from the ANOVA fit and is a test of normality, while the Bartlett test looks at the variance across the samples and tests for different variances between groups. The resulting p-values are presented in density histograms in **Figure 2 for Reviewers only** below, which indicate that neither test showed a strong skew

toward significant p-values less than 0.05. This confirms that for the vast majority of genomic bins the assumptions of the one-way ANOVA test were satisfied, and thus supports the use such test for our analysis.

Figure 2 for Reviewers only. Results of the indicated statistical tests calculated on the PC1 score values for each of the 500 Kb genomic bins considered for statistical analysis using one-way ANOVA. The resulting p-values are shown as density histograms.

Regarding the use of the term “dynamic” in **old line 97** and **old Figure 1i-j** and **Supplementary Figure 3**, we realize that our phrasing might have been unclear. Indeed, these analyses examined the different properties of contact matrices within A or B compartments, but not within regions that are switching A/B. We have clarified this in the text at **updated lines 110-112**: *“By integrating the A/B compartment information across differentiation with the interaction contact maps, we noticed that many of the strongest gains in long-range interactions are associated with the B compartment (Supplementary Fig. 3c)”* (also refer to the reply to **Comment 8** below). We also clarified the heading for this section at **updated line 79** by re-phrasing it to *“interrogation of higher order chromatin structure during human cardiogenesis”*. Collectively, these modifications clarify that we are not looking at fluctuations within the A or B compartments because, as the Reviewer pointed out, the PC1 statistic is not well suited to reliably determine such fluctuations.

Comment 8: “Figure 1i. What is the meaning of “cis-interaction score”? How was it generated quantitatively? Similar to figure 1j, what is trans interaction score?”

Data plotted in **old Figure 1i-j** indicated the enrichment of homotypic interactions (A-A or B-B) over heterotypic interactions (A-B), expressed as ratio of the median raw counts for read pairs falling within these categories in a given sample. The data was computed separately for interactions happening within the same chromosome (*cis*) or between chromosomes (*trans*). In **updated Figure 1i**, **Supplementary Figure 3e-f**, and **Supplementary Figure 5e** we have improved this analysis to also control for varying genomic distances between compartments by generating “compartmentalization saddle plots” as described in (Schwarzer et al., 2017). These plots represent distance-normalized average contact frequencies between compartments subdivided by PC1 score percentile. Briefly, each bin is assigned to the corresponding decile based on its PC1 value, and the count of all interactions involving another bin in a given decile is assigned an observed/expected value based on the average value of all interactions observed at the same genomic distance. The \log_2 average observed/expected for each pair of percentiles is then plotted. This was done for both *cis* and *trans* contacts. The conclusions drawn from the new analyses confirm our earlier results showing that during differentiation the interaction between B-B in *cis* is strengthened while the A-A interactions in *cis* are weakened. In contrast, A-A interactions are favored in *trans* consistently across differentiation.

To further expand on these observations, we have also plotted *cis* and *trans* contact frequencies calculated as just described against the genomic distance involved (**updated Figure 1j and Supplementary Figure 3g**). While B-B *cis* interactions show the strongest signal at short range in all stages of differentiation, long range *cis* interactions are remarkably strengthened in CM. These analyses provide quantitative and genome-wide support for the observation that B compartments in *cis* compact at long range during differentiation (exemplified in **updated Supplementary Figure 3c**).

Comment 9: “Similar questions for statistic significance arise in other figures: e.g. Fig3e, Fig6a. There do not appear to be differences in the medians between groups, and yet the difference is deemed statistically significant?”

All data used to generate the Figures are provided in **new Source Data Tables** and can be consulted to confirm the results of our statistical analyses. When working with very large datasets with hundreds to thousands of values it is not uncommon that even small changes in median/mean value can lead to significant p-values. Thus, it is important to give biological interpretation to the results not only from the standpoint of statistical significance but also considering of the magnitude of the observed change.

Specifically to **old Figure 3e**, the statistically significant differences reported were the result of multiple comparisons across all groups, and thus in some cases they did not address a clear biological question. To clarify the goal of the analysis, in **updated Figure 3d** we indicate p-values from a one-sample t-test to determine whether there is a change in gene expression from hESCs for each of the analyzed groups (a median expression change value of 0 would indicate no difference). Confirming our earlier conclusions, we observe a significant upregulation of the nearest gene to a TAD boundary gained in CMs. Of note, we repeated this analysis also having applied a second TAD calling method (an aspect described in more detail in the replies to **Comments 17 and 18** below), which confirmed our conclusion by clearly showing an upregulation of the nearest gene to CMs neo-TADs. We have updated the text to reflect these new analyses and to clarify that the absolute difference we observed is modest in its magnitude (**updated lines 171-173**): *“In contrast, boundaries that are gained during differentiation are associated with a modest but significant activation of the nearest gene, but are not associated with a transition from inactive to active compartment (Fig. 3c-d, Supplementary Fig 6e)”*.

With respect to **Figure 6a**, we confirm that there is a modest but significant difference in all timepoints between z-scored *TTN* *trans* contacts to the A or B compartments. However, the key conclusion of this analysis was that *TTN* *trans* contracts switch from being slightly enriched within the B compartment in hESCs to being clearly enriched in the A compartment in CM. We have now included the median z-scores in the text to emphasize this point at **updated lines 270-271**: *“(Fig. 6a, hESC median Z-score A: -0.26 B: -0.07, CM median Z-score A: 0.19, B: -0.29)”*.

Comment 10. “There are some bits of novelty within: e.g. Line 97. “Examining the dynamic regions, many of the strongest gains in long-range interactions are associated with the B compartment. .. stronger intra-chromosomal contacts occur between homotypic regions” .. but what does this mean functionally? There is only one example given in Suppl Fig 3A. what is the global picture?”

We appreciate the Reviewer pointing this out. As we described in our response to **Comment 8 above**, we have performed additional genome-wide analyses for homotypic and heterotypic *cis* interactions and shown that in hESCs we see the strongest long-range contacts in A-A, while in CMs this trend is reversed in favor of a strongest signal for B-B. The biological interpretation of this phenomenon is presented at **updated lines 121-**

126: “Together these observations suggest a model whereby during differentiation heterochromatic (B compartment) regions condense and pack more tightly, specifically within chromosomes in cis, while inter-chromosomal interactions are most likely to occur between active regions (A compartment) independent of cell state. This is consistent with recent electron micrograph studies showing that heterochromatic regions are more densely packed relative to euchromatic regions in differentiated cells¹⁸”.

Comment 11: “Line 111. states the authors set out to “investigate for genes that may be drivers of compartment changes...” but they end up not doing so. At least not with their RNA-seq. Instead it was their motif analysis of ATAC-seq related sites that throw up GATA and NKX which may be implicated in this role?”

We apologize for the confusion about this statement: we had intended to refer to those genes whose change of expression might result in (or be the result of) a compartment change. Given that a genomic bin of 500 Kb may have 1 to 3 genes in it and that many genes in the human genome are not expressed, we integrated RNA-seq data with the A/B compartment calls to identify genes whose expression may be correlated with compartment changes. We have clarified this by changing the sentence **at updated lines 129-131** to now read: “To investigate genes whose expression may be regulated in relationship with compartment changes, we performed RNA-seq on the same stages of differentiation, along with the fetal hearts (Supplementary Table 3)”.

On the other hand, as the Reviewer correctly points out our original analyses of ATAC-seq data pointed at GATA4 and NKX2-5 as potential *trans*-acting drivers of compartment changes by virtue of their transcription factor role. In the revised manuscript we have expanded these analyses to include integration of ATAC-seq and Hi-C data with published ChIP-seq for these and other cardiac transcription factors. This aspect was described in detail in the response to **Comment 6 from Reviewer 1** above, and is further expanded in the response to **Comment 13** below.

Comment 12: “Line 115. “In contrast to Hi-C. most of the dynamic changes in gene expression are cell-type specific.”. The authors should elaborate further on this. What are the gene signatures that are cell type specific? Are these differentially expressed genes?”

We were indeed referring to differentially expressed genes that are expressed in a largely cell-type specific manner. We have clarified this at **updated lines 132-134**: “In contrast to Hi-C, differentially expressed genes are largely expressed in a cell-type specific manner (Fig. 2b, Supplementary Fig. 4a)”.

Comment 13: “Line 174. Only one example of the motif is given at the NEBL locus. what are others? It is the important message that GATA and NKX could be the key drivers of chromatin organisational change?”

We agree with the Reviewer this is an important point. As also described in detail in the response to **Comment 6 from Reviewer 1** above, we have now incorporated analysis of ChIP-seq data for GATA4, NKX2-5, and TBX5 in cardiomyocytes (Anderson et al., 2018; Ang et al., 2016) and for GATA4 in hESCs (Tsankov et al., 2015). We observed that in CMs all these cardiac transcription factors are enriched in regions that switch from B to A, and have greater overlap to those regions compared to the rest of the genome (**updated Figure 4g**). This and other analyses shown in **updated Figure 4 and Supplementary Figure 7** strengthen the conclusion that these TFs may contribute to changes in both chromatin accessibility and compartmentalization during cardiogenesis.

Regarding *NEBL*, in agreement with our original motif analyses we observe binding of all three factors at the promoter (**updated Figure 4j**). We have also included another example of this in the *ACTN2* gene in **updated Supplementary Figure 7h**.

Comment 14: *Fig 2f. the PC1 levels to suggest the transition B -> A -> B are small vis a vis neighbouring PC. Besides the regions that go from A-B-A and B-A-B appear to be regions that are already in the borderline zone of compartments. This makes it concerning that the change could be noise as generated by the algorithm calling the compartments. How do the authors exclude the possibility that these small changes are not noise?"*

We shared the Reviewer's concerns on this, and to mitigate this possibility we had also utilized the one-way ANOVA p-value for PC1 differences to filter out regions whose fluctuations between replicates was highly variable around the 0 axis (uncertain compartment calls, as also discussed in the reply to **Comment 7** above). We do of course recognize that a small subset of imputed compartment transitions might still be the result of inaccurate compartment calls. Nevertheless, our analyses of the correlation between gene expression changes and transitory compartment switches supported the notion that at least some of these dynamics could have regulatory function. Indeed, we observed that these regions exhibited distinct enrichment in gene expression patterns compared to B-A or A-B regions, and that in particular the B-A-B regions were enriched in genes with peak expression in CP (**Figure 2c**).

As already described in the reply to **Comment 10 from Reviewer 1** above, to show a clearer example of B-A-B transition we have substituted the data for *CXCR4* (now showed in **updated Supplementary Figure 4c**) with that for *BMPER* (**updated Figure 2f**). Further, in **Figure 3 for Reviewers only** below we show an additional representative example of a portion of chromosome 5 in which two regions show B-A-B or A-B-A transition that are clearly demarcated and outside of compartment border zones.

Figure 3 for Reviewers only. Gene tracks of PC1 values for a portion of chromosome 5 involved in dynamic compartment transitions during differentiation. Regions of note are indicated by dashed boxes.

Overall, while our general conclusion remains that the majority of compartment changes are unidirectional, these observations suggest that at least a subset of putative transitory transitions might be functionally relevant.

Comment 15: *“Figure 3a. The dendrogram is not visible to make any meaningful conclusion. Also, by given a 2-window width (80kb) allowance for error seems less stringent. Even at such low stringency, the agreement between two replicates is only about 60-70%.”*

We realize that the dendrogram and heatmap in **old Figure 3a** were unclear because of the inclusion of self-correlations with a value of 1. To bypass this limitation, we have replaced this figure with t-SNE plots clustering the samples by their Jaccard distance based on TAD boundary overlap. Notably, as already discussed in the reply to **Comment 1 from Reviewer 1**, we have expanded and strengthened our TAD analyses by also repeating them with a second TAD caller based on the insulation score (Crane et al., 2015). The t-SNE plots of Jaccard index values for both TAD callers are shown in the **new Supplementary Figures 6a-b**, while the raw Jaccard values are reported in **Supplementary Table 6**.

Regarding the determination of shared TAD boundary between two samples, we chose to use a flexible window approach since all currently available TAD callers are subject to local noise which affects the precise location of the boundary. This is exemplified by a recent comparative analysis of TAD calling methods in which no flexible windows was implemented to calculate Jaccard indexes for concordance of TAD boundaries. (Forcato et al., 2017). The authors observed showed a low median agreement for sample replicates, which ranged between 2% and 37% depending on the TAD caller used (Forcato et al., 2017). Similarly, while we saw a replicate concordance based on Jaccard indexes of about 60-70% using the DI method, we noticed a higher score of approximately 75-85% using the insulation score method. This discrepancy may be the result of the distinct approaches to define a TAD: the TAD caller based on the DI method that we originally used allows for regions of boundary (meaning that adjacent TADs are not required to start and end in consecutive bins), while the insulation method generates a continuous set of TADs. These points illustrate the current limitations in reliably determining TAD boundaries, and show that replicate concordance is highly dependent on the specific method used. Collectively, we deemed the use of a flexible window necessary to avoid a large number of false positive differential TAD boundaries by increasing the stringency of the criteria that had to be fulfilled to define a neo-TAD.

We devised the windowing method following Schmitt et al., 2016: *“Merging of adjacent boundary bins was performed because often times larger TAD boundaries (up to 400Kb) may result in slightly shifted (by a few bins) boundary calls between samples, and though they do not directly overlap, then both are a bin within the same boundary region.”* Thus we used a medium stringency cut-off of 80 Kb to compare the similarity between biological replicates, and we used a stronger cut-off of 200 Kb to define distinct TAD boundaries between time points in order to reduce the burden of false positive differentiation-specific boundary changes due to technical/computational artifacts. Finally, we performed these analyses using the two aforementioned distinct TAD callers, and both confirmed our key findings (**updated Figure 3 and Supplementary Figure 6**).

Comment 16. “Fig 3f. TAD changes as defined by the algorithm dont match to the heat map visually. It simply looks like the original TAD marks between ANK2 and CAMK2D are blurred?”

As also described in the reply to **Comment 3 from Reviewer 1** this was a valuable observation, and indeed the weak neo-boundary originally identified by the DI method was not confirmed by the second TAD caller. We have thus included a clearer example of a CM-specific neo-TAD around the *LMO7* locus (**updated Figure 3e**). Upon differentiation a new boundary appears between the *LMO7* and *COMMD6* genes, coincidental with *LMO7* upregulation. Of note, this new example was supported by both TAD calling methods.

Comment 17: “The authors called A/B compartment at 500kb resolution while they call TADs at 40kb resolution. Yet, they claim that TAD organization is dynamic across differentiation but independent of AB compartment changes. The differences observed could be an artefact of different resolution used.”

The choice to call TADs and A/B compartments at different resolutions is common in the field. Indeed, while TAD determination is based on the count of short-range interactions that are very abundant, A/B compartments are calculated based on correlations between long-range interactions across the entire chromosome that are less frequent. Thus, A/B compartment calls are substantially improved by increasing the read count per bin as a result of an increased bin size. This principle is exemplified by these manuscripts:

- (Wu et al., 2017) → 500 Kb resolution for compartments; 40 Kb resolution for TADs
- (Schmitt et al., 2016) → 1 Mb resolution for compartments; 40 Kb resolution for TADs
- (Stadhouders et al., 2018) → 100 Kb resolution for compartments; 50Kb resolution for TADs

Aside from this important technical reason, we decided to focus our analyses on A/B compartments at 500 Kb resolution because this also reduces the instances of genes falling into multiple compartment types.

Having clarified our rationale, we appreciate the suggestion of the Reviewer to confirm that the different dynamics of TAD and A/B compartment changes during differentiation is not due to the different resolutions used for these analyses. To this end we also performed the A/B compartment analyses at 40 Kb resolution. First, we observed that a similar fraction of the genome was deemed to change compartment during cardiogenesis (16.4% vs 19.1% calculated at the original 500 Kb resolution), and that in agreement with our original conclusion the majority of these changes were unidirectional (33.28% A→B and 49.47% B→A, vs 33.10% and 48.60%, respectively, as calculated at the original 500 Kb resolution). Furthermore, integrating these results with the each of the two differential TAD boundaries analyses confirmed our initial conclusions (**Figure 4 for Reviewers only**, below). Our main point is that those boundaries that are lost in CMs are enriched in constitutive B compartment and regions transitioning A to B, and this point holds true in all four analyses. This indicates that our conclusions regarding the relationship between A/B compartments and TADs are neither an artifact of our methodology nor of the resolution.

Figure 4 for Reviewers only. Enrichment of TAD boundaries between hESC and CM state within A/B compartment dynamics calculated with the DI method or insulation score method. A/B compartment calls were done at 500 Kb resolution (top) or 40 Kb resolution (bottom). Observed/Union Set p-values calculated by chi-sq test; * <0.05 ; ** <0.01 ; *** <0.01 . The 500 Kb analyses are shown in **updated Figures 3c and Supplementary Figure 6d**.

Comment 18: *“It is also not justified on the criteria used to assess if a TAD is overlapping with a AB compartment. It is also unsure how many boundaries are called by the authors, It is very hard to assess the quality of the TAD calling as well unless the authors are diligent in providing clearer heatmaps showing TADs. The authors are lacking evidence of TADs transition during the differentiation process”.*

For comparison between TADs and A/B compartments, TADs entirely included within a compartment were assigned to it, while TADs spanning a compartment boundary were assigned to the compartment that had the largest overlap ($>50\%$). This same approach was taken to assign differential TAD boundaries to compartments, in which case the overlap was measured for the region between the two adjacent TADs. We have clarified these aspects in the Methods at **updated lines 568-574**: *“For comparison between A/B compartments and TADs we used `bedtools intersect -u -f 0.51 -a <TADs.bed> -b <compartment.bed>` to assign the TADs to a specific compartment, and the majority compartment if it spans a boundary. For differential and shared boundaries between ESC and CM we used `bedtools intersect -u -f 0.51 -a <TADs_boundary.bed> -b <compartment.bed>`, p-values were calculated based on a Chi-squared test between the distribution of differential or shared TAD boundaries versus the distribution of the union set of TAD boundaries.”*

The number of TADs called at the various time points is indicated in **updated Figure 3b and Supplementary Figure 6c**, which also report the average TAD size. We observed more TADs with smaller average size using the insulation score method compared to the DI method. However, both methods have significantly larger average TADs within the B compartment, confirming our earlier conclusion.

As described in the reply to **Comment 16** above, we now include a clearer example of TAD structure in **updated Figure 3e**, in which the TAD boundaries inferred by the two methods we used are also indicated.

We trust that the replies to **Comment 15 through 18** have clarified and corroborated our conclusion regarding TAD dynamics during differentiation.

Comment 19: *“As commented, the presence of a splicing factory is not convincing. Chromatin organization, particularly CTCF mediated chromatin loops has been shown to regulate splicing, however the link between spatial proximity of genes and co-splicing is relatively new. The concept of “splicing factories” in addition to the more conventional “transcription factories” is important. The genes may be grouped together because they are co-transcribed, but not necessarily because they are co-spliced. Since RBM20 is an RNA binding protein, would there have to be communication with another DNA binding protein that brings these gene loci together. More evidence is needed to make this bold claim.”*

This is a fair point. As also described in the reply to **Comment 9 from Reviewer 1**, we have performed a large number of additional experiments that further validate our original findings and provide strong evidence for the mechanism we had postulated. These data are presented in **updated Figure 6f-g, new Figures 7-10, new Supplementary Figures 9-10**, and in the **updated text at lines 292-384**. Also refer to the updated discussion at **updated lines 422-461**.

In summary, we functionally perturbed our biological system in three orthogonal ways:

- Pharmacological inhibition of transcription, which greatly weakened the localization of RBM20 into foci and resulted in impaired interactions between *TTN* and other RBM20 target genes located in *trans*.
- Genetic ablation of *TTN* transcription, which remarkably reproduced the effects of global transcriptional inhibition indicating that the *TTN* mRNA is required for the nucleation of RBM20 into foci. This conclusion was further strengthened by results in another gene edited line where a premature nonsense mutation in *TTN* abolished protein expression but did not affect its transcription, as in this case no effect was observed on RBM20 foci and on the *trans* interactions.
- Genetic ablation of RBM20 expression, which also weakened the *trans* interactions involving *TTN* and RBM20 targets.

Collectively, these experiments point toward the existence of an RBM20-dependent *trans* interacting chromatin domain. Notably, disruption of this structure due to ablation of *TTN* transcription has a *trans*-acting effect on the alternative splicing of the RBM20 target mRNAs *CACNA1C* and *CAMK2D*, which suggest that RBM20 forms a splicing factory as we had originally hypothesized.

The Reviewer suggested that *TTN* could be colocalized with other RBM20 transcripts because they are co-transcribed rather than co-spliced, and that DNA binding proteins could alternatively mediate this process. However, these possibilities were ruled out by our findings that loss of RBM20 expression (which affects the splicing but not the transcription of these genes) was sufficient to impair the colocalization of these loci. Mechanistically, this can be explained by the fact that RBM20 binds to nascent pre-mRNAs and, accordingly, co-localizes with the *TTN* locus (**updated Figure 7a-b**), its main target possessing more than 50 binding sites (Li et al., 2013; Maatz et al., 2014). Thus, we propose that the proximity between *TTN* and other genomic loci in *trans* is a result of RBM20 binding to the relevant nascent mRNAs which are still chromatin-associated (**updated Figure 10e**). Of course, this model does not exclude that other RNA- or DNA-binding proteins whose localization depends on RBM20 might also be involved in establishing and/or stabilizing this interaction.

Comment 20: “Finally, given that the changes in higher order chromatin architecture as described here are not surprising, the authors should have to give more insight into the local Enhancer-Promoter interaction changes during development. For example, In Page 9, line 193, they imply that the isolation of important developmental genes allows for individual regulation of these genes, they arrive at this conclusion by analysing separation of these cardiac genes from the next gene promoter. While these genes may not be situated very close to other genes, they may be regulated by distal enhancers which a deeper interactome would be able to map out. Hence it will be interesting to look at the Enhancer-Promoter interaction dynamics, beyond just the structural changes which have turned out to be as expected.”

As discussed in the reply to **Comment 1 above**, we did not design our study with the primary goal of identifying local interactions such as those between promoter-enhancers, which would be more reliably studied using tethered or capture Hi-C (Hughes et al., 2014). Accordingly, we did not strive for a sequencing depth optimal for the detection of promoter-enhancer interactions. While our datasets could still be mined to gather some information on these aspects, we submit that this would be beyond the goals of the current work in which we wanted to primarily focus on large scale chromatin changes. Indeed, considering the already very substantial scope and length of the revised manuscript, we believe that further investigations stemming from our publicly available datasets will be better suited for future reports.

To address the specific point raised by the Reviewer, while we agree that the separation of cardiac genes from the nearest neighbor does not preclude some of them being regulated by long-range enhancers, studies have shown that approximately half of all enhancers still act on the nearest active gene, and that the probability of

chromatin interactions strongly decays over increasing genomic distances (Sanyal et al., 2012). Thus, the larger than average separation of transcriptional start sites of developmental genes would in principle provide more genomic space for enhancers or insulators to uniquely act on such genes, thus facilitating local regulation of gene expression. While currently largely speculative, this is an intriguing hypothesis that will be the focus of future investigation.

Reviewer 3

Reviewer 3 general considerations:

I thought this was an interesting paper and would be happy to recommend publication if the authors could improve some of the computational analysis as detailed below:

We thank the Reviewer for the encouraging and useful comments. We have revised our study according to his/her recommendations to strengthen the computational analyses.

Reviewer 3 comments:

Comment 1: *“At around line 97 the authors describe their examination of the dynamic regions, but it was not clear how they corrected for the fact that interactions are more likely to be with adjacent regions in the same compartment. They say – “In the pluripotent state, the median cis interaction signal between A compartments...”. How was the median cis interaction signal derived? For (e.g. A-A) is it simply the median of normalized contacts for all possible pairs of A compartment bins? Or does the calculation involve further normalization e.g. comparing observed to that expected? If it is the former, I think that some comparison against the background contact level in relation to sequence separation is needed.”*

We appreciate the Reviewer pointing this out. We addressed this extensively in the response to **Point 8 from Reviewer 2**. Briefly, we recognized that our original analyses were not corrected for genomic distance as we only relied on normalization of homotypic interactions (A-A or B-B) over heterotypic ones (A-B). We therefore generated new analyses shown in **updated Figure 1i-j and Supplementary Figure 3d-g** that are now corrected for distance and reflect the observed/expected based on the empirical average contacts for a given distance. The conclusions drawn from the new analysis agree with our original ones and also provide additional insights into how homotypic interactions at different ranges change during differentiation (also refer to the reply to the next comment).

Comment 2: *The authors suggest in their analysis that the average proportion of the genome belonging to either the A or B compartments does not change much during differentiation. However, it is not clear whether the A or B compartments become more/less fragmented (i.e. what are the average sizes of pieces of A and B compartment)? If there is a big change, the shift in the distribution of interaction signals could be partially attributed to fragmentation. To resolve this, the authors should calculate the observed/expected ratio instead of using the contact score itself. The expected value could be straightforwardly derived from an empirical averaging of contacts for different sequence separations (e.g. at 40 kb, 80 kb, 120 kb etc.) within each chromosome. Alternatively, and perhaps more informatively, instead of using a simple median value they could group the A-A or A-B contacts into different groups according to sequence separation and compare each group specifically. That would give a better idea whether the observed change in homotypic/heterotypic compartment score is mostly due to short or long range contacts, or both.*

This was a very valuable suggestion. As we described in relation to the previous point, in addition to generating distance-normalized contact matrices for compartment interactions, we have also generated plots of relative contact frequency between compartments at different distances in either ESCs or CMs (**updated Figure 1i-j and Supplementary Figure 3d-g**). These analyses clarified that the stronger interaction in *cis* between B compartments observed during differentiation is predominantly due to an increase in long range interactions

We also included in **updated Supplementary Figure 3b** the empirical distribution of compartment sizes across differentiation. While there is no change in A compartment sizes, there is a minor but significant shift in B compartment sizes, which are slightly smaller in CMs. However, given that the change in enrichment for homotypic/heterotypic interactions was observed even after distance-normalization, the small change in B compartment sizes is not likely to account for the dramatic shift we see in the *cis* interaction profiles.

Comment 3: “Moving to the analysis of the dynamics of TAD organisation during differentiation (lines 140-154), there are a lot of statements made where the results are only weakly significant (see e.g. Fig 3e) or unsubstantiated. For example, is it really true that the changes in TAD boundaries are independent of A/B dynamics (lines 150-151)?”

We agree that the analysis of TAD boundaries needed to be strengthened. This is also extensively discussed in the replies to **Comment 1 from Reviewer 1 and Comments 15 through 18 from Reviewer 2**. To summarize, in order to improve the confidence in our results on TAD dynamics we repeated all our analyses with a second TAD caller, such that we now present results with both the DI method (Dixon et al., 2015) and the insulation score method (Crane et al., 2015) (**updated Figure 3 and Supplementary Figure 6**). We find that while only a small portion of TAD boundaries change their distribution, the results are similar using both methods (**updated Figure 3a**). Further, we confirm that TAD boundaries that are gained in CMs are near up-regulated genes (**updated Figure 3d and Supplementary Figure 6e**). In contrast, TAD boundaries that are lost during differentiation are enriched within the B compartment. Finally, our new analyses confirmed our original conclusion that differential boundaries do not seem to be significantly correlated with A/B compartment switching (**updated Figure 3c and Supplementary Figure 6d**).

Comment 4: “In Figure 5d, there are two TSS within the TTN locus – I think one must be missing from this Figure? In the references there are papers describing the internal TSS in more detail, but it would be helpful for the reader's if the authors could show them in the Figure and discuss the two TSS briefly and what is special about them”

We have now added an indication for the internal TSS in *TTN* to **updated Figure 5d**, as can be seen below. We have edited **updated lines 245-247** to emphasize the importance of the second isoform: “*Both isoforms have important developmental roles as mutations downstream of the Cronos TSS are significantly more deleterious than those only in the long isoform*³¹”.

Comment 5: “In Figure 5d, the authors do not say what sort of cells were used to determine the CTCF binding data? The citation is for the ENCODE project paper, but the dataset used is not specified. Is it relevant for the cells they are looking at, or do the authors need to carry out their own experiments?”

According to the Reviewer's useful suggestion, in the revised manuscript we have integrated cell type-specific CTCF ChIP-seq data from ENCODE for both hESCs (ENCSR000AMF) and *in vitro* differentiated CMs (ENCSR713SXF), along with motif orientation direction for those. We discuss this aspect in depth in the reply to **Comments 6 and 8 from Reviewer 1**. To summarize, we found that the ATAC peaks with decreased signal during differentiation correlated with loss of CTCF binding, while gained ATAC peaks corresponded to well-known transcription factors (GATA4, NKX2-5, and TBX5). This pattern is consistent genome-wide, as we find

that CTCF is more associated with hESC-specific ATAC peaks compared CM-specific peaks **updated Figure 5f**.

Comment 6: “Furthermore, also in Figure 5d, the observation that ATAC signal diminished around CTCF sites during lineage specification is interesting. However, I didn’t understand how the authors concluded this might be due to a physically proximal chromatin hub in hESC’s? It would be interesting to discuss this”.

We apologize for the confusion on this point. We had hypothesized this partially based on a pre-print (now published) that showed how at the single cell level co-accessibility (ATAC peaks) correlates with physical proximity between loci (Pliner et al., 2018). Accordingly, virtual 4C had indicated a stronger intragenic signal from the promoter of *TTN* in hESCs, which was suggestive of gene compaction. We were excited to see that these intuitions were further corroborated by the analysis of cell-specific CTCF peaks just described above. Indeed, locations that have decreasing ATAC and CTCF signal in CMs show CTCF motifs largely in a convergent direction, which is predictive of intergenic looping (de Wit et al., 2015). Taken together we concluded that during differentiation *TTN* experiences decompaction within in the gene body coincident with activation. We have now added text to clarify this point at **updated lines 251-258**: *“Using CTCF ChIP-seq data from hESCs and CMs, we find that the CTCF sites that overlap hESC ATAC peaks show decreased occupancy during differentiation. With the exception of one CTCF site located at the transcriptional termination site, the CTCF motifs on TTN are orientated in a convergent direction, predictive of the presence of intragenic looping. Indeed, during differentiation there is a decreased interaction between the TTN TSS and the TTN gene body, while there is little change involving the upstream region (Fig. 5e). Together this suggests that intragenic looping may be a mechanism to maintain silencing of TTN within the B compartment in hESCs, and that this is potentially mediated through CTCF³⁴”.*

Comment 7: “In Figures 6c-d, according to a background model of random gene distribution along each chromosome, the association of 16 co-regulated genes is significantly different from the background in CM but not in ESC. However, the randomization of genes does not take into account their A/B compartment identity. It could simply be that those genes get activated and move from B to A in CM, which leads to their observed higher than expected interaction by chance if they are randomly distributed in either A/B. It would therefore be useful to compare this with random sets of genes with the same A/B identity as the 16 genes of interest.”

This is a very important point. Unfortunately, we could not generate a fully paired randomized background because some of the genes we had considered are on the X chromosome, for which we could not compute A/B compartment analysis due to the confounding effect of X inactivation in this female hESC line (as a pair of X chromosomes with very distinct 3D structure co-exist). We thus chose to focus on the genes that are upregulated and are found constitutively in the A compartment, and repeated our random permutation analysis comparing a gene set of 9 such RBM20 targets against the comparable background set of 1988 genes (**updated Supplementary Figure 8f**). Supporting our original findings, we observed an enrichment of *trans* interactions within the gene set in CMs (empirical p-value of 0.07) which is not seen in ESCs (empirical p-value of 0.46). We have now added the following text at **updated lines 288-291**: *“To control for the possibility that this could be simply the result of compartment switching, we repeated the analysis with only those genes that are constitutively in the A compartment against a comparable background gene set and still observed a gain in association (Supplementary Fig. 8f)”.*

On top of this additional computational control, in our revised manuscript we provide extensive experimental validation of the mechanism we had proposed regarding the existence of an RBM20-dependent splicing factory

involving multiple genes interacting in *trans*. This is discussed extensively in the reply to **Comment 9 from Reviewer 1 and Comment 19 from Reviewer 2**, and the data is presented in **updated Figure 6f-g, new Figures 7-10, new Supplementary Figures 9-10**, and in the **updated text at lines 292-384**. Also refer to the updated discussion at **updated lines 422-461**.

Specifically relevant to the important aspect raised by the Reviewer, all of the functional experiments in CMs which weakened the *trans* interactions involving *TTN* and RBM20 target loci did so without strongly affecting the localization of these genes with relation to the nuclear periphery (**new Supplementary Figures 9b-c**). Of note, 3D DNA FISH measurements of the distances between multiple loci and the nuclear periphery were in close agreement with Hi-C-inferred compartmentalization (**updated Figure 6g**), demonstrating their functional relevance. Therefore, we conclude that the *trans* interactions involving *TTN* and RBM20 target loci do not merely result from higher chances of proximity for genes within the same chromatin compartment, but require an active mechanism to be established. Based on the results of our pharmacological and gene editing perturbations we propose that RBM20 foci are indeed needed for such inter-chromosomal associations (**updated Figure 10e**).

Comment 8: "Regarding the definition of A/B compartments, did the authors assign a genome region to be within A if $PC1 > 0$ and B if $PC1 < 0$? Or were there further probabilistic measures involved?"

We apologize for having omitted this from the Methods section; we had indeed defined A and B compartments based on a positive or negative sign for PC1. We have now added this to **updated lines 541-544**: *"Using eigenvalue decomposition of the contact maps PC1 (A/B) scores were calculated using HOMER64 on the valid pairs file at 500Kb resolution and with no additional windowing (super-resolution also set at 500Kb). Bins were assigned to A compartment if their average value between replicates were greater than 0 and B if less than 0"*.

Comment 9: "In Figure 6f, in the example cell for the FISH experiments it looks as if there is one allele that shows co-localisation between the red and green dots, but not the other. It would be interesting to know whether this occurs across the majority of cells or it is just a co-incidence?"

Updated Figure 6f and new Figures 7d and 10a provide more examples of the types of interactions we observed between *TTN* and *CACNA1C*. While in several CMs both of the alleles for these loci are very proximal, in a large number of CMs only one of the two alleles is strongly co-localized. To test whether on average each of the alleles of *TTN* is found in increased proximity with those of other RBM20 target loci in CMs, we calculated the maximal distance between pairs of loci (**updated Supplementary Figure 9a-c**). Similarly for what we observed for the minimal distance (**updated Figures 6g, 7e, and 10b**), cardiomyocyte differentiation led to increased proximity between loci, which was impaired by transcriptional inhibition, loss of *TTN* transcription, or loss of RBM20. These results suggest that while not all CMs show a biallelic interaction between *TTN* and other RBM20 target loci, there is no obvious preference for a monoallelic interaction (which would result in only the minimal distance between loci to be affected by RBM20-dependent interactions). Further, this indicates that the different interaction patterns we observed within a CM population are likely the result of on-off dynamics for these *trans* chromosomal interactions. We speculate that this aspect could depend on the transcriptional activity (and therefore on the amount of splicing) at a given allele. This would be consistent with the model we propose in **updated Figure 10e**, as only the presence of nascent mRNAs would result in increased chances of localization at RBM20 foci of the corresponding locus.

Reviewer 3 minor comments:

Comment 10: “Line 702: Figure legend: The upper value is the number of TADs, and the lower value is the median size. Not sure what the upper/lower values refer to in Figure 3c?”

We apologize for the error: the legend referred to an earlier version of the figure which also indicated the median TAD size. We have revised the figure legend at **updated lines 1132-1134** to read: *“(b) Boxplot of TAD sizes and number within A and B compartments across differentiation for DI method. n represents the number of TADs. (Wilcoxon test ***=p-value < 0.001)”*.

Comment 11: “Lines 219 to 224: Would this be better in the Discussion?”

The relevant section was rewritten during the revision to incorporate the new functional validations of the RBM20-dependent splicing factory. This concept is now expressed at **updated lines 317-319**, where it serves as background to the experiment described thereafter: *“It was previously shown that RBM20 foci in HL-1 immortalized mouse cardiomyocytes are transcription-dependent³⁶, which offers an avenue to test the effect of their disruption”*.

Comment 12: “From line 239: “In contrast, heterochromatic, silent regions in hESCs are relatively accessible compared to differentiated cells but compact during differentiation coincident with increased long range Hi-C signals and a loss of ATAC-peaks. This is similar to results seen in CTCF and cohesin depletion studies where loss of local TAD structure does not alter compartmentalization and in fact strengthens long-range interactions.” I understand that the two observations mentioned above were made independently, but find it hard to follow the logic of why those two observations are similar.”

Our findings regarding chromatin changes in the B compartment during physiological development seem to recapitulate certain aspects of what was observed following genetic perturbations leading to the loss of CTCF or cohesin. This would indicate that during differentiation a decrease of CTCF and/or cohesin activity might be responsible for some of the changes we observe with the B compartment. We have clarified this discussion point at **updated lines 399-406**: *“In contrast, heterochromatin is relatively accessible in hESCs compared to differentiated cells, but compact during differentiation. This coincides with loss of ATAC-peaks and TAD boundaries while long range Hi-C signal increases. This process is similar to that resulting from CTCF or cohesin depletion, which results in loss of local TAD structure but does not alter compartmentalization, and in fact strengthens long range interactions^{47,48}. Thus, we speculate that loss or decrease of CTCF/cohesin activity in B compartment along with a gain in heterochromatin proteins during differentiation may provide the driving force behind compaction.”*

References

- Anderson, D.J., Kaplan, D.I., Bell, K.M., Koutsis, K., Haynes, J.M., Mills, R.J., Phelan, D.G., Qian, E.L., Leitoguinho, A.R., Arasaratnam, D., Labonne, T., Ng, E.S., Davis, R.P., Casini, S., Passier, R., Hudson, J.E., Porrello, E.R., Costa, M.W., Rafii, A., Curl, C.L., Delbridge, L.M., Harvey, R.P., Oshlack, A., Cheung, M.M., Mummery, C.L., Petrou, S., Elefanty, A.G., Stanley, E.G., and Elliott, D.A. (2018). NKX2-5 regulates human cardiomyogenesis via a HEY2 dependent transcriptional network. *Nat. Commun.* 9, 1373.
- Li, Y., He, Y., Liang, Z., Wang, Y., Chen, F., Djekidel, M.N., Li, G., Zhang, X., Xiang, S., Wang, Z., Gao, J., Zhang, M.Q., and Chen, Y. (2018). Alterations of specific chromatin conformation affect ATRA-induced leukemia cell differentiation. *Cell Death Dis.* 9, 200.
- Montefiori, L.E., Sobreira, D.R., Sakabe, N.J., Aneas, I., Joslin, A.C., Hansen, G.T., Bozek, G., Moskowitz, I.P., McNally, E.M., and Nóbrega, M.A. (2018). A promoter interaction map for cardiovascular disease genetics. *Elife* 7, 1–35.
- Pliner, H.A., Packer, J.S., McFaline-Figueroa, J.L., Cusanovich, D.A., Daza, R.M., Aghamirzaie, D., Srivatsan, S., Qiu, X., Jackson, D., Minkina, A., Adey, A.C., Steemers, F.J., Shendure, J., and Trapnell, C. (2018). Cicero Predicts cis-Regulatory DNA Interactions from Single-Cell Chromatin Accessibility Data. *Mol. Cell* 71, 858–871.e8.
- Stadhouders, R., Vidal, E., Serra, F., Di Stefano, B., Le Dily, F., Quilez, J., Gomez, A., Collombet, S., Berenguer, C., Cuartero, Y., Hecht, J., Filion, G.J., Beato, M., Marti-Renom, M.A., and Graf, T. (2018). Transcription factors orchestrate dynamic interplay between genome topology and gene regulation during cell reprogramming. *Nat. Genet.* 50, 238–249.
- Yardımcı, G.G., Ozadam, H., Sauria, M.E.G., Ursu, O., Yan, K.-K., Yang, T., Chakraborty, A., Kaul, A., Lajoie, B.R., Song, F., Zhan, Y., Ay, F., Gerstein, M., Kundaje, A., Li, Q., Taylor, J., Yue, F., Dekker, J., and Noble, W.S. (2018). Measuring the reproducibility and quality of Hi-C data. *BioRxiv* 188755.
- Bonev, B., Mendelson Cohen, N., Szabo, Q., Fritsch, L., Papadopoulos, G.L., Lubling, Y., Xu, X., Lv, X., Hugnot, J.-P., Tanay, A., and Cavalli, G. (2017). Multiscale 3D Genome Rewiring during Mouse Neural Development. *Cell* 171, 557–572.e24.
- Forcato, M., Nicoletti, C., Pal, K., Livi, C.M., Ferrari, F., and Bicciato, S. (2017). Comparison of computational methods for Hi-C data analysis. *Nat. Methods* 14, 679–685.
- Montefiori, L., Hernandez, L., Zhang, Z., Gilad, Y., Ober, C., Crawford, G., Nobrega, M., and Jo Sakabe, N. (2017). Reducing mitochondrial reads in ATAC-seq using CRISPR/Cas9. *Sci. Rep.* 7, 2451.
- Nothjunge, S., Nührenberg, T.G., Grüning, B.A., Doppler, S.A., Preissl, S., Schwaderer, M., Rommel, C., Krane, M., Hein, L., and Gilsbach, R. (2017). DNA methylation signatures follow preformed chromatin compartments in cardiac myocytes. *Nat. Commun.* 8, 1667.
- Schwarzer, W., Abdennur, N., Goloborodko, A., Pekowska, A., Fudenberg, G., Loe-Mie, Y., Fonseca, N.A., Huber, W., Haering, C., Mirny, L., and Spitz, F. (2017). Two independent modes of chromatin organization revealed by cohesin removal. *Nature* 551, 51–56.
- Wang, Y., Fan, C., Zheng, Y., and Li, C. (2017). Dynamic chromatin accessibility modeled by Markov process of randomly-moving molecules in the 3D genome. *Nucleic Acids Res.* 45, e85.
- Wu, P., Li, T., Li, R., Jia, L., Zhu, P., Liu, Y., Chen, Q., Tang, D., Yu, Y., and Li, C. (2017). 3D genome of multiple myeloma reveals spatial genome disorganization associated with copy number variations. *Nat. Commun.* 8, 1937.
- Yang, T., Zhang, F., Yardımcı, G.G., Song, F., Hardison, R.C., Noble, W.S., Yue, F., and Li, Q. (2017). HiCRep: assessing the reproducibility of Hi-C data using a stratum-adjusted correlation coefficient. *Genome Res.* 27, 1939–1949.
- Ang, Y.S., Rivas, R.N., Ribeiro, A.J.S., Srivas, R., Rivera, J., Stone, N.R., Pratt, K., Mohamed, T.M.A., Fu, J.D., Spencer, C.I., Tippens, N.D., Li, M., Narasimha, A., Radzinsky, E., Moon-Grady, A.J., Yu, H., Pruitt, B.L., Snyder, M.P., and Srivastava, D. (2016). Disease Model of GATA4 Mutation Reveals Transcription Factor Cooperativity in Human Cardiogenesis. *Cell* 167, 1734–1749.e22.

- Franke, M., Ibrahim, D.M., Andrey, G., Schwarzer, W., Heinrich, V., Schöpflin, R., Kraft, K., Kempfer, R., Jerković, I., Chan, W.-L., Spielmann, M., Timmermann, B., Wittler, L., Kurth, I., Cambiaso, P., Zuffardi, O., Houge, G., Lambie, L., Brancati, F., Pombo, A., Vingron, M., Spitz, F., and Mundlos, S. (2016). Formation of new chromatin domains determines pathogenicity of genomic duplications. *Nature* 538, 265–269.
- Ramani, V., Cusanovich, D.A., Hause, R.J., Ma, W., Qiu, R., Deng, X., Blau, C.A., Distèche, C.M., Noble, W.S., Shendure, J., and Duan, Z. (2016). Mapping 3D genome architecture through in situ DNase Hi-C. *Nat Protoc.* 11, 2104–2121.
- Schmitt, A.D., Hu, M., Jung, I., Xu, Z., Qiu, Y., Tan, C.L., Li, Y., Lin, S., Lin, Y., Barr, C.L., and Ren, B. (2016). A Compendium of Chromatin Contact Maps Reveals Spatially Active Regions in the Human Genome. *Cell Rep.* 17, 2042–2059.
- Wu, J., Huang, B., Chen, H., Yin, Q., Liu, Y., Xiang, Y., Zhang, B., Liu, B., Wang, Q., Xia, W., Li, W., Li, Y., Ma, J., Peng, X., Zheng, H., Ming, J., Zhang, W., Zhang, J., Tian, G., Xu, F., Chang, Z., Na, J., Yang, X., and Xie, W. (2016). The landscape of accessible chromatin in mammalian preimplantation embryos. *Nature* 534, 652–657.
- Barutcu, A.R., Lajoie, B.R., McCord, R.P., Tye, C.E., Hong, D., Messier, T.L., Browne, G., van Wijnen, A.J., Lian, J.B., Stein, J.L., Dekker, J., Imbalzano, A.N., and Stein, G.S. (2015). Chromatin interaction analysis reveals changes in small chromosome and telomere clustering between epithelial and breast cancer cells. *Genome Biol.* 16, 214.
- Crane, E., Bian, Q., McCord, R.P., Lajoie, B.R., Wheeler, B.S., Ralston, E.J., Uzawa, S., Dekker, J., and Meyer, B.J. (2015). Condensin-driven remodelling of X chromosome topology during dosage compensation. *Nature* 523, 240–244.
- Dixon, J.R., Jung, I., Selvaraj, S., Shen, Y., Antosiewicz-Bourget, J.E., Lee, A.Y., Ye, Z., Kim, A., Rajagopal, N., Xie, W., Diao, Y., Liang, J., Zhao, H., Lobanenkov, V. V., Ecker, J.R., Thomson, J.A., and Ren, B. (2015). Chromatin architecture reorganization during stem cell differentiation. *Nature* 518, 331–336.
- Fraser, J., Ferrai, C., Chiariello, A.M., Schueler, M., Rito, T., Laudanno, G., Barbieri, M., Moore, B.L., Kraemer, D.C., Aitken, S., Xie, S.Q., Morris, K.J., Itoh, M., Kawaji, H., Jaeger, I., Hayashizaki, Y., Carninci, P., Forrest, A.R., Semple, C.A., Dostie, J., Pombo, A., and Nicodemi, M. (2015). Hierarchical folding and reorganization of chromosomes are linked to transcriptional changes in cellular differentiation. *Mol. Syst. Biol.* 11, 852–852.
- Tsankov, A.M., Gu, H., Akopian, V., Ziller, M.J., Donaghey, J., Amit, I., Gnirke, A., and Meissner, A. (2015). Transcription factor binding dynamics during human ES cell differentiation. *Nature* 518, 344–349.
- de Wit, E., Vos, E.S.M., Holwerda, S.J.B., Valdes-Quezada, C., Verstegen, M.J.A.M., Teunissen, H., Splinter, E., Wijchers, P.J., Krijger, P.H.L., and de Laat, W. (2015). CTCF Binding Polarity Determines Chromatin Looping. *Mol. Cell* 60, 676–684.
- Hughes, J.R., Roberts, N., McGowan, S., Hay, D., Giannoulatou, E., Lynch, M., De Gobbi, M., Taylor, S., Gibbons, R., and Higgs, D.R. (2014). Analysis of hundreds of cis-regulatory landscapes at high resolution in a single, high-throughput experiment. *Nat. Genet.* 46, 205–212.
- Maatz, H., Jens, M., Liss, M., Schafer, S., Heinig, M., Kirchner, M., Adami, E., Rintisch, C., Dauksaite, V., Radke, M.H., Selbach, M., Barton, P.J.R., Cook, S.A., Rajewsky, N., Gotthardt, M., Landthaler, M., and Hubner, N. (2014). RNA-binding protein RBM20 represses splicing to orchestrate cardiac pre-mRNA processing. *J. Clin. Invest.* 124, 3419–3430.
- Li, S., Guo, W., Dewey, C.N., and Greaser, M.L. (2013). Rbm20 regulates titin alternative splicing as a splicing repressor. *Nucleic Acids Res.* 41, 2659–2672.
- Sanyal, A., Lajoie, B.R., Jain, G., and Dekker, J. (2012). The long-range interaction landscape of gene promoters. *Nature* 489, 109–113.
- Lieberman-Aiden, E., van Berkum, N.L., Williams, L., Imakaev, M., Ragoczy, T., Telling, A., Amit, I., Lajoie, B.R., Sabo, P.J., Dorschner, M.O., Sandstrom, R., Bernstein, B., Bender, M.A., Groudine, M., Gnirke, A., Stamatoyannopoulos, J., Mirny, L.A., Lander, E.S., and Dekker, J. (2009). Comprehensive mapping of long-range interactions reveals folding principles of the human genome. *Science* 326, 289–293.

Reviewers' Comments:

Reviewer #1:

Remarks to the Author:

The authors have considerably revised their manuscript to address all reviewer queries. I'm happy to support publication.

Reviewer #2:

None

Reviewer #3:

None

Point by point answer to the reviewers

Dynamics of genome reorganization during human cardiogenesis reveal an RBM20-dependent splicing factory

Bertero, Fields et al.

We are pleased to see that our revision was well-received by all three reviewers. We have performed further minor revision of the text and figures to address the remaining comments from reviewer 2. Further, we modified the text according to the editorial requirements for publication.

Reviewer 1

The authors have considerably revised their manuscript to address all reviewer queries. I'm happy to support publication.

We thank the reviewer for his/her feedback which helped to improve our manuscript.

Reviewer 2

The authors have done an impressive job at this revision. The submitted piece is now quite compelling. As commented before, the concept of “splicing factories” is really interesting, and there is now empirical evidence offered in this revision. Specifically, the dissection using experiments with TTN-delta-prom, compared to TTN-KO and RBM20- KO, and thereby showing which of these RBM20 foci depends upon, is really commendable.

Nonetheless the conclusions of “strongest gains in long-range intra-chromosomal interactions associated with B compartment”, and “switch as a result of gain in long range B-B interactions during differentiation” will need validation from other groups over time.

I would still recommend the following detail technical edits:

We thank the reviewer for the laudatory comments and additional suggestions for improvement; we have address them all as described below. Please note that while the revised manuscript Word file includes track changes as *per* editorial requirements, the line numbering references below apply to the document read in modality “no markup” (continuous line numbering).

Methods:

1. Line 530: "excluding pairs less than 1kb". Please be more specific on the definition on pairs. Di-tag length?

Hi-Pro is able to filter out reads that map close together such that they may be a product of self-ligation of the same fragment and not a result of proximity ligation of distal fragments. To increase the robustness of our HiC data analysis, we thus filtered paired-end reads that mapped within 1 Kb of each other. To clarify this point, we have now rephrased **line 541** to say “*excluding read pairs that mapped within 1 Kb*”.

2. Line 532: "ICE balanced matrix". Please cite this normalization method.

The ICE balancing is included in the HiC-Pro package cited in the prior line (**line 540; reference 56** to Servant et al 2015). To clarify this point **reference 56** is now also mentioned on **line 542**.

3. Line 534: "Heatmaps for all Hi-C data were generated through Cooler". Are the hi-C interaction counts used for heatmap generation also ICE-balanced? It wasn't stated very clearly in the method.

The cooler package includes its own balancing method based on a similar ICE algorithm, which was used to generate heatmaps in cooler starting from the raw counts. Of note, these heatmaps were only used for visualization purposes. We have now included on **lines 545-547**: *“Heatmaps for all Hi-C data were generated through Cooler (<https://github.com/mirnylab/cooler>), based on the raw counts as cooler includes its own ICE balancing, and were used only for visualization purposes”*.

4. Line 536: “HiC-Rep scores were calculated using a resolution of 500Kb with a max distance of 5Mb and h=1”. What is the significance of using HiC-REP? The relevance or importance seems negligible in the result section. Suggest to remove this.

The inclusion of clustering with both HiC-Rep and PC1 scores was to take advantage of two orthogonal methods to demonstrate that the HiC assay is reproducible across biological replicates and separates samples by the differentiation stage. HiC-Rep measures aspects of HiC data reproducibility which may not be fully captured by the analysis of chromatin compartmentalization (Yardımcı et al., 2018). Thus, while we appreciate the reviewer’s suggestion to simplify, we believe the inclusion of HiC-Rep strengthens our conclusions and should remain in the manuscript as it could prove informative for the more specialist readers. To clarify this aspect, we have changed the text at **lines 95-97** to say: *“Using t-SNE to visualize and cluster in two dimensions either PC1 scores or HiC-Rep scores¹³ closely pairs replicates while generating a differentiation trajectory, demonstrating the reproducibility of the assay”*.

5. Line 539: “PC1 (A/B) scores were calculated using HOMER”. Again, Are the hi-C interaction counts used here also ICE-balanced? This is important to ensure consistency in the analysis. Please comment.

HOMER includes its own normalization and takes in a set of valid interactions, which were generated by HiC-Pro. This is the list of all pairs of reads after filtering for distance and MAPQ score. To clarify this, we have now added on **lines 552-554**: *“The valid pairs file from HiC-Pro (after filtering for distance and MAPQ score) was used as input into HOMER⁵⁷ for eigenvalue decomposition of the contact maps to calculate PC1 (A/B) scores at 500 Kb resolution and with no additional windowing (super-resolution also set at 500 Kb)”*.

6. Line 541: Have the authors tabulated number of bins with PC1 values of 0? Any statistics? If they exist, how were the “A” and “B” status assigned?

We can confirm that no bins had a value of exactly 0 and thus as all bins could be assigned to either A or B compartment.

7. Line 544: The test is rather confusing. Were the ANOVA test 1-way or 2-way ANOVA? Was it carried out across the differentiation or pairwise? How do the authors control for false positive? If the authors do not control for false positive, please comment on the confidence of the finding.

We performed a one-way ANOVA test since only one independent variable was being tested: the time point of differentiation. We compared the variance of PC1 scores across differentiation to test the significance of the observed changes relative to the variance within the biological replicates at each time point. We have clarified this on **line 557-560**: *“Switching A/B compartments were determined by a one-way ANOVA p-value < 0.05 (2 replicates per time point, across the 4 time points of differentiation) and at least one time point having an average PC1 score greater than 0 and at least one less than 0”*.

As described in detail in our earlier point-by-point reply, there is currently no gold standard to determine A/B compartment changes. Our approach of combining a significance cutoff for PC1 scores and then further filtering for regions where the average PC1 score changed from positive to negative (or vice versa) is more stringent than other approaches that either used no cut-off except for a PC1 sign change (Wu et al., 2017; Barutcu et al., 2015) or an arbitrary PC1 score change (Nothjunge et al., 2017). Indeed, we aimed to reduce the burden of false positive regions where PC1 sign changes merely result from small fluctuations around 0 (thus representing unclear compartment calls). Consistently with a previous study that followed this same approach (Dixon et al., 2015), we did not include a multiple hypothesis correction for the ANOVA test.

8. Line 557: Please cite ggplot2 for "geom_smooth"

We have now included the citation for ggplot2 at **line 571 (reference 58)**.

9. Line 559: Did the authors use 3 TAD-calling methods? "Directionality Index (DI) method and domain call pipeline and insulation score method".

We used two distinct methods: the Directionality Index approach (DI), and the insulation score method. The domain call algorithm is part of the DI method pipeline, per the software package. To clarify the point we have changed the text at **lines 573-574** to say: *"TADs were determined using two distinct methods: (1) Directionality Index (DI) method and domain call pipeline¹⁷; (2) insulation score method¹⁸ at 40 Kb resolution"*.

10. Line 563: Line 556: "within 80kb" and "within 200kb". Are the authors referring to +80kb from the boundary or +40kb from the boundary?

These analyses were performed considering overlaps in windows of ± 80 Kb and ± 200 Kb. We have corrected the methods section accordingly at **lines 577 and 580**, respectively.

11. Line 563: Also, please comment on the choice of a less stringent distance cut-off (200kb, instead of 80kb) used here for "time point boundaries" overlap since the authors reported based on these hiC data that "TAD boundaries are highly constitutive across differentiation timepoint"(Line 164). Is this self-fulfilling prophecy?

We chose such stringent for this analysis since all currently available TAD callers are subject to local noise which strongly affects the precise location of the boundary (Forcato et al., 2017). This is described in detail in Schmitt et al., 2016: *"Merging of adjacent boundary bins was performed because often times larger TAD boundaries (up to 400Kb) may result in slightly shifted (by a few bins) boundary calls between samples, and though they do not directly overlap, then both are a bin within the same boundary region. Moreover, in previous reports, TAD boundaries have been defined as 40-400Kb (Dixon et al., 2012) while regions >400kb are characterized as regions of "disorganized chromatin"*.

We recognize that the use of a stringent cutoff for boundary changes might have led to some false negatives. However, we determined it was more important to focus on the most significant changes to reduce the burden of false positives. To clarify the point on **lines 166-167**, we have now rephrased it to clarify that a stringent cut-off was used: *"Using a stringent cut-off, both methods show a majority of TAD boundaries that are constitutive across differentiation (Fig. 3a)"*.

12. Line 571: what was the test used for "differential boundary" analysis.

This analysis relied on determining the absence/presence of boundary overlaps across time points, and thus no statistical test was used as a null model for this phenomenon could not be defined. We defined boundaries shared between two time points when these were found within a 400 Kb window. On the contrary, differential boundaries are those that did not have nearby a TAD boundary in the other time point. We have sought to clarify this on **lines 579-582**: *"For comparison between time points boundaries were considered shared if there was a boundary within ± 200 Kb, time point specific boundaries were identified by the lack of a shared boundary with the respective comparison"*.

13. Line 557: what was the multiple-testing control on the p-value.

No multiple-testing control was applied; please refer to the response to point 7 above explaining this analysis in additional detail.

14. Line 583: Besides "chromosomal distribution", are there any other potential factors that might bias the permutation test? Also, please define chromosomal distribution: distance? number of genes per region?

We refer to chromosomal distribution as the same number of genes per chromosome. Since we are focusing exclusively on *trans* interactions, we did not control for the distance between the genes on the chromosome. This is consistent with Witten and Noble, 2012, which showed that this is the best method to evaluate *trans* interactions. We have amended the text at **lines 597-600** to clarify this: *"The background model was based on a set of 1,000 random permutations, each time selecting a new set of genes with the same number of genes per chromosome as the seed set, similar to an earlier report⁶⁰, and totaling the respective ICE balanced counts"*.

As part of our previous revision we performed an additional permutation analysis to control for any changes associated with upregulation or B-A transitions by restricting the test to only those genes that are constitutively A and upregulated, and found a similar trend (**Supplementary Figure 8f**).

15. Line 602: "P-values were calculated based on a Chi-squared test between the total number of genes in a type of compartment transition versus the number of differentially expressed genes within that region (sub-divided by time point of peak expression)." It is rather confusing here. Firstly, are both repressed and over-expressed genes counted separately towards the "total number of genes"? Secondly, it is not clear what is meant by "sub-divided by time point of peak expression".

For this test we compared each type of transition (B-A, A-B, B-A-B, A-B-A) with the genes that are differentially expressed and peaking at each time point (ESC, MES, CP, CM). For example, we performed a Chi-squared test to see if there is significant overlap of: (1) genes that are differentially expressed with peak expression in CMs; and (2) genes that are in B-A regions (**Figure 2c**). All raw values are also included in the **Source Data File**. We have clarified the text at **lines 618-621**: *"P-values were calculated by a Chi-squared test between the total number of genes in a type of compartment transition relative to the number of differentially expressed genes within that region. For example, comparing the number of genes in B to A regions with the number of genes upregulated at CM"*.

16. Line 606: Was the p-value calculated using Wilcoxon test.

Yes, this is stated on **lines 621-623**: *"Gene sizes and gene distances were calculated from the ensemble hg38 annotation. P-values were calculated by the Wilcoxon test in R"*.

17. Line 610: Are these proteins measured specifically in cardiomyocytes?

We did not perform these measurements: this comparison was performed by taking advantage of previously published publicly available data from 37 adult tissues. The results described on **lines 238-240** pertain to expression in adult heart, rather than isolated cardiomyocytes. Our finding that *"those genes that are upregulated and go from B to A are more heart-enriched compared to other upregulated genes"* would likely be even more significant if we had data from adult cardiomyocytes. However, that data was not available in the cited study, and we lacked access to healthy cardiac tissue that would be required for this experiment.

We have now added for clarity to **lines 623-625**: *"For comparison of genes across adult tissues including adult heart from the EMBL Protein Atlas, RNA expression values were downloaded from EMBL-EBI (E-MTAB-2836)"*.

Figures:

1. Figure 1b: What is the numerical scale for "high" and "low"?

Cooler generates normalized values, such that each row and column sums to 1. The values are plotted on a log scale, ranging from -1 (high) to -4.5 (low). We have now included the corresponding values to the heatmap legend of **Figure 1b**.

2. Figure 1i: What are the rows and columns? Are they different stages of differentiation? These are not labeled.

As described in the legend, each bin represents a decile of the genome by PC1 score. The figure is a symmetric matrix of the \log_2 observed/expected value comparing CM to hESCs. To clarify this we have now added to **Figure 1i** the text: "*Decile PC1 score.*" This was also done for **Supplementary Figure 3d-f**, which show the same type of plots.

3. Figure 2d: "Benjamini p-value": have they been log-transformed?

While the axis was log-transformed to better display the data, the p-value indicated is the actual p-value output from the GO analysis. To clarify this point, we have added to the legend for **Figure 2d** the text (**lines 1119-1120**): "*(d) GO term enrichment in CM peak expression genes in B to A compartments, p-values plotted on log scale.*"

4. Figure 3e: Again, please show the numerical scale for "high" and "low".

We have now included values corresponding to high (-1) and low (-4) to the heatmap legend of **Figure 3e**.

5. Figure 4g: The enrichment of TBX5 and NKX2-5 are very different from genome-wide background. What did the authors use as background, how do they prove that this observation is not just by chance?

This figure indicates that the overlap between GATA4 peaks and NKX2-5 and TBX5 peaks was greater in B to A regions than in the rest of the genome. A Chi-squared test indicates a significant difference in such overlaps based on an observed to expected ratio, and therefore we reject the null hypothesis that there is no difference in overlap based on genomic region. This test considers the underlying distribution of the ChIP data to show significant overlap.

6. Figure 6a: The medians don't look different. Yet the p-values are significant? Have the source data been shown somewhere?

The source data is included in the **Source Data File**, which confirms that the median values are different, albeit slightly. Nevertheless, as already discussed in our previous point-by-point reply, these modest differences result in highly significant p-values because the data includes all *trans* contacts for the *TTN* locus, which represents thousands of data points. Moreover, the key conclusion of this analysis is that *TTN trans* contracts switch from being slightly enriched within the B compartment in hESCs to being clearly enriched in the A compartment in CM. We have now included the median z-scores in the text to emphasize this point at updated **lines 276-277**: "*(Fig. 6a, hESC median Z-score A: -0.26 B: -0.07, CM median Z-score A: 0.19, B: -0.29)*".

7. Suggest that the legend of Figure 6(c-d) needs re-phrasing.

We have clarified the legend at **lines 1167-1171** to read: "*(c-d) Association of upregulated RBM20 target genes in hESCs (c) and CM (d). Dashed red line indicates cumulative sum of ICE normalized Hi-C read*"

between target genes in trans, histogram represent the background of 1000 random permutations of selected genes from similar chromosomal distribution".

Results:

1. Line 97: "Fetal heart Hi-C most closely resembles in vitro cardiomyocytes but clusters separately, likely reflective of lower cardiomyocyte purity." The fetal hearts are clearly clustered with the derived cardiomyocytes in both t-sne and clustering plot. It is unclear how the authors have attributed the difference to cell-type contamination. Please comment.

While there is similarity with respect to the x-axis of the t-SNE plot in **Figure 1c** for CM and Fetal samples, they separate further on the y-axis than do the replicates for CM or Fetal samples. We have stated that this is "likely reflective of lower cardiomyocyte purity." While this could also reflect a different development stage or other biological/technical factors, we believe that a difference in cardiomyocyte purity is the most likely explanation as it is well established that the heart has a large number of non-myocytes (Zhou and Pu, 2016; Banerjee et al., 2007).

2. Line 105: "Most of these changes are unidirectional (B-A or A-B)." Dixon et al (Nature ,2015) also noticed a 4%-25% compartment changes in differentiation of ES to different lines. However, there seems to be more A->B changes rather than B->A unlike what has been reported in this study. In figure 1g, >60% of the compartment switches are from B->A. Please comment if this is an artifact from resolution issue, or this is a biological meaningful observation.

This is not surprising as in Dixon et al. the authors both used a different cell line and examined a distinct type of differentiation. Interestingly, another study found more regions transitioning from B to A than A to B in a terminal endothelial differentiation (Niskanen et al., 2018). As we were able to validate many of our findings on compartmentalization changes in a second hPSC line, we conclude that the differences in A/B compartment transitions between our study and Dixon et al. likely reflects biologically meaningful observations across distinct lineages.

3. Line 114-117: This finding through hi-C data has been reported in Bonev et al(Cell, 2017). They also reported that the interaction strength between A compartments are decreased, while contacts within the B compartment became stronger between neuronal cells differentiated from stem cell. Please cite.

We have now included the reference to Bonev et al., 2017. In such work they used a mouse differentiation system: the fact that they showed similar findings suggests that this may be a general phenomenon of differentiation. We have edited the text at **lines 111-115** to state: "*By integrating the A/B compartment information across differentiation with the interaction contact maps, we noticed that many of the strongest gains in long-range intra-chromosomal (cis) interactions are associated with the B compartment (Supplementary Fig. 3c), this is consistent with prior studies in mouse neuronal specification¹⁵, suggesting it may be a general phenomenon of differentiation*".

4. Line 128-156: Result section: "Activated genomic regions occur coincident with upregulation of expression." The authors missed the opportunity to quantitatively link the percentage of the compartment switches to gene expression changes. If they have, how many percent of these genes are involved in cardiac differentiation. Likewise, how many percent of the differentially expressed genes are found in compartment switching? These statistics will be more evident to suggest if chromatin architecture plays any role in cell differentiation. This piece of information is glaring omission from an important result in the paragraph here.

The overlap between differentially expressed genes and genes found in switching compartments was reported in the **Source Data Table for Figure 2c**, and is now found in the matching worksheet in the condensed **Source Data File**. This information was the input used to generate **Figure 2c**, in which we calculated the significance of the overlaps between these gene sets based on Chi-squared test. We chose this representation

over multiple Venn diagrams because it conveys the same information in a visually compact manner, while also facilitating the comparison of enrichments across multiple data sets.

Among other things, we show that there is a statistically significant enrichment for differentially expressed genes peaking in CM in genes located in B to A compartment. **Figure 2d** confirms that these genes are enriched in factors involve in cardiac development. Further, we have shown in **Supplemental Tables 4 and 5** that genes that are upregulated in CM and found in B to A compartments are more cardiac specific then those that are upregulated in CM in general.

To clarify and highlight this overlap based on the reviewer's feedback, we have included a new Venn diagram in **Supplemental Figure 4b** demonstrating the substantial overlap between upregulated genes and genes in B to A compartment. While due to space constraints we cannot present similar Venn diagrams for all other types of overlaps between gene sets, this data is reported in the **Source Data File** in the worksheet for Figure 2c.

5. Line 164: Are the constitutive TAD boundaries CTCF-enriched / Cohesin-enriched?

6. Line 167: Are the gained/lost boundaries enriched for CTCF?

To address points 5 and 6, we have analyzed the pattern of CTCF binding in relation to TAD boundaries determined by the DI or insulation score methods in hESCs or CM. We find that that CTCF is more enriched in the proximity (< 1 Kb) of constitutive TAD boundaries compared to gained/lost boundaries. On the other hand, as expected more than 50% of boundaries have a CTCF peak within 1 Kb irrespective of the boundary class or TAD calling method. We have included this new analyses in **Supplemental Figure 6e**, and described these results in the text at **lines 171-175**: *“Consistent with prior studies¹⁹, we find that TAD boundaries are located near to CTCF peaks in both hESCs and CMs with ~70% of TAD boundaries having a CTCF peak with 1 Kb (Supplementary Fig. 6e). Moreover CTCF peaks are located closer on average to constitutive boundaries rather than differential boundaries by both TAD methods (Supplementary Fig. 6e, chi-sq test, p < 0.001)”*.

7. Line 173: Do the authors also have another example to support the claim that gain of TAD boundary is associated with increased genes expression?

We have included in **Supplementary Figure 6g** a gene track of the genomic locus containing the *KCNN2* gene: this shows a gain of TAD boundary (based on both the DI and insulation score method) proximal to *KCNN2* and coincident with its upregulation.

8. Line 184: How do the authors explain the reducing number of ATAC peaks or open chromatin region with the increasing number of A compartment? It is contradictory on the face of it.

We would like to point out that the percentage of the genome found in the A compartment in hESC and CM is not substantially different, and approximately of 50% (**Figure 1d**). Moreover, the reduced number of ATAC peaks in CM involves the whole genome (both A and B compartments), while we show that there is a strong reduction of ATAC peaks in CM specifically in regions that are constitutively in the B compartment or transition from A to B in CM (**Figure 4c**). Thus, the results indicate an overall decrease in accessibility of the B compartment during differentiation, which is consistent with an increase in interaction strength within the B compartment (**Figure 1i**). This is described on **lines 194-195** where we state: *“This supports the model of increased heterochromatin packing in CM coincident with decreased accessibility”*.

9. Line 228: It is interesting to find that the up-regulated genes are longer for those that are involved in B->A compartment switching. Have the authors performed any stats test to help draw any significant conclusions on this?

We have indeed performed statistical tests to show that CM-upregulated genes in B to A compartments are in general larger than other upregulated genes (**Figure 5a**). These genes appear to be lineage specific, structural genes which tend to be larger than smaller transcription factor-type genes.

10. Line 253: Have the authors attempted to look into their hi-C data at higher resolution such as 10kb to observe if CTCF loops are present?

As has been shown in other studies (such as Bonev et al., 2017; Rao et al., 2014), generation of high quality contact maps at 10 Kb resolution from conventional HiC data requires a sequencing depth greater by an order of magnitude than the one we adopted (which is however ideal for assessment of large-scale chromatin organization changes; Belton et al., 2012). As discussed in our earlier point-by-point reply, the focus of the present study was on global chromatin organization changes during human cardiogenesis, and on their relationship with chromatin accessibility and gene expression dynamics. Therefore, the determination of chromatin loops is beyond the scope of this manuscript, while this has been the subject of other excellent reports whose goal was to look at this fine level of chromatin structure regulation by using promoter capture Hi-C (Montefiori et al., 2018; Gilsbach et al., 2018; Choy et al., 2018) or deeply sequenced Hi-C (Zhang et al., 2018). All of these manuscripts are cited in our updated paper.

11. Line 375: Would the authors comment on whether knocking out of *CAMK2D* and *CACNA1C* would show any effect on the RBM20 foci, similarly to how *TTN* KO has shown.

Our expectation is that impairment of *CAMK2D* or *CACNA1C* expression would have a small effect, if any, on RBM20 foci. Indeed, compared to the *TTN* mRNA, which possesses > 100 RBM20 binding sites leading to exclusion of >100 exons, the *CAMK2D* and *CACNA1C* mRNAs have only a few RBM20 binding sites and, accordingly, a much more limited regulation of alternative splicing (Maatz et al., 2014). Thus, our model is one by which *TTN* is the main nucleating factor for RBM20 foci due to its peculiar splicing regulation which leads to highly cooperative RBM20 binding. Nevertheless, we agree that in future studies it will be interesting to test the functional effect of impairing transcription one or more other RBM20 target on the activity of the RBM20 splicing factory.

Finally, on the intriguing conclusion of a splicing factory, orchestrated in the author's example by RBM20, I am not sure if the following possibility is discussed. There may indeed be a splicing factory, but RBM20 may also just be taking advantage of the chromatin architecture to co-splice other genes. It gets nucleated by Titin mRNA then while there, splices other genes and in the process of doing that brings them even closer. If the message is that RBM-20 orchestrates the trans-interaction, then another Hi-C after RBM-20 knockout will be needed to show the loss of the trans-interaction. Yes, the DNA FISH shows some level of separation after the KO experiments, but these genes are already located in proximity in the first place. Such a Hi-C/Chromatin architecture dataset may also serve as way to predict which genes are co-spliced rather than the RBM-20 determines which genes interact together.

The interpretation of our data proposed by the reviewer is indeed in line with our model, according to which the formation of the RBM20 splicing factory is both the result of chromatin compartment transitions involving *TTN* and other RBM20 targets, and the formation of RBM20 foci (**Figure 10e**). As acknowledged by the reviewer, this model is supported by extensive validation using DNA 3D FISH after impairment of the RBM20 splicing factory by either knockout of RBM20 or of the *TTN* promoter. We agree that building upon our results it will be interesting in future studies to generate genome-wide chromatin interaction data (such as 4C or promoter-capture HiC) to explore the *trans* interactions involving the RBM20 splicing factory, as well as other yet-to-be-defined TIDs. This would be, however, clearly beyond the scope of the current, already extensively revised study.

Overall, an impressive revision which has clearer novelty within now.

We thank the reviewer for his/her feedback which helped to improve our manuscript.

Reviewer 3

I have been through the paper and the authors response to the referees. I think the authors have made a very good effort to address all of the points made by the referees and I would be happy to recommend publication.

There are still many questions as to the significance of the changes, and what is causing what, but I think they are inherent in the approach and this field at the moment and require more definitive experiments. The authors will be making a large amount of data available that others can analyse and they deserve to publish their work.

We thank the reviewer for his/her feedback which helped to improve our manuscript.

References

- Banerjee, I., J.W. Fuseler, R.L. Price, T.K. Borg, and T.A. Baudino. 2007. Determination of cell types and numbers during cardiac development in the neonatal and adult rat and mouse. *Am. J. Physiol. Circ. Physiol.* 293:H1883–H1891. doi:10.1152/ajpheart.00514.2007.
- Barutcu, A.R., B.R. Lajoie, R.P. McCord, C.E. Tye, D. Hong, T.L. Messier, G. Browne, A.J. van Wijnen, J.B. Lian, J.L. Stein, J. Dekker, A.N. Imbalzano, and G.S. Stein. 2015. Chromatin interaction analysis reveals changes in small chromosome and telomere clustering between epithelial and breast cancer cells. *Genome Biol.* 16:214. doi:10.1186/s13059-015-0768-0.
- Belton, J.-M., R.P. McCord, J.H. Gibcus, N. Naumova, and Y. Zhan. 2012. Hi-C: A comprehensive technique to capture the conformation of genomes. *Methods.* 58:268–276. doi:10.1016/J.YMETH.2012.05.001.
- Bonev, B., N. Mendelson Cohen, Q. Szabo, L. Fritsch, G.L. Papadopoulos, Y. Lubling, X. Xu, X. Lv, J.-P. Hugnot, A. Tanay, and G. Cavalli. 2017. Multiscale 3D Genome Rewiring during Mouse Neural Development. *Cell.* 171:557–572.e24. doi:10.1016/j.cell.2017.09.043.
- Choy, M.K., B.M. Javierre, S.G. Williams, S.L. Baross, Y. Liu, S.W. Wingett, A. Akbarov, C. Wallace, P. Freire-Pritchett, P.J. Rugg-Gunn, M. Spivakov, P. Fraser, and B.D. Keavney. 2018. Promoter interactome of human embryonic stem cell-derived cardiomyocytes connects GWAS regions to cardiac gene networks. *Nat. Commun.* 9. doi:10.1038/s41467-018-04931-0.
- Dixon, J.R., I. Jung, S. Selvaraj, Y. Shen, J.E. Antosiewicz-Bourget, A.Y. Lee, Z. Ye, A. Kim, N. Rajagopal, W. Xie, Y. Diao, J. Liang, H. Zhao, V. V. Lobanenko, J.R. Ecker, J.A. Thomson, and B. Ren. 2015. Chromatin architecture reorganization during stem cell differentiation. *Nature.* 518:331–336. doi:10.1038/nature14222.
- Forcato, M., C. Nicoletti, K. Pal, C.M. Livi, F. Ferrari, and S. Bicciato. 2017. Comparison of computational methods for Hi-C data analysis. *Nat. Methods.* 14:679–685. doi:10.1038/nmeth.4325.
- Gilsbach, R., M. Schwaderer, S. Preissl, B.A. Grüning, D. Kranzhöfer, P. Schneider, T.G. Nührenberg, S. Mulero-Navarro, D. Weichenhan, C. Braun, M. Dreßen, A.R. Jacobs, H. Lahm, T. Doenst, R. Backofen, M. Krane, B.D. Gelb, and L. Hein. 2018. Distinct epigenetic programs regulate cardiac myocyte development and disease in the human heart in vivo. *Nat. Commun.* 9. doi:10.1038/s41467-017-02762-z.
- Maatz, H., M. Jens, M. Liss, S. Schafer, M. Heinig, M. Kirchner, E. Adami, C. Rintisch, V. Dauksaite, M.H. Radke, M. Selbach, P.J.R. Barton, S.A. Cook, N. Rajewsky, M. Gotthardt, M. Landthaler, and N. Hubner. 2014. RNA-binding protein RBM20 represses splicing to orchestrate cardiac pre-mRNA processing. *J. Clin. Invest.* 124:3419–3430. doi:10.1172/JCI74523.
- Montefiori, L.E., D.R. Sobreira, N.J. Sakabe, I. Aneas, A.C. Joslin, G.T. Hansen, G. Bozek, I.P. Moskowitz, E.M. McNally, and M.A. Nóbrega. 2018. A promoter interaction map for cardiovascular disease genetics. *Elife.* 7:1–35. doi:10.7554/eLife.35788.
- Niskanen, H., I. Tuszyńska, R. Zaborowski, M. Heinäniemi, S. Ylä-Herttua, B. Wilczynski, and M.U. Kaikkonen. 2018. Endothelial cell differentiation is encompassed by changes in long range interactions between inactive chromatin regions. *Nucleic Acids Res.* 46:1724–1740. doi:10.1093/nar/gkx1214.
- Nothjunge, S., T.G. Nührenberg, B.A. Grüning, S.A. Doppler, S. Preissl, M. Schwaderer, C. Rommel, M. Krane, L. Hein, and R. Gilsbach. 2017. DNA methylation signatures follow preformed chromatin compartments in cardiac myocytes. *Nat. Commun.* 8:1667. doi:10.1038/s41467-017-01724-9.
- Rao, S.S.P., M.H. Huntley, N.C. Durand, E.K. Stamenova, I.D. Bochkov, J.T. Robinson, A.L. Sanborn, I. Machol, A.D. Omer, E.S. Lander, and E.L. Aiden. 2014. A 3D map of the human genome at kilobase resolution reveals principles of chromatin looping. *Cell.* 159:1665–1680. doi:10.1016/j.cell.2014.11.021.
- Schmitt, A.D., M. Hu, I. Jung, Z. Xu, Y. Qiu, C.L. Tan, Y. Li, S. Lin, Y. Lin, C.L. Barr, and B. Ren. 2016. A Compendium of Chromatin Contact Maps Reveals Spatially Active Regions in the Human Genome. *Cell Rep.* 17:2042–2059. doi:10.1016/j.celrep.2016.10.061.
- Witten, D.M., and W.S. Noble. 2012. On the assessment of statistical significance of three-dimensional colocalization of sets of genomic elements. *Nucleic Acids Res.* 40:3849–55. doi:10.1093/nar/gks012.
- Wu, P., T. Li, R. Li, L. Jia, P. Zhu, Y. Liu, Q. Chen, D. Tang, Y. Yu, and C. Li. 2017. 3D genome of multiple myeloma reveals spatial genome disorganization associated with copy number variations. *Nat. Commun.* 8:1937. doi:10.1038/s41467-017-01793-w.
- Yardımcı, G.G., H. Ozadam, M.E.G. Sauria, O. Ursu, K.-K. Yan, T. Yang, A. Chakraborty, A. Kaul, B.R. Lajoie, F. Song,

- Y. Zhan, F. Ay, M. Gerstein, A. Kundaje, Q. Li, J. Taylor, F. Yue, J. Dekker, and W.S. Noble. 2018. Measuring the reproducibility and quality of Hi-C data. *bioRxiv*. 188755. doi:10.1101/188755.
- Zhang, Y., T. Li, S. Preissl, J. Grinstein, E. Farah, E. Destici, A.Y. Lee, S. Chee, Y. Qiu, K. Ma, Z. Ye, Q. Zhu, H. Huang, R. Hu, R. Fang, S. Evans, N. Chi, and B. Ren. 2018. 3D Chromatin Architecture Remodeling during Human Cardiomyocyte Differentiation Reveals A Novel Role of HERV-H In Demarcating Chromatin Domains. *bioRxiv*. 485961. doi:10.1101/485961.
- Zhou, P., and W.T. Pu. 2016. Recounting Cardiac Cellular Composition. *Circ. Res.* 118:368–70. doi:10.1161/CIRCRESAHA.116.308139.